# Partition First, Embed Later: Laplacian-Based Feature Partitioning for Refined Embedding and Visualization of High-Dimensional Data

## Abstract

The utility of embedding and visualization techniques for high-dimensional data in exploratory analysis is well-established. However, when the data embody intricate structures governed by multiple latent variables, standard techniques may distort or even mask part of the phenomenon under study. This paper explores scenarios where the observed features can be partitioned into mutually exclusive subsets, each capturing a different smooth substructure. In such cases, visualizing the data based on each feature partition can better characterize the underlying processes and structures in the data, leading to improved interpretability. To partition the features, we propose solving an optimization problem that promotes a graph Laplacian-based smoothness in each partition, thus prioritizing partitions with simpler geometric structures. Our approach generalizes traditional embedding and visualization techniques such as t-distributed Stochastic Neighbor Embedding and Diffusion Maps, allowing them to learn multiple embeddings simultaneously. We establish that if several independent or partially dependent manifolds are embedded in distinct feature subsets in high-dimensional space, then our framework can reliably identify the correct subsets with theoretical guarantees. Finally, we demonstrate the effectiveness of our approach in extracting multiple low-dimensional structures and partially independent processes from both simulated and real data.

Visualization and embedding techniques play a pivotal role in extracting scientific insights from high-dimensional data. Such methods aim to reduce the dimensionality of a dataset while preserving its underlying structural characteristics. Specifically, given $N$ observations with $D$ features, $\{\boldsymbol{y}_i\}_{i=1}^N \subset \mathbb{R}^D$, standard techniques such as Laplacian Eigenmaps (Belkin & Niyogi, 2003), Diffusion Maps (Coifman & Lafon, 2006), t-distributed Stochastic Neighbor Embedding (tSNE) (Van der Maaten & Hinton, 2008), and UMAP (McInnes et al., 2018), first construct an $N \times N$ similarity graph between the given data points using all $D$ features. Then, they embed the $N$ graph nodes in a low-dimensional space where certain structural characteristics of the graph are preserved.

However, when the underlying structure of the data is highly complex, e.g., it is governed by many latent variables, standard methods may fail to capture and embed the data's structure accurately. First, complex structures may require prohibitively large sample sizes to guarantee that the similarity graph reliably represents the underlying geometry. For instance, if the data points are sampled from a manifold, then the number of samples $N$ required for the graph Laplacian to accurately approximate its population counterpart grows exponentially in the intrinsic dimension (Hein et al., 2005; Singer, 2006). Second, even if the graph reliably represents the data's geometry, embedding and visualization of complex data in low-dimensional space can be restrictive. In particular, data visualizations with tSNE and UMAP in two or three dimensions may be severely distorted if the data's underlying geometry is higher-dimensional or cannot be effectively embedded in these dimensions (Chari & Pachter, 2023; Kohli et al., 2021). For Laplacian Eigenmaps and Diffusion Maps, since they rely on the eigenvectors of the graph Laplacian, there is a degree of redundancy in their embedding (Blau & Michaeli, 2017) that grows significantly with the number of underlying latent variables. Consequently, to reliably capture the structure of the data, the embedding dimension using these methods may need to exceed the data's intrinsic dimension considerably.

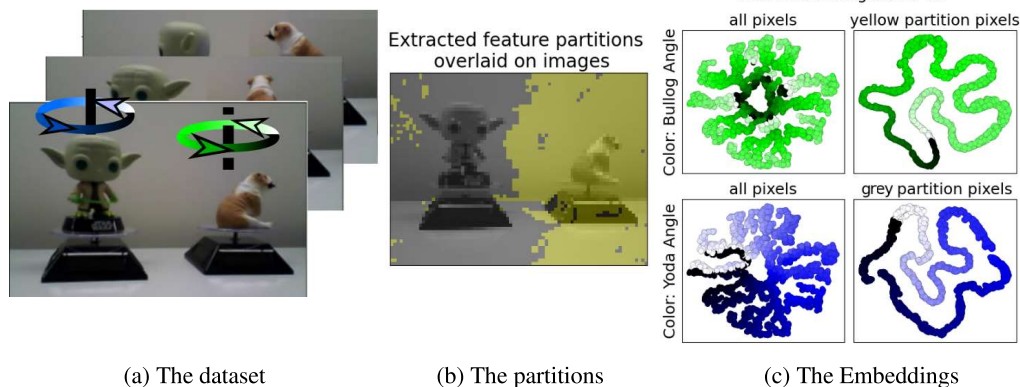

(a) The dataset        (b) The partitions        (c) The Embeddings

Figure 1: (a) The dataset consists of N=500 images with D=4800 pixels, capturing two rotating figurines with different angular velocities. (b) The two feature partitions extracted by our approach (see Algorithm 1). (c) The standard tSNE embedding based on all the pixels colored by the Yoda figurine's angle (top left) and Bulldog figurine's angle (bottom left). On the top right plot, we show the tSNE embedding of images based on the yellow extracted pixel partition (shown in 1b) colored by the Bulldog figurine's angle. On the bottom right plot, we provide the tSNE embedding based on the grey extracted pixel partition colored by the Yoda figurine's angle.

In this work, we propose a new approach to address the challenges outlined above when the observed features can be partitioned into disjoint subsets with simpler underlying sub-structures than the entire dataset. Given a prescribed number of partitions $K$, we propose to learn a partitioning of the $D$ features into $K$ mutually exclusive subsets along with $K$ corresponding similarity graphs of size $N \times N$. Each similarity graph encodes the pairwise affinities between all $N$ observations restricted to the corresponding subset of features. By creating separate similarity graphs for different sets of features, we can simplify complex data and represent it more faithfully. Then, the data in each subset of features can be embedded and visualized more effectively.

To find the best partitioning and associated similarity graphs, we propose to minimize an objective function relying on a certain graph Laplacian-based smoothness score summed over all partitions and all data points. Our approach naturally extends the graph learning step in common embedding and visualization techniques, including tSNE and Diffusion Maps, to learn multiple graphs simultaneously from adaptively chosen feature partitions. We analyze this objective function in a setup where multiple low-dimensional manifolds are embedded in a high-dimensional space. Each manifold is embbeded along with a shared manifold within a distinct subset of features, which promotes partial dependence among the different subsets. We show that in this asymptotic regime, the minimum of the objective function is attained only when the features are partitioned correctly into the subsets containing the individual embedded manifolds.

In Figure 1, we illustrate our approach on a video depicting two rotating figurines, each with a distinct angular velocity. Each frame is treated as a data point, with pixels as features. Thus, the underlying structure of this dataset is governed by the two viewing angles of the figurines. The tSNE embedding based on all the features (pixels) is difficult to interpret due to significant distortion from embedding a product of two closed curves in two dimensions. Our approach automatically identifies and partitions the pixels associated with each figurine. Consequently, the tSNE visualization based on each partition forms a simple closed curve, effectively capturing the data's structure.

Feature partitioning was considered in bi-clustering (Dhillon, 2001; Kluger et al., 2003), where the goal was to jointly cluster the features and observations simultaneously into subsets with similar entries. Our approach here, in contrast, does not require the observations in each feature partition to be clusterable, allowing for more general structures encoded by a similarity graph. Clustering techniques, such as k-means (Lloyd, 1982) and spectral clustering (Ng et al., 2001), can be adapted to cluster features by treating the features as samples. However, such methods assume that features within a cluster have proximal values, which is a challenging assumption when dealing with dependent features. For instance, such features may span different ranges and therefore will be separated. In contrast, our approach partitions the features based on similar underlying structures across the samples, allowing a more general and flexible partitioning.

Another related approach is to seek linear projections of the data into subspaces that are statistically independent or have different structures (Theis, 2006; Niu et al., 2010). These approaches decompose the data into subspaces, potentially decoupling more complicated data structures than our feature partitioning approach. However, these approaches are less interpretable in high dimensional regimes as it includes much more parameters than our partitioning. Additionally, the analytical properties of their solutions are not well understood, especially when the substructures in the data are partially dependent. A related research area is unsupervised feature selection, which focuses on identifying important data features (Solorio-Fernández et al., 2020). Methods for feature selection (He et al., 2005; Lindenbaum et al., 2021) typically utilize the similarity graph to rank the features according to a Laplacian-based smoothness score. However, when the data has multiple substructures, such methods may favor features from one substructure while overlooking others.

Finally, we mention Zhang et al. (2021); He et al. (2023), which focus on decoupling data sampled from a product of manifolds using the eigenvectors and eigenvalues of the graph Laplacian. One limitation of such approaches is that the statistical error in graph Laplacian-based quantities grows with the intrinsic dimension, which can be prohibitively large for product manifolds. In our approach, if the product manifold is decomposable across feature subsets, it's beneficial to construct the graph Laplacian separately for each subset, where the intrinsic dimension is lower.

**Notations:** Bold symbols represent vectors or matrices. The $d$-th coordinate of a vector $\boldsymbol{y}$ is denoted by $(\boldsymbol{y})_d$ or $y_d$. For any $\boldsymbol{x} \in \mathbb{R}^D$ and $\boldsymbol{\omega} \in \mathbb{R}_+^D$, denote the weighted norm by $\|\boldsymbol{x}\|_{\boldsymbol{\omega}}^2 = \sum_{d=1}^D \omega_d (\boldsymbol{x})_d^2$.

# 1 OUR APPROACH

In this section, we provide relevant background and propose a feature partitioning approach by solving a suitable optimization problem. We begin in Section 1.1 by reviewing a common graph construction step employed by traditional data embedding and visualization techniques. We demonstrate that this step can be formulated as an optimization problem minimizing a Laplacian-based score, which favors graphs with smoother characteristics. Building on this, in Section 1.2, we propose an optimization problem for partitioning the features into disjoint subsets and constructing corresponding graphs for each. These partitions are determined adaptively so that the corresponding graphs exhibit maximal smoothness according to the aforementioned Laplacian-based score.

## 1.1 TRADITIONAL GRAPH CONSTRUCTION AND THE GRAPH SCORE

Given a set of observed data points $\boldsymbol{y}_1, \ldots, \boldsymbol{y}_N \in \mathbb{R}^D$, common embedding and visualization techniques initially construct a graph representing their pairwise similarities. A popular choice of the graph affinity matrix $\boldsymbol{W} \in \mathbb{R}_+^{N \times N}$ is a row-normalized Gaussian kernel defined by

$$W_{i,j} = \begin{cases} \exp\left(-\|\boldsymbol{y}_i - \boldsymbol{y}_j\|^2/\epsilon_i\right)/D_{i,i} & i \neq j \\ 0 & i = j \end{cases} \qquad i,j = 1, \ldots, N, \qquad (1)$$

where $D_{i,i} \equiv \sum_{j=1}^N \exp(-\|\boldsymbol{y}_i - \boldsymbol{y}_j\|^2/\epsilon_i)$ for $i \in \{1, \ldots, N\}$, and $\{\epsilon_i\} \subset \mathbb{R}_+$ represent bandwidth parameters controlling the effective neighborhood size around each point. Zeroing out the main diagonal aligns with tSNE's graph construction step and also makes the resulting affinity matrix $\boldsymbol{W}$ more robust to noise (Karoui, 2010; Landa et al., 2021). Other embedding techniques, such as Diffusion Maps and Laplacian Eigenmaps, construct the graph affinity matrix similarly without zeroing out the main diagonal and use a single global bandwidth parameter.

The tSNE algorithm determines the bandwidth parameters $\epsilon_1, \ldots, \epsilon_N$ from Eq. 1 by imposing an entropy constraint on the rows of $\boldsymbol{W}$. This constraint is given by

$$-\sum_{j=1}^N W_{i,j} \log W_{i,j} = \log(\alpha), \qquad (2)$$

for $i \in \{1, \ldots, N\}$, where $\alpha$ denotes a predefined global neighborhood size parameter known as the *perplexity*, typically set between 5 and 30. The tSNE graph construction enforces this constraint by adjusting the bandwidth parameters $\{\epsilon_i\}$ adaptively to the local sampling density. In contrast, other common graph construction techniques usually employ a global bandwidth constraint of the form $\epsilon_1 = \ldots = \epsilon_N$ discussed in Lindenbaum et al. (2017); Singer et al. (2009).

Many methods for feature selection (He et al., 2005; Lindenbaum et al., 2021) utilize the affinity matrix for identifying a meaningful subset of features. In particular, given a graph affinity matrix $\tilde{\boldsymbol{W}} \in \mathbb{R}^{N \times N}$, these methods utilize a score of the form

$$S(\tilde{\boldsymbol{W}}, d) = \sum_{i,j=1}^{N} \tilde{W}_{i,j} \left( (\boldsymbol{y}_i)_d - (\boldsymbol{y}_j)_d \right)^2, \tag{3}$$

to measure the smoothness of $d$th coordinate of the data over the graph. This score is often referred to as the Laplacian score (He et al., 2005) for certain choices of the affinity matrix $\tilde{\boldsymbol{W}}$. Summing this score over all coordinates, we define the graph score $g$ as

$$J(\tilde{\boldsymbol{W}}, \{\boldsymbol{y}_1, \ldots, \boldsymbol{y}_N\}) = \sum_{i,j=1}^{N} \tilde{W}_{i,j} \|\boldsymbol{y}_i - \boldsymbol{y}_j\|^2. \tag{4}$$

The following proposition, proved in Appendix E.1, shows that the affinity matrix $\boldsymbol{W}$ defined in Eq. 1 can be interpreted as the minimizer of the graph score in Eq. 4 under constraints of perplexity and stochasticity for each row while zeroing out the main diagonal.

**Proposition 1.** *The matrix $\boldsymbol{W} \in [0,1]^{N \times N}$ defined in Eq. 1 is a solution to*

$$\operatorname*{argmin}_{\tilde{\boldsymbol{W}} \in [0,1]^{N \times N}} J(\tilde{\boldsymbol{W}}, \{\boldsymbol{y}_1, \ldots, \boldsymbol{y}_N\}), \tag{5}$$

*subject to the constraints $\tilde{W}_{i,i} = 0$, $\sum_{j=1}^{N} \tilde{W}_{i,j} = 1$ and $-\sum_{j=1}^{N} \tilde{W}_{i,j} \log \tilde{W}_{i,j} = \log(\alpha)$ for all $i \in \{1, \ldots, N\}$, where $\epsilon_1, \ldots, \epsilon_N \in \mathbb{R}_+$ from Eq. 1 satisfy the entropy constraint in Eq. 2.*

A set of works has derived closely related results, including Cuturi (2013); Van Assel et al. (2024). When the data is sampled from a Riemannian manifold and we fix the bandwidth parameters $\epsilon_1, \ldots, \epsilon_N \in \mathbb{R}_+$ (ignoring the entropy constraints), the following proposition characterizes the relation between the objective function minimized in Eq. 5 and the manifold's intrinsic dimension.

**Proposition 2.** *Let $\mathcal{M} \subset \mathbb{R}^D$ be a smooth compact Riemannian manifold with an intrinsic dimension denoted by $dim(\mathcal{M}) < D$. Assume $\boldsymbol{y}_1, \ldots, \boldsymbol{y}_N \in \mathcal{M}$ are independently sampled from a smooth non-vanishing distribution $f$ over $\mathcal{M}$, and that $\boldsymbol{W}$ is constructed as described in Eq. 1. Then, for all sufficiently small $\epsilon_1, \ldots, \epsilon_N \in \mathbb{R}_+$, we have for all $i \in \{1, \ldots, N\}$,*

$$\sum_{j=1}^{N} W_{i,j} \|\boldsymbol{y}_i - \boldsymbol{y}_j\|^2 \xrightarrow[a.s.]{N \to \infty} \frac{\epsilon_i}{2} \cdot dim(\mathcal{M}) + O(\epsilon_i^2). \tag{6}$$

The proof of Proposition 2 can be found in Appendix E.1. Hence, for the affinity matrix $\boldsymbol{W}$ from Eq. 1 with sufficiently small bandwidth parameters, the objective function in Eq. 5 approximates the intrinsic dimension of the manifold multiplied by the sum of bandwidth parameters. This quantity is smaller for manifolds with lower intrinsic dimensions, i.e., simpler manifolds governed by fewer latent variables, or when the bandwidth parameters $\{\epsilon_i\}$ are smaller.

## 1.2 Feature partitioning and multi-graph learning

We consider the problem of feature partitioning in scenarios where the features can be divided into $K$ mutually exclusive sets and accompany each set with a corresponding affinity matrix. Intuitively, the data restricted to each feature set should have a smoother and simpler structure than the data across all features combined. To find the feature partitions and their associated graphs, we propose to minimize the sum of graph scores in Eq. 4 over all partitions.

Let $\{\tilde{\boldsymbol{\omega}}^{(1)}, \ldots, \tilde{\boldsymbol{\omega}}^{(K)}\} \in \{0,1\}^D$ be a feasible feature partitioning, where $\tilde{\boldsymbol{\omega}}_d^{(k)} = 1$ if the $d$th feature is used within the $k$th partition, and $\tilde{\boldsymbol{\omega}}_d^{(k)} = 0$ otherwise. The feature partitions cover all features and are mutually exclusive; namely, they satisfy $\sum_{k=1}^{K} \tilde{\omega}_d^{(k)} = 1$ for all $d \in \{1, \ldots, D\}$. The affinity matrices corresponding to the $K$ partitions are denoted by $\tilde{\boldsymbol{W}}^{(1)}, \ldots, \tilde{\boldsymbol{W}}^{(K)} \subset \mathbb{R}_+^{N \times N}$. We now extend the graph score defined in Eq. 4 to support multiple feature partitions as

$$G(\{\tilde{\boldsymbol{W}}^{(k)}\}_{k=1}^{K}, \{\tilde{\boldsymbol{\omega}}^{(k)}\}_{k=1}^{K}, \{\boldsymbol{y}_1, \ldots, \boldsymbol{y}_N\}) = \sum_{k=1}^{K} \sum_{i,j=1}^{N} \tilde{W}_{i,j}^{(k)} \|\boldsymbol{y}_i - \boldsymbol{y}_j\|_{\tilde{\boldsymbol{\omega}}^{(k)}}^2, \tag{7}$$

recalling that $\|\boldsymbol{v}\|_{\tilde{\boldsymbol{\omega}}^{(k)}}^2 = \sum_{d=1}^{D} \tilde{\omega}_d^{(k)} v_d^2$ for any $\boldsymbol{v} \in \mathbb{R}^D$. Notably, when all the affinity matrices are the same ($\tilde{\boldsymbol{W}}^{(1)} = \ldots = \tilde{\boldsymbol{W}}^{(K)} = \tilde{\boldsymbol{W}}$), then this score coincides with the score defined in Eq. 4, i.e., $G(\{\tilde{\boldsymbol{W}}^{(k)}\}_{k=1}^K, \{\tilde{\boldsymbol{\omega}}^{(k)}\}_{k=1}^K, \{\boldsymbol{y}_1, \ldots, \boldsymbol{y}_N\}) = J(\tilde{\boldsymbol{W}}, \{\boldsymbol{y}_1, \ldots, \boldsymbol{y}_N\})$.

To find the feature partitioning and the corresponding affinity matrices, we propose to solve the following optimization problem.

**Problem 1.**

$$\min_{\left\{\tilde{\boldsymbol{W}}^{(k)}\right\}_k, \left\{\tilde{\boldsymbol{\omega}}^{(k)}\right\}_k} G(\{\tilde{\boldsymbol{W}}^{(k)}\}_{k=1}^K, \{\tilde{\boldsymbol{\omega}}^{(k)}\}_{k=1}^K, \{\boldsymbol{y}_1, \ldots, \boldsymbol{y}_N\}), \tag{8}$$

*under the constraints* $\sum_{k=1}^K \tilde{\omega}_d^{(k)} = 1$, $\sum_{j=1}^N \tilde{W}_{i,j}^{(k)} = 1$, $\tilde{W}_{i,i}^{(k)} = 0$, *and* $-\sum_{j=1}^N \tilde{W}_{i,j}^{(k)} \log\left(\tilde{W}_{i,j}^{(k)}\right) = \log(\alpha)$, *for* $i = 1, \ldots, N$, $k = 1, \ldots, K$, *and* $d = 1, \ldots, D$, *where* $\alpha$ *is a perplexity parameter.*

In the next proposition, proved in Appendix E.1, we characterize the solutions to this problem.

**Proposition 3.** *An optimal partitioning solution* $\{\boldsymbol{\omega}^{(k)}\}_{k=1}^K$ *and the corresponding affinity matrices* $\{\boldsymbol{W}^{(k)}\}_{k=1}^K$ *that solve Problem 1 must satisfy*

$$\omega_d^{(k)} = \begin{cases} 1 & \text{if } k = \tilde{k} \text{ for some } \tilde{k} \in \Omega(d) \\ 0 & \text{else} \end{cases}, \tag{9}$$

$$W_{i,j}^{(k)} = \begin{cases} \exp\left(-\|\boldsymbol{y}_i - \boldsymbol{y}_j\|_{\boldsymbol{\omega}^{(k)}}^2 / \epsilon_{k,i}\right) / D_{i,i}^{(k)} & \text{if } i \neq j \\ 0 & \text{else} \end{cases}, \tag{10}$$

$$\Omega(d) = \underset{k \in \{1, \ldots, K\}}{\arg\min} \, S(\boldsymbol{W}^{(k)}, d), \tag{11}$$

*for* $d = 1, \ldots, D$ *and* $i, j = 1, \ldots, N$, *where* $D_{i,i}^{(k)} \equiv \sum_{j=1}^N \exp(-\|\boldsymbol{y}_i - \boldsymbol{y}_j\|_{\boldsymbol{\omega}^{(k)}}^2 / \epsilon_{k,i})$, *the bandwidth parameters* $\{\epsilon_{k,i}\}$ *are set to satisfy the entropy constraints in Problem 1, and S is the Laplacian-type score defined in Eq. 3.*

Therefore, the $d$th feature of the data is assigned to the $k$th partition if it is smoother with respect to the affinity matrix $W^{(k)}$ than the other affinity matrices, as measured by the Laplacian-type score in the right-hand side of Eq. 11. Alternatively, the affinity matrix $\boldsymbol{W}^{(k)}$ is simply a row-normalized Gaussian kernel constructed from the features in the $k$th partition, analogously to the traditional construction in Eq. 1. Therefore, our approach naturally extends the traditional graph construction techniques discussed in Section 1.1 by forming multiple graphs from disjoint feature partitions, which are optimized to minimize the total smoothness of the features across their associated graphs.

To further motivate Problem 1, consider data formed by concatenating $K$ feature groups, where the features in each group were sampled independently from a different Riemannian manifold. For the correct partitioning and the optimal affinity matrices derived in Eq. 10, the graph score in Eq. 7 converges to a weighted sum of the manifolds' intrinsic dimensions (see Proposition 2). The weights depends on the corresponding bandwidth parameters $\{\epsilon_{k,i}\}$, which are set to enforce the negative entropy constraints. Intuitively, for incorrect partitioning, the data points restricted to each feature partition would reside on a product of manifolds with a higher intrinsic dimension, and consequently, the required bandwidth parameters that satisfy the entropy constraints would be larger. Hence, we can interpret the optimization problem as dividing the feature space into partitions whose maximal intrinsic dimension is as small as possible.

In Appendix C we present an algorithm for approximately solving Problem 1; see Algorithm 1. We justify this procedure analytically and empirically and discuss its computational complexity. In Section 3 we evaluate the algorithm's performance on various datasets. The software along with the pseudo-code are given in https://anonymous.4open.science/r/FP-70F4/.

## 2 ANALYSIS

In this section, we examine a version of the feature partitioning problem (see Problem 1) using a data generative model where there are $K$ partially dependent subsets of features. We analyze the

objective function landscape in a high-dimensional asymptotic regime and establish that the optimal partitioning minimizes it. Additionally, we provide numerical evidence in Figure 2 showing that its landscape closely resembles the objective landscape of the original problem.

We begin by defining a variant of the graph score as

$$\tilde{G}\left(\epsilon, \{\tilde{\boldsymbol{W}}^{(k)}\}_{k=1}^{K}, \{\tilde{\boldsymbol{\omega}}^{(k)}\}_{k=1}^{K}, \{\boldsymbol{y}_1, \ldots, \boldsymbol{y}_N\}\right) \quad = \quad \sum_{k=1}^{K}\sum_{i,j=1}^{N} \tilde{W}_{i,j}^{(k)} \cdot \frac{\|\boldsymbol{y}_i - \boldsymbol{y}_j\|_{\tilde{\boldsymbol{\omega}}^{(k)}}^2}{(1/D)\sum_d \tilde{\omega}_d^{(k)}}. \quad (12)$$

This variant adjusts the Laplacian-based score by normalizing the portion related to each partition based on the average number of features used within that partition. This adjustment will account for the simplification of the optimization problem that we define next.

Building on this score, we propose a simplified version of the original feature partitioning problem (Problem 1) to identify the feature partitions and their corresponding affinity matrices. This formulation utilizes a regularized minimization approach that incorporates the negative entropy constraint into the objective, making the problem easier to analyze.

**Problem 2.** *Consider the optimization problem defined by*

$$\min_{\left\{\tilde{\boldsymbol{W}}^{(k)}\right\}_k, \left\{\tilde{\boldsymbol{\omega}}^{(k)}\right\}_k} \tilde{G}\left(\{\tilde{\boldsymbol{W}}^{(k)}\}_{k=1}^{K}, \{\tilde{\boldsymbol{\omega}}\}_{k=1}^{K}, \{\boldsymbol{y}_1, \ldots, \boldsymbol{y}_N\}\right) + \epsilon \sum_{k=1}^{K}\sum_{i,j=1}^{N} \tilde{W}_{i,j}^{(k)} \log\left(\tilde{W}_{i,j}^{(k)}\right) \quad (13)$$

*with the following constraints* $\sum_{k=1}^{K} \tilde{\omega}_d^{(k)} = 1$, $\sum_{j=1}^{N} \tilde{W}_{i,j}^{(k)} = 1$, $\tilde{W}_{i,i}^{(k)} = 0$, *for* $i = 1, \ldots, N$, $k = 1, \ldots, K$ *and* $d = 1, \ldots, D$.

We characterize the affinity matrices that minimize this objective in the following corollary.

**Corollary 1.** *Let* $\{\tilde{\boldsymbol{\omega}}^{(k)}\}$ *be a partitioning that satisfies the constraints in Problem 2. Then, the graph affinity matrices that minimize Eq. 13 while fixing the partitioning parameters* $\{\tilde{\boldsymbol{\omega}}^{(k)}\}$ *are*

$$W_{i,j}^{(k)} = \begin{cases} \exp\left(-\frac{\|\boldsymbol{y}_i - \boldsymbol{y}_j\|_{\tilde{\boldsymbol{\omega}}^{(k)}}^2}{\epsilon \cdot (1/D)\sum_{d=1}^{D}\tilde{\omega}_d^{(k)}}\right) / \sum_{t=1}^{N}\exp\left(-\frac{\|\boldsymbol{y}_i - \boldsymbol{y}_t\|_{\tilde{\boldsymbol{\omega}}^{(k)}}^2}{\epsilon \cdot (1/D)\sum_{d=1}^{D}\tilde{\omega}_d^{(k)}}\right) & if\ i \neq j \\ 0 & else \end{cases}. \quad (14)$$

Its proof can be found in Appendix E.2. Note that by using $\tilde{G}$, the bandwidth parameters of the affinity matrices adapt according to the number of features in each of their respective partitions.

Next, we derive the asymptotic value of the objective under a high-dimensional regime. We consider a generative data model in which the observed space is based on subsets of features that exhibit partial dependence. Let $\mathcal{M}_1 \subset \mathbb{R}^{d_1} \ldots, \mathcal{M}_{K+1} \subset \mathbb{R}^{d_{K+1}}$ be latent smooth compact Riemannian manifolds with corresponding smooth, non-vanishing densities $f_1, \ldots, f_{K+1}$. The latent samples $\{\boldsymbol{x}_i^{(s)}\}_{i=1}^{N} \in \mathcal{M}_s$ are independently sampled according to $f_s$ for $s = 1, \ldots, K+1$ and $i = 1, \ldots, N$. The observed data points, denoted by $\{\boldsymbol{y}_i\}_{i=1}^{N} \in \mathbb{R}^D$, are constructed by

$$\boldsymbol{y}_i^T = \left((\boldsymbol{y}_i^{(1)})^T, \ldots (\boldsymbol{y}_i^{(K)})^T\right) \in \mathbb{R}^D, \quad \boldsymbol{y}_i^{(s)} = \boldsymbol{P}^{(s)} \begin{bmatrix} \boldsymbol{x}_i^{(s)} \\ \boldsymbol{x}_i^{(K+1)} \end{bmatrix} \in \mathbb{R}^{D_s} \qquad s = 1, \ldots, K, \quad (15)$$

where $D = \sum_{s=1}^{K} D_s$, and the entries of each matrix in $\boldsymbol{P}^{(s)} \in \mathbb{R}^{D_k \times (d_k + d_{K+1})}$ are independently sampled from $\mathcal{N}(0, 1/D_s)$ for $s = 1, \ldots, K$.

In the next theorem, whose proof can be found in Appendix E.2, we examine the objective of Problem 2 with an added constant term in a high dimensional asymptotic regime where $D, N \to \infty$ and $D/N \to \infty$. We assume that the relative size of each feature subset satisfies $D_s/D \to \beta_s \in (0, 1)$ for any $s \in \{1, \ldots, K\}$. For simplicity, we denote each partition as the concatenation partitions over the $K$ subsets by $\boldsymbol{\omega}^{(k)} = (\boldsymbol{\omega}^{(k,1)}, \ldots, \boldsymbol{\omega}^{(k,K)})$ and its relative magnitude with respect to each part by $p_s^{(k)} = \sum_{d=1}^{D_s} \omega_d^{(k,s)}/D_s$, for any $k, s \in \{1 \ldots, K\}$.

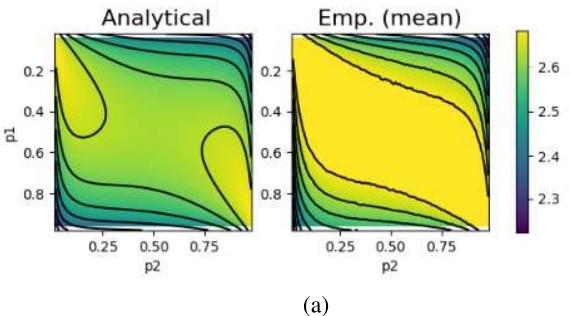 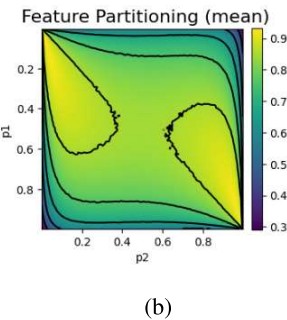

(a)                                                                                      (b)

Figure 2: An experiment based on Section 2 for $K = 2$. (a) The analytical loss landscape of Problem 2 presented in Eq. 17 (Left) and the mean empirical loss as defined in Eq. 16 (Right). For the empirical case, the $p_1, p_2 \in (0, 1)$ indicate the proportion of features out of the two feature subsets used by $\boldsymbol{\omega}^{(1)}$, while $\boldsymbol{\omega}^{(2)}$ taking the remainder. The color represents the mean value over 100 simulations, based on the affinity matrices defined in Corollary 1. (b) We consider a similar mean empirical loss based on Problem 1 ($min_{\{\boldsymbol{W}^{(k)}\}} g(\{\boldsymbol{W}^{(k)}\}, \{\boldsymbol{\omega}^{(k)}\}, \{\boldsymbol{y}_i\}_{i=1}^N)$), where the affinity matrices are as defined in Eq. 10. Additional details can be found at the end of Section 2.

**Theorem 1.** *Let $\{\boldsymbol{\omega}^{(k)}\}$ be a partitioning that satisfies the constraints in Problem 2. Then, there exists $\tilde{\epsilon}(\mathcal{M}, f) \le 1$ such that for any $\epsilon \le \tilde{\epsilon}$ and $p_s^{(k)} \in [\sqrt{\epsilon}, 1 - (K-1)\sqrt{\epsilon}]$ we have*

$$\min_{\{\tilde{\boldsymbol{W}}^{(k)}\}} \frac{1}{\epsilon N} \left( \tilde{G}\left( \left\{ \tilde{\boldsymbol{W}}^{(k)} \right\}, \left\{ \boldsymbol{\omega}^{(k)} \right\}, \{\boldsymbol{y}_i\} \right) + \epsilon \left( \sum_{i,j=1}^N \tilde{W}_{i,j}^{(k)} \log \left( \tilde{W}_{i,j}^{(k)} \right) \right) \right) + K \log(N) \quad (16)$$

$$\overset{N,D \to \infty}{\longrightarrow} \sum_{k=1}^K \frac{dim(\mathcal{M}_{K+1})}{2} \log \left( \frac{\sum_{s=1}^K p_s^{(k)}}{\sum_{t=1}^K p_t^{(k)} \beta_t} \right) + \sum_{k,s=1}^K \frac{dim(\mathcal{M}_s)}{2} \log \left( \frac{p_s^{(k)}}{\sum_{t=1}^K p_t^{(k)} \beta_k} \right) \quad (17)$$

$$-K \sum_{s=1}^{K+1} \frac{dim(\mathcal{M}_s)}{2} \log \left( \pi \epsilon \right) + K \sum_{s=1}^{K+1} h_s(f_s) + O(\sqrt{\epsilon}),$$

*where the convergence is in probability and $h_k(f_k) = -\int_{\boldsymbol{z} \in \mathcal{M}_k} f_k(\boldsymbol{z}) \log f_k(\boldsymbol{z}) d\boldsymbol{z}$ is the differential entropy of the density $f_k$ over $\mathcal{M}_k$.*

Evidently, only the first two terms in Eq. 17 are influenced by the feature partitions, while the rest are mainly impacted by $\epsilon$, the manifolds properties and their densities. In the next theorem, we show that for two partitions, $K = 2$, its minimizer must separate the data features accurately as $\epsilon \to 0$.

**Theorem 2.** *Define $f : (0, 1)^2 \to \mathbb{R}$ by*

$$f(p_1, p_2) = \sum_{k=1}^K \frac{dim(\mathcal{M}_{K+1})}{2} \log \left( \frac{\sum_{s=1}^K p_s^{(k)}}{\sum_{t=1}^K p_t^{(k)} \beta_t} \right) + \sum_{k,s=1}^K \frac{dim(\mathcal{M}_s)}{2} \log \left( \frac{p_s^{(k)}}{\sum_{t=1}^K p_t^{(k)} \beta_k} \right), \quad (18)$$

*where $p_1^{(1)} = p_1$ and $p_2^{(1)} = p_2$ and therefore $p_2^{(1)} = 1 - p_1^{(1)}$, $p_2^{(2)} = 1 - p_2^{(2)}$. The minimizers $p_1^*, p_2^* = \lim_{\epsilon \to 0} \arg\min_{p_1, p_2 \in [\sqrt{\epsilon}, 1 - \sqrt{\epsilon}]^2} f(p_1, p_2)$ exist and satisfy $(p_1^*, p_2^*) \in \{(0, 1), (1, 0)\}$.*

The proof of Theorem 2 is in Appendix E.2. To conclude, we proposed a variant of the feature partitioning problem in Problem 2 and analyzed its loss landscape in an asymptotic regime. We considered a data-generating process where the features are composed of $K$ partially dependent subsets of features, making the partitioning task nontrivial. Finally, we showed that in this nontrivial case when $K = 2$ the loss is minimized when the partitioning solution aligns with the ground truth.

In Figure 2a, we compare the empirical landscape of the loss function in Eq. 16 with its analytical counterpart derived in Eq. 17 for $K = 2$. The dataset consists of $N = 1000$ samples drawn uniformly from three latent manifolds, $\mathcal{M}_1, \mathcal{M}_2, \mathcal{M}_3 \subset \mathbb{R}^2$, each forming a unit circle, with $\mathcal{M}_3$ acting as the shared manifold. The observed feature dimensions is $D_1 = 2500$ and $D_2 = 7500$. As expected from Theorem 1, the minimal values are found near the ground truth partitioning solutions

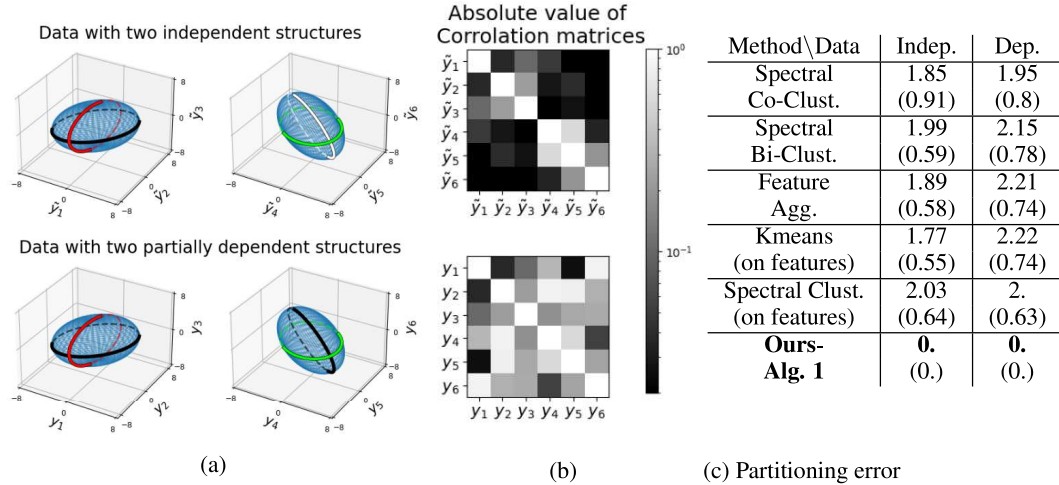

| Method\Data | Indep. | Dep. |
|---|---|---|
| Spectral Co-Clust. | 1.85 (0.91) | 1.95 (0.8) |
| Spectral Bi-Clust. | 1.99 (0.59) | 2.15 (0.78) |
| Feature Agg. | 1.89 (0.58) | 2.21 (0.74) |
| Kmeans (on features) | 1.77 (0.55) | 2.22 (0.74) |
| Spectral Clust. (on features) | 2.03 (0.64) | 2. (0.63) |
| **Ours-Alg. 1** | **0.** (0.) | **0.** (0.) |

(a)          (b)          (c) Partitioning error

Figure 3: Illustrative example from Section 3.1. (a) Two datasets in $\mathbb{R}^6$, each consists of points lying on two concatenated 2-dimensional ellipsoids. These ellipsoids are parameterized by polar coordinates, each depicted by a different colored ellipse. In the first dataset the ellipsoids are independent (Top), while in the second they are partially dependent (Bottom) since one polar coordinate is shared between the two ellipsoids (the black ellipse). (b) The correlation matrix of each dataset. (c) The performance of different methods in partitioning the six features. The partitioning error, detailed in Appendix D, measures the $L1$ distance between the estimated and true partitions up to a permutation of the ordering. The table shows the mean error across 100 trials with the standard deviation in parentheses. We note that the true partitions separates the first three coordinates from the last three.

($p_1 = 1, p_2 = 0$ or vice versa). In Figure 2b, we provide the loss landscape of the original loss of Problem 1 under the same data configuration. Evidently it resembles the loss landscape of its simplified version considered throughout this section.

## 3 EXPERIMENTS

This section demonstrates our approach and its advantages using synthetic and real data. In Section 3.1, we illustrate its effectiveness in independent and partially dependent feature partitioning problems on simulated data. In Section 3.2, we consider data consisting of pairs of images captured by two different cameras of rotating objects. By applying our method over the image pairs, the partitions reveal regions from both cameras that are either common, indicating a shared intrinsic variable, or camera-specific. Finally, Section 3.3 demonstrates the effectiveness of our approach in uncovering latent processes of interest within genomic data from single-cell RNA sequencing.

### 3.1 PRODUCT OF 2-DIMENSIONAL ELLIPSOIDS

We consider two datasets in $\mathbb{R}^6$: one with two independent features subsets and another with partially dependent subsets, and the task is to retrieve these subsets ($K = 2$). In the independent case, samples are drawn from the concatenation of two rotated $2D$ ellipsoids. In the partially dependent case, one the polar coordinate is shared between the ellipsoids. See Appendix D.1 for further details.

This experiment considers several algorithms tailored for the partition task- Spectral Co-Clustering (Dhillon, 2001), Bi-Clustering (Kluger et al., 2003), Feature Agglomeration (Kaufman & Rousseeuw, 2009), K-means with $K = 2$, and Spectral Clustering with $K = 2$ and with 2 embedding dimensions (Von Luxburg, 2007). We evaluated Spectral Co-Clustering and Bi-Clustering using their feature clustering. For Feature Agglomeration, we used the first level of the hierarchical tree for clustering. Additionally, we applied K-Means (Lloyd, 1982) and Spectral Clustering (Ng et al., 2001) on the features, treating samples as coordinates (i.e., applied on the transposed data) .

In Figure 3a, the independent (Top) and partially dependent (Bottom) scenarios are visualized. In Figure 3b, we provide the correlation matrices related to each scenario to get a grasp on these problems. Specifically, while in the independent case the correlation matrix has a block diagonal

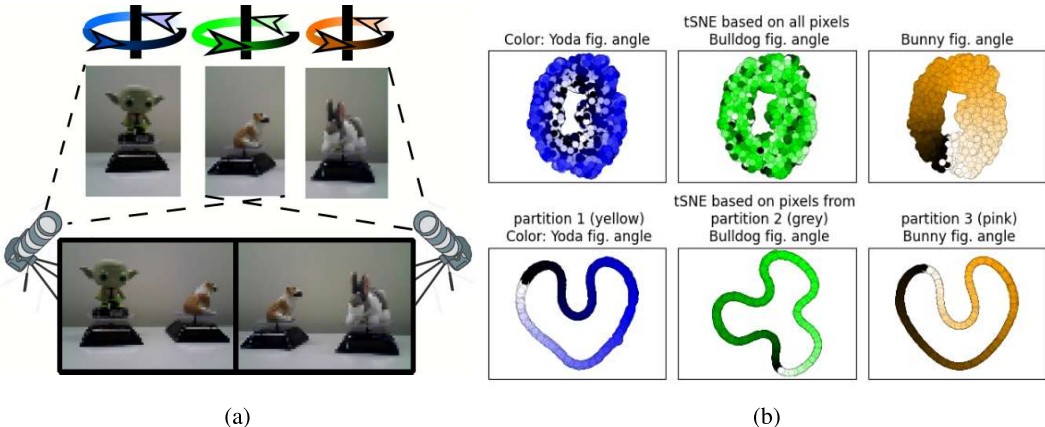

(a)                                                              (b)

Figure 4: The experiment detailed in Section 3.2. The dataset includes $N = 5000$ greyscale images with $D = 9600$ pixels from a video but without their temporal order. (a) Illustration of the observation process of the dataset with images captured simultaneously by two different cameras. Each camera captures two of the three figurines that are rotating in different angular speeds. (b) The plots include the tSNE embedding based on all pixels colored by the angle of the Yoda (top left), Bulldog (top middle) and Bunny (top right) figurines. The bottom row of the plots shows the tSNE embeddings based on the first (bottom left), second (bottom middle) and third extracted partitions (bottom right), with data-points colored by the angles of the Yoda, Bulldog and Bunny figurine, respectively.

structure separating the first three coordinates from the latter ones, in the partially dependent case it is not the case. In Figure 3c, we assess the performance of the mentioned algorithms in both scenarios, where the error calculates the $L1$ distance between the true partition and the estimated one up to the reordering of the partitions. Our algorithm clearly outperforms the other methods.

## 3.2 TWO CONCATENATED VIEWS OF ROTATING FIGURINES

The experiment considers a more challenging setting compared to the one showed in Figure 1, where two stationary cameras captures three rotating figurines with distinct angular speeds synchronously. Both cameras record the same bulldog figurine and an extra unique figurine, as depicted in Figure 4a. Each sample is a concatenated pair of images captured by the two cameras, with the underlying latent variables being the figurines' angles. The task is to separate the pixels of the concatenated images from all videos into $K = 3$ partitions, each governed by a different figurine's movement, without using any knowledge of the temporal order in which the images were captured.

Extracting a pixel partition that identifies the common Bulldog figurine in the two different angles within the concatenated image relates to matching features across different measurement devices. Usually, spectral algorithms are employed to extract the underlying common variables based on the two graphs, each is based on samples from a different source (e.g., images from different cameras) (Lederman et al., 2015; Lindenbaum et al., 2015). When the features can be divided according to their governing variables, even with partially correspondence as discussed before, our algorithm can effectively extract both the common and specific variables from each set of measurements.

We apply our method on the concatenated pairs of images and extract $K = 3$ pixel partitions. In Figure 4b we compare the tSNE embedding based on all the pixels compared to the embedding based on each pixel partition. As illustrated, the embedding based on each partition successfully captures the rotational structure originating in its corresponding single figurine. Additionally, we evaluate the quality of our extracted pixel partitions alongside those generated by K-means and spectral clustering, which are applied to the pixel space (treating the samples as coordinates).

Finally, we evaluate the quality of our extracted pixel partitions alongside those generated by K-means and spectral clustering, which are applied to the pixels while treating the samples as coordinates (i.e., applied on the transposed data). We examined the overlap between the 50-nearest neighbor based on the extracted feature partitions with the neighbors determined by the figurine's true angles. The percentage of the overlap is as follows: Ours at $85.1\%$, K-means at $25.9\%$, spectral

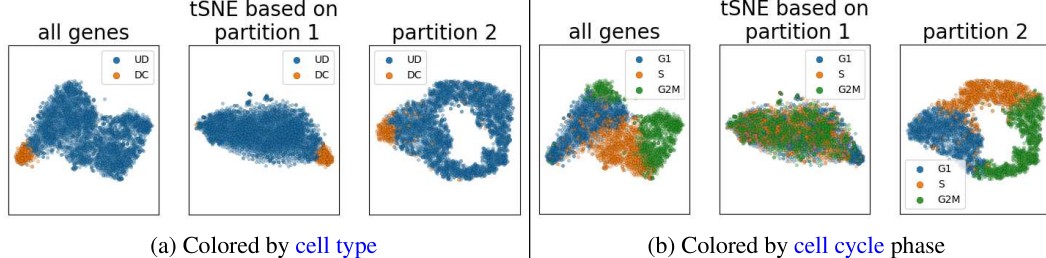

(a) Colored by cell type          (b) Colored by cell cycle phase

Figure 5: The tSNE embedding of $N = 3745$ cells characterized by their $D = 1708$ genes values as detailed in Section 3.3 colored by different attributes of the cells.

clustering at 24.7%, and using all features at 11.74%. Clearly, our method outperforms the others. Additional details of the metric, along with figures and evaluations is given in Appendix D.2.

### 3.3 SCRNA-SEQ DATA WITH CYCLIC CELL STRUCTURE

In this experiment, we demonstrate the advantages of our approach for separating co-occurring cellular processes in single-cell RNA-sequencing (scRNA-seq) data, which also enable a clear visualization of each individual process. Specifically, the data contains dermal cells from mouse skin embryos (Qu et al., 2024). These cells exhibit a cyclic phase structure with three distinct phases (G1, G2, and S) between which they transition in a repeating cycle, while simultaneously undergoing a development from an undifferentiated (UD) state to a dermal condensate (DC) cell type. Each of these processes is associated with different subsets of genes of which many are known Tirosh et al. (2016), thus confirming the relevance of our approach for this data.

The dataset comprises $N = 3745$ cells and $D = 1708$ features, each representing the number of genes sequenced within a cell after standard processing is applied; see Appendix D for details. The authors of Qu et al. (2024) examined the genes through a gene similarity graph derived from the cells' affinity matrix and their gene profiles. In contrast, our approach partitions the genes according to their graph structure, generating a separate graph for each subset.

Using our method, we divide the genes into $K = 2$ partitions and compare in Figure 5 the tSNE embedding based on all features with the embeddings based on each partition. The embedding based on all features shows the cell type structure (left) but not the cell cycle phase structure (right). In contrast, the partition-based embeddings reveal both structures: Partition 2 highlights the cell type structure and is largely agnostic to the cell cycle, while Partition 1 reveals the cell cycle structure. In Appendix D.4, we provide detailed visualizations and extract informative genes from each partition using an unsupervised feature selection method. Additionally, in figure 9, we show the Diffusion Maps embeddings, and in Figure 10, we further explore the tSNE embeddings.

Overall, our approach separates the genes according to the two underlying processes governing them. These processes are partially dependent, as shown in the embedding based on partition 2, where the DC-type cells are situated among the G1 cell cycle phase cells.

## 4 DISCUSSION

This paper presents a novel approach to improve the quality and interpretability of visualization and embedding techniques. By elevating a common graph construction step, our approach divides the features into partitions with smoother structures across the samples, and then generates an embedding for the data-points based on each feature partition. We demonstrate its effectiveness both analytically and empirically using simulated and real-world data, even when the features consist of partially dependent subsets. In Appendix G, we discuss practical considerations of our approach, e.g., the selection of the number of feature partitions.

We identify several potential future directions. First, it is desirable to derive automatic procedures to determine the number of partitions. Second, exploring more effective optimization techniques for the feature partition problem (Problem 1) or deriving a convex version of it would be valuable. Finally, extending the analysis to the feature partitioning problem could offer further insights.

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
