## A  CODE REPOSITORY

The code for this paper is given in `https://anonymous.4open.science/r/FP-70F4/`

## B  THE ALGORITHM

---
**Algorithm 1** Multi- Generate multiple views of the data

---
**Input:** $\boldsymbol{y}_1, \ldots, \boldsymbol{y}_N \in \mathbb{R}^D$ - data samples, $K \in \mathbb{N}$ - Amount of partitions, T- amount iterations
**Output:** $\boldsymbol{\omega}^{(1)}, \ldots, \boldsymbol{\omega}^{(K)} \in [0,1]^D$ - Partition of the features into the $K$ sets
 1:  Init feature partition vectors $\boldsymbol{\omega}^{(1)}, \ldots, \boldsymbol{\omega}^{(K)}$ by

$$\omega_d^{(k)} = \tilde{\omega}_d^{(k)} / \sum_{s=1}^{K} \tilde{\omega}_d^{(s)}, \qquad k = 1, \ldots, K, \ d = 1, \ldots, D, \qquad (19)$$

   and $\tilde{\boldsymbol{\omega}}^{(1)}, \ldots, \tilde{\boldsymbol{\omega}}^{(K)} \overset{i.i.d}{\sim} Uniform[0,1]^D$
 2:  Set $\delta \leftarrow \delta_{init}$ according to Eq. 26
 3:  **for** t=1,..,T **do**
 4:      If $t = T$: $\delta = 0$
 5:      **while**  Score decreases  **do**
 6:          Update the Graph matrices - $\boldsymbol{W}^{(1)}, \ldots, \boldsymbol{W}^{(K)} \in [0,1]^{N \times N}$ according to Eq. 10
 7:          Update the feature partition vectors- $\boldsymbol{\omega}^{(1)}, \ldots, \boldsymbol{\omega}^{(K)} \in [0,1]^D$ according to Eq. 23
 8:          Calculate the score according to Eq. 22
 9:          $\delta \leftarrow \delta/2$
10:  Return $\boldsymbol{\omega}^{(1)}, \ldots, \boldsymbol{\omega}^{(K)}$.
**Note:** In our simulations, we run this algorithm multiple times with different random initializations and return the solution with the smallest score.

---

## C  ALGORITHMIC DETAILS

This section presents an algorithm for approximately solving the feature partitioning problem described in Problem 1. The algorithm solves approximately a regularized version of this problem at each iteration, gradually reducing the regularization coefficient until the problem aligns with the original problem. Throughout this section, we derive its update rules, compare it to a naive solution to Problem 1 and derive its complexity. The proofs for this section are given in Appendix E.3.

Based on Proposition 3, one can derive an alternating optimization directly for Problem 1. However, this method is susceptible to converge to local minima. Specifically, an update rule for the partitioning weights that follows Eq. 9 yields solutions situated on the boundary of the partitioning domain. These solutions, potentially local minima, may lead to rapid convergence regardless of the initialization. In the right plot of Figure 6 we provide an example.

Now, we begin by defining a regularized version of the algorithm based on $\delta \in \mathbb{R}_+$ by

$$G_{reg}(\delta, \{\tilde{\boldsymbol{W}}^{(k)}\}_{k=1}^{K}, \{\tilde{\boldsymbol{\omega}}^{(k)}\}_{k=1}^{K}, \{\boldsymbol{y}_i\}_{i=1}^{N}) \quad = \quad G(\{\tilde{\boldsymbol{W}}^{(k)}\}, \{\tilde{\boldsymbol{\omega}}^{(k)}\}, \{\boldsymbol{y}_i\}_{i=1}^{N}) + \qquad (20)$$

$$\delta \left( D \log(K) + \sum_{d=1}^{D} \sum_{k=1}^{K} \tilde{\omega}_d^{(k)} \log(\tilde{\omega}_d^{(k)}) \right) \quad (21)$$

The regularization term is non-negative because $\sum_{k=1}^{K} \omega_d^{(k)} = 1$ for all $d \in \{1, \ldots, D\}$ and the fact that negative entropy achieves its lower bound when the probability distribution is uniform ($\omega_d^{(k)} = 1/K$ for all $d \in \{1, \ldots, D\}$ and $k \in \{1, \ldots, K\}$).

Now we can rewrite the new optimization problem by

Figure 6: An experiment comparing the dynamics of our algorithm updates Algorithm 1 (left), with an alternating minimization technique based on Eq. 3 (right). Specifically on the left, each column shows the loss landscape around the partitioning solution given in each iteration $t \in \{1, \ldots, 3\}$, where the partitioning updates in steps $5 - 8$ are drawn on top of that. Specifically, in the first row, the loss landscape concerns the first two partition coordinates, while the last two are constrained to be the initial solution at this step. The bottom row is the but regarding the third and fourth coordinate, while constraining the first two. The black arrows depict the partitioning parameter updates applied throughout each algorithm iteration. The data is composed of $N = 400$ samples in $D = 4$ dimensions that consist of two concatenated circles, meaning that the correct solution is $\boldsymbol{\omega}^{(1)} = (0, 0, 1, 1)$. In the regularized case (Left) after the algorithm converges to a specific configuration $t$ its updated parameter values are assigned to $\boldsymbol{\omega}^{\{t+1\}}$, as described in Algorithm 1.

**Problem 3.** *Let $\delta \in \mathbb{R}_+$. Consider the optimization problem defined as follows*

$$\min_{\left\{\tilde{\boldsymbol{W}}^{(k)}\right\}_{k=1}^{K}, \left\{\tilde{\boldsymbol{\omega}}^{(k)}\right\}_{k=1}^{K}} G_{reg}(\delta, \{\tilde{\boldsymbol{W}}^{(k)}\}_{k=1}^{K}, \{\tilde{\boldsymbol{\omega}}^{(k)}\}_{k=1}^{K}, \{\boldsymbol{y}_i\}_{i=1}^{N}), \tag{22}$$

*with the following constraints: $\sum_{k=1}^{K} \tilde{\omega}_d^{(k)} = 1$, $\sum_{j=1}^{N} \tilde{W}_{i,j}^{(k)} = 1$, $\tilde{W}_{i,i}^{(k)} = 0$ for $i = 1, \ldots, N$, $k = 1, \ldots, K$, and $d = 1, \ldots, D$.*

Our algorithm approximately solves Problem 3 for some $\delta = \delta_{init}$, and then will gradually decrease it until the problem aligns with Problem 1. We begin by focusing on the solution of the problem for each $\delta$. Given that there are two sets of parameters, the feature partitions $\{\boldsymbol{\omega}^{(k)}\}$ and their corresponding affinity matrices $\{\boldsymbol{W}^{(k)}\}$, we propose an alternating minimization approach. Meaning, the method alternately minimizes the objective function, focusing first on the affinity matrices while keeping the partition weights fixed and vice versa. These parameters are derived below-

**Proposition 4.** *Let $\delta \geq 0$, $\{\boldsymbol{\omega}^{(k)}\}_{k=1}^{K} \subset [0, 1]^D$ be a partitioning weights and $\{\boldsymbol{W}^{(k)}\}_{k=1}^{K}, \subset [0, 1]^{N \times N}$ be affinity matrices that satisfy the constraints of Problem 3.*

*Define $\{\boldsymbol{W}^{(k)*}\} \subset [0, 1]^{N \times N}$ as in Eq. 10 and $\{\boldsymbol{\omega}^{(k)*}\}_{k=1}^{K}$ by*

$$\omega_d^{(k)*} = \exp\left(-\frac{\sum_{i,j} W_{i,j}^{(k)} \left((\boldsymbol{y}_i)_d - (\boldsymbol{y}_j)_d\right)^2}{\delta}\right) \Big/ \sum_{s=1}^{K} \exp\left(-\frac{\sum_{i,j} W_{i,j}^{(s)} \left((\boldsymbol{y}_i)_d - (\boldsymbol{y}_j)_d\right)^2}{\delta}\right). \tag{23}$$

*Then, a set of parameters that minimize the objective function and are with the domain of Problem 3 are*

$$\{\boldsymbol{W}^{(k)*}\} \quad = \quad argmin_{\{\tilde{\boldsymbol{W}}^{(k)}\}} G_{reg}(\delta, \{\tilde{\boldsymbol{W}}^{(k)}\}_{k=1}^K, \{\boldsymbol{\omega}^{(k)}\}_{k=1}^K, \{\boldsymbol{y}_i\}_{i=1}^N), \quad (24)$$

$$\{\boldsymbol{\omega}^{(k)*}\} \quad = \quad argmin_{\{\tilde{\boldsymbol{\omega}}^{(k)}\}} G_{reg}(\delta, \{\boldsymbol{W}^{(k)}\}_{k=1}^K, \{\tilde{\boldsymbol{\omega}}^{(k)}\}_{k=1}^K, \{\boldsymbol{y}_i\}_{i=1}^N). \quad (25)$$

The proof of Proposition 4 can be found in Appendix C. For $\delta = 0$, we can see that this proposition aligns with Proposition 3.

Finally, we offer a heuristic to initialize $\delta_{init}$ so that the solution to the regularized problem discussed above will be around the uniform partitioning, i.e. $\omega_d^{(k)} \approx 1/K$ for all $k \in \{1, \ldots, K\}$ and $d \in \{1, \ldots, D\}$. Then, as $\delta$ decreases, it will slowly shift towards the right partitioning described in Proposition 3.

**Proposition 5.** *Let $\{\overline{\boldsymbol{\omega}}^{(k)}\}_{k=1}^K \subset [0,1]^D$ be a uniform partitioning, i.e. $\overline{\omega}_d^{(k)} = 1/K$, and the corresponding affinity matrices $\{\overline{\boldsymbol{W}}^{(k)}\}_{k=1}^K$ defined as in Proposition 4 based on these partitions. Let $\{\boldsymbol{\omega}^{(k)}\}_{k=1}^K \subset \{0,1\}^D$ and $\{\boldsymbol{W}^{(k)}\}_{k=1}^K$ be the optimal partitioning solution as discussed in Proposition 3.*

*Define*

$$\delta_{init} \quad \equiv \quad \frac{G(\{\overline{\boldsymbol{W}}^{(k)}\}_{k=1}^K, \{\overline{\boldsymbol{\omega}}^{(k)}\}_{k=1}^K, \{\boldsymbol{y}_i\}_{i=1}^N)}{D \cdot \log(K)}. \quad (26)$$

*Then,*

$$G_{reg}(\delta_{init}, \{\overline{\boldsymbol{W}}^{(k)}\}_{k=1}^K, \{\overline{\boldsymbol{\omega}}^{(k)}\}_{k=1}^K, \{\boldsymbol{y}_i\}_{i=1}^N) \leq G_{reg}(\delta_{init}, \{\boldsymbol{W}^{(k)}\}_{k=1}^K, \{\boldsymbol{\omega}^{(k)}\}_{k=1}^K, \{\boldsymbol{y}_i\}_{i=1}^N) \quad (27)$$

The proof of Proposition 5 can be found in Appendix C.

In Algorithm 1, we outline the steps of our proposed algorithm based on Propositions 4 and 5. In Figure 6, we present an experiment based on the discussed update rules based on Proposition 3 and our algorithm. The plots illustrate the convergence behavior of both optimization processes. As observed on the right plot, the optimization in the original case quickly converges to an incorrect solution and remains there. However, on the left plot, the regularization guides the solution towards a uniform partition at $t = 1$. As the regularization parameter $\delta^{\{t\}}$ decreases, the solution gradually shifts towards the correct partition. Moreover, the regularization effectively controls the influence of the partitioning weights, as intended.

**Computational Complexity.** In the next two paragraphs, we will provide the computational complexity of our algorithm, outlined in Algorithm 1, when the input is either the data or its singular value decomposition (SVD) approximation. To analyze the complexity in the first case, consider a dataset with $N$ observations in $\mathbb{R}^D$, where the features are partitioned into $K$ subsets. The computational complexity of constructing the similarity graph in tSNE using all the features is $O(N^2 D)$. On the other hand, the computational complexity of obtaining $K$ partitions and their corresponding similarity graphs using our approach is $O(KN^2 D)$. Hence the computational complexity of our approach incurs an additional factor of $K$, which is typically small. Nonetheless, our approach can be significantly slower due to the iterative procedure we employ for solving our optimization problem.

To enhance scalability, we also describe an implementation that exploits a low-rank approximation of the data, which can considerably reduce the running time for large datasets. Given that the data is approximated using the $S \ll \min(N, D)$ leading singular vectors of the data, the computational complexity of the graph construction in tSNE will reduce to $O(N^2 S)$, while the computational complexity of obtaining $K$ partitions and their corresponding similarity graphs will be $O(K(S^2 N^2 + S^2 D))$, based on the next proposition.

**Proposition 6.** *Let the data consist of $N$ data points in $\mathbb{R}^D$. Suppose the data is given in the form of a singular value decomposition (SVD) approximation of rank $S \ll N, D$. Then, the computational complexity of the updates suggested in Proposition 4 is $O(K(S^2 N^2 + S^2 D))$.*

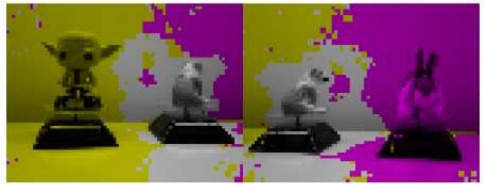

(a) The three extracted partitions using
our method.

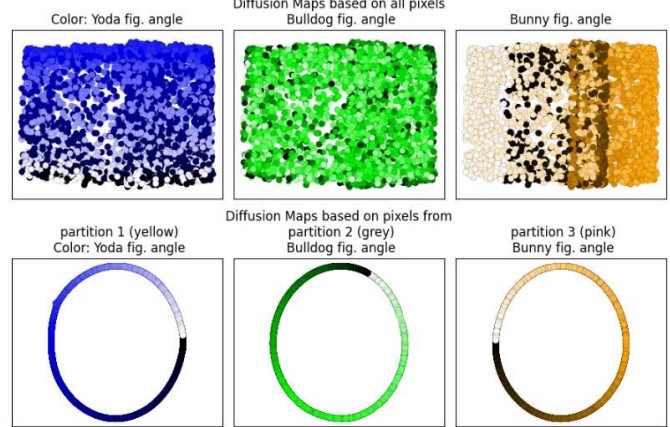

(b) The diffusion maps embedding based on all pixels compared with
the embedding based on each extracted feature partition.

Figure 7: The partitions extracted by our method in Section 3.2

# D   EXPERIMENT DETAILS-

## D.1   EXPERIMENT DETAILS SECTION 3.1

In this experiment, we define an independent dataset and a partially dependant dataset denoted by $\mathcal{Y}, \tilde{\mathcal{Y}} \subset \mathbb{R}^6$, respectively. The two datasets are defined based on an $2-$dimensioanl ellipse defined by

$$\mathcal{A} = \{(8cos(\theta)sin(\phi), 6sin(\theta)sin(\phi), 4cos(\theta))^T \quad | \quad \theta \in [0, 2\pi], \theta \in [0, \pi]\}, \tag{28}$$

and $R, S \in O(3)$ be two orthogonal matrices. Now, we can define the two datasets by

$$\tilde{\mathcal{Y}} = \left\{ \begin{pmatrix} \boldsymbol{Ra} \\ \boldsymbol{Sb} \end{pmatrix} \quad | \quad \boldsymbol{a}, \boldsymbol{b} \in \mathcal{A} \right\} \tag{29}$$

$$\mathcal{Y} = \left\{ \begin{pmatrix} \boldsymbol{Ra} \\ \boldsymbol{Sb} \end{pmatrix} \quad | \quad \boldsymbol{a}, \boldsymbol{b} \in \mathcal{A}, \quad \theta(\boldsymbol{a}) = \theta(\boldsymbol{b}) \right\}, \tag{30}$$

where $\theta(\boldsymbol{a})$ returns the $\theta$ parameterization of the point based on Eq. 28.

The error measure we consider can be understood in different ways. One way is through a non-normalized Jaccard distance between the true feature partition and the estimated one, up to a permutation of the ordering of partitions. Another way is using the $L1$ distance by

$$d(\{\boldsymbol{\omega}^{(k)}\}_{k=1}^K, \{\tilde{\boldsymbol{\omega}}^{(k)}\}_{k=1}^K) = min_{I \in Perm(D)} \sum_{k=1}^K \|\boldsymbol{\omega}^{(k)} - \tilde{\boldsymbol{\omega}}^{(I_k)}\|_1, \tag{31}$$

where $Perm(D)$ is the set of all permutations over $\{1, \dots, D\}$.

## D.2   EXPERIMENT DETAILS SECTION 3.2

The experiment considers the embedding of a set of greyscale images, each formed of a concatenation of a pair of images that were captured by two cameras at the same time. To generate the

embedding of the tSNE embedding based on all the features in Figure 4 we used perplexity of 40 with 100 simulations. The visualizations using the tSNE algorithm of the data based on each partition uses a perplexity of 110. The high perplexity can be attributed to a known issue with tSNE, where it sometimes distorts circular embeddings, resulting in discontinuities. A common solution to this problem is to increase the perplexity of the embedding.

In Figure 7 we present the extracted partitions from this experiment using all features. Additionally, we provide the Diffusion Maps embedding based on all the features and features from each of the extracted partitions. The shown embedding includes the first two non trivial eigenvectors.

Finally we examine the performance of our approach with K-means and spectral clustering. These methods are applied on the features while treating the samples as the coordinates (i.e., applied on the data transpose). The quality of each method is quantified with respect to the true angles of the three figurines, as they are the parameters governing the data (as described in Section 3.2). The score of a partition for a figurine is based on the overlap percentage of the k-nearest neighbors based on the feature partition with those from the figurine's angle of each sample (overlap divided by $k$). This is defined by

$$\rho_{mean}(\mathcal{N}^{method,j}, \mathcal{N}^{figurine}) = \frac{1}{N} \sum_{i=1}^{N} \frac{|\mathcal{N}_i^{figurine} \cap \mathcal{N}_i^{method,j}|}{|\mathcal{N}_i^{figurine}|} \cdot 100,$$

$$\rho_{std}(\mathcal{N}^{method,j}, \mathcal{N}^{figurine}) =$$
$$\sqrt{\frac{1}{N} \sum_{i=1}^{N} \left( \frac{|\mathcal{N}_i^{figurine} \cap \mathcal{N}_i^{method,j}|}{|\mathcal{N}_i^{figurine}|} \cdot 100 - \rho_{mean}(N^{method,j}, N^{figurine}) \right)^2}$$

where $\mathcal{N}_i^{figurine}$ denotes the set of k-nearest neighbors of the i-th sample, determined by the ground truth of the figurine, while $\mathcal{N}_i^{method,j}|$ denotes the set of the k-nearest neighbors of the i-th sample, determined by $j$-th feature partition of the method.

The score of a method with respect to a single figurine, is then calculated based on the partition that achieved the maximal intersection (in expectation over the samples) by

$$\frac{1}{\#figurines} \sum_{figurine} \max_j \rho_{mean}(\mathcal{N}^{method,j}, \mathcal{N}^{figurine}).$$

In the same manner we define the standard deviation that we will use in the results, the mean of the standard deviation for the best assignement. In Table 1 we provide details about the results of the different methods. Finally, in Figure 8 we provide the feature partitions extracted by spectral clustering and K-means.

We note that the parameters used for K-means and spectral clustering were the default parameters used with Sklearn. The exact parameters that were used for each algorithm is given in Figure 7. Specifically for spectral clustering we used a nearest neighbors graph.

### D.3    EXPERIMENT DETAILS SECTION 3.3

In Section 3.3 we applied our method to the dataset used in Qu et al. (2024) using a perplexity of 40 with 100 simulations. As for the preprocessing of the data, the count data matrix initially included 3745 cells with 32285 genes. We used the Normalization scheme used in the Seurat package on it. Meaning, the genes in each cell were divided by the total amount of genes within the cell and then scaled by 10000. Each entry was then individually transformed using log normalization, where each value $x$ was transformed into $\log(1+x)$. Then, we used the highly variable genes method to reduce the amuont of used genes automatically, which resulted in 1708 genes. Finally, we z-scored each gene across the different samples and applied PCA into dimension 100.

In Section 3.3, we applied our method to the dataset used in Qu et al. (2024) that included a count data matrix comprised $3,745$ cells and $32,285$ genes. We employed the normalization scheme from the Seurat package, where the gene counts in each cell were divided by the total number of genes within that cell and then scaled by $10,000$. Each entry was subsequently log-normalized using the

| Method | Agreement of 30-nearest neighbors (std) | Agreement of 50-nearest neighbors (std) |
|---|---|---|
| All features | 8.21 (4.91) | 11.74 (4.75) |
| K- means (K=3) | 18.49 (9.74) | 25.93 (10.61') |
| K-means (K=4) | 18.23 (9.11) | 25.42 (10.31) |
| Spectral clustering (nn=10,K=3, eigs=3) | 18.28 (8.4) | 24.72 (8.82) |
| Spectral clustering (nn=10,K=4,eigs= 4) | 18.85 (8.71) | 25.49 (9.19) |
| Spectral clustering (nn=30,K=3, eigs=3) | 18.12 (8.5) | 24.47 (8.97) |
| Spectral clustering (nn=30,K=4,eigs=4) | 18.19 (8.5) | 24.53 (8.98) |
| Spectral clustering (nn=10,K=3,eigs=5) | 18.04 (8.5) | 24.41 (8.98) |
| Spectral clustering (nn=10,K=4,eigs=5) | 18.63 (8.76) | 25.25 (9.26) |
| Spectral clustering (nn=10,K=3,eigs=10) | 17.51 (7.94) | 23.8 (8.22) |
| Spectral clustering (nn=30,K=4,eigs=10) | 18.94 (10.35) | 25.49 (11.51) |
| **FP (Ours)** | **75.93** (12.22) | **85.15** (8.59) |

Table 1: Performance quality of different clustering methods in capturing the movement patterns of each figurine that govern the data. The table includes the results of applying the metric in Appendix D.2 on all the features, our method (FP), K-means and spectral clustering. The K-means and spectral clustering are applied on the pixels treating the images as coordinates (i.e., applied on the transposed data). The K indicates the amount of clusters used, nn indicates the amount of nearest neighbors, and eig are the amount eigenvectors used. For the Feature Partioning method we partitioned the pixels into 3 partitions.

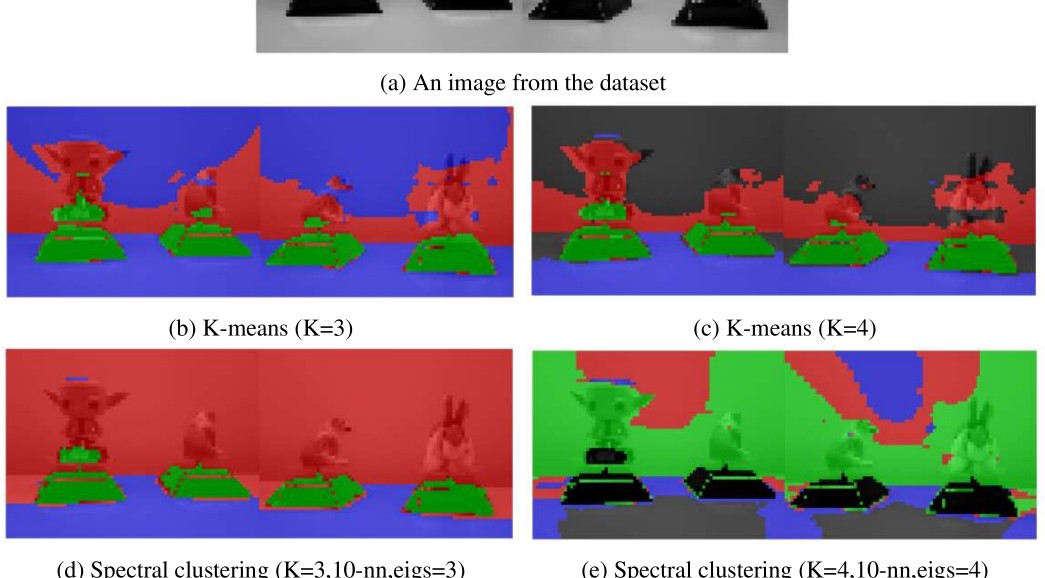

(a) An image from the dataset

(b) K-means (K=3)          (c) K-means (K=4)

(d) Spectral clustering (K=3,10-nn,eigs=3)          (e) Spectral clustering (K=4,10-nn,eigs=4)

Figure 8: The pixels partitions generated using K-means and spectral clustering, by clustering the pixel data while treating the samples as the coordinates (applied on the transposed data). The $K$ indicates the amount of clusters used, 10-nn means that a 10-nearest neighbors graph was used, and eigs means the amount of eignevectors used.

transformation $\log(1+x)$. We then applied the highly variable genes method to automatically reduce the number of genes, resulting in $1,708$ genes. Finally, we z-scored each gene across the samples and performed PCA to reduce the dimensionality to 100. As for our method, we applied Algorithm 1 on the data using a perplexity of 40 and 100 simulations.

Finally, in Figure 9 we show the Diffusion Maps embedding of the data based on the two partitions. As shown, the embedding relates to both attributes of the given cells, unlike the embedding based on all the genes.

### D.4 BIOLOGICAL DATA ANALYSIS - FEATURE SELECTION OF GENES

In this subsection we would like to explore the genes governing the two structures we unraveled in Section 3.3. We employ a feature selection approach in the spirit of He et al. (2005); Lindenbaum et al. (2021) that sorts the features according to the scoring function $S(\tilde{\boldsymbol{W}}, d)$ for each dimension $d$ and affinity matrix $\tilde{\boldsymbol{W}}$, as defined in Eq. 3. Then, we choose the features with the smallest values that intuitively promote the most smooth features with respect to the structure $\boldsymbol{W}$.

When considering partition 1, meaning $\boldsymbol{W}^{(1)}$ and the genes within this partition, we get a Laplacian score for each of these genes. The 40 genes with the smallest score are - Agtr2, Foxd1, Fibin, Col12a1, Dpt, Sox18, Sox11, Thbs2, Lox, Nrp2, Ebf1, Cdkn1c, H19, Dclk1, Igf2, Fgf10, Igf1, Mest, Abi3bp, Sox2, Lef1, Kitl, Itm2a, Sox9, Angptl1, Cd24a, Cdkn1a, Ogn, Cpne5, Nfib, Ptch1, Zfhx4, S100a6, Col23a1, Pde3a, Klf4, Sgip1, Postn, Bmp4, Syne1 (ordered from smallest to highest).

As for the partition 2, we considered the same problem but with respect to $\boldsymbol{W}^{(2)}$ and its genes. The 40 genes with the smallest scores are -Top2a, Cenpf, Nusap1, Cdca8, Hmmr, Ube2c, Ccnb1, Smc4, Tpx2, Prc1, Cenpe, Mki67, Smc2, Spc25, Hist1h1b, Cks2, Hist1h2ap, Cdc20, Birc5, Knl1, Kif23, Ckap2l, Pimreg, Tacc3, Cdk1, Ccna2, Arl6ip1, Sgo2a, Mis18bp1, Incenp, Aurkb, Hmgb2, Plk1, Kif20b, Cenpa, Esco2, Pclaf, Hist1h1e, H2afx, Tuba1c (ordered from smallest to highest).

In Figures 10c and 10d we color the tSNE embeddings based on the different partitions according to the genes extracted above. We can see that the values of most genes from partition 1 are indeed smooth with respect to the tSNE embedding of partition 1, but not of partition 2. This claim holds vise versa, which indicates that the gene partitions have a different underlying structure. It is worth mentioning that a subset of the genes, like Foxd1, are smooth with respect to both embeddings. Specifically, for Foxd1, it relates to genes that are active in DC cell types, which are defined as G1 cell cycle phase type.

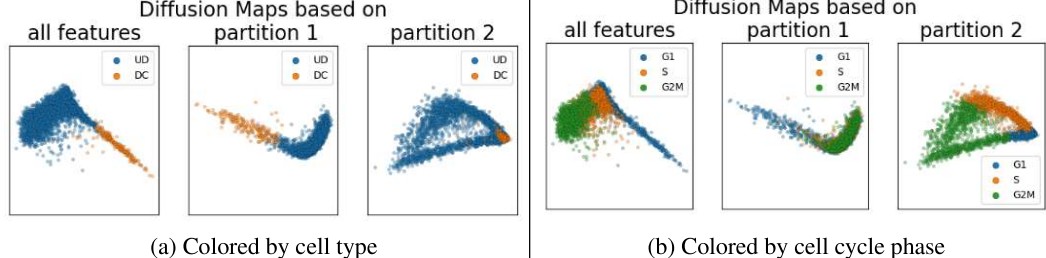

(a) Colored by cell type  (b) Colored by cell cycle phase

Figure 9: The Diffusion Maps embedding of $N = 3745$ cells characterized by their $D = 1708$ genes values as detailed in Section 3.3 colored by different attributes of the cells. In Figure 10, we provide further investigation of these embeddings.

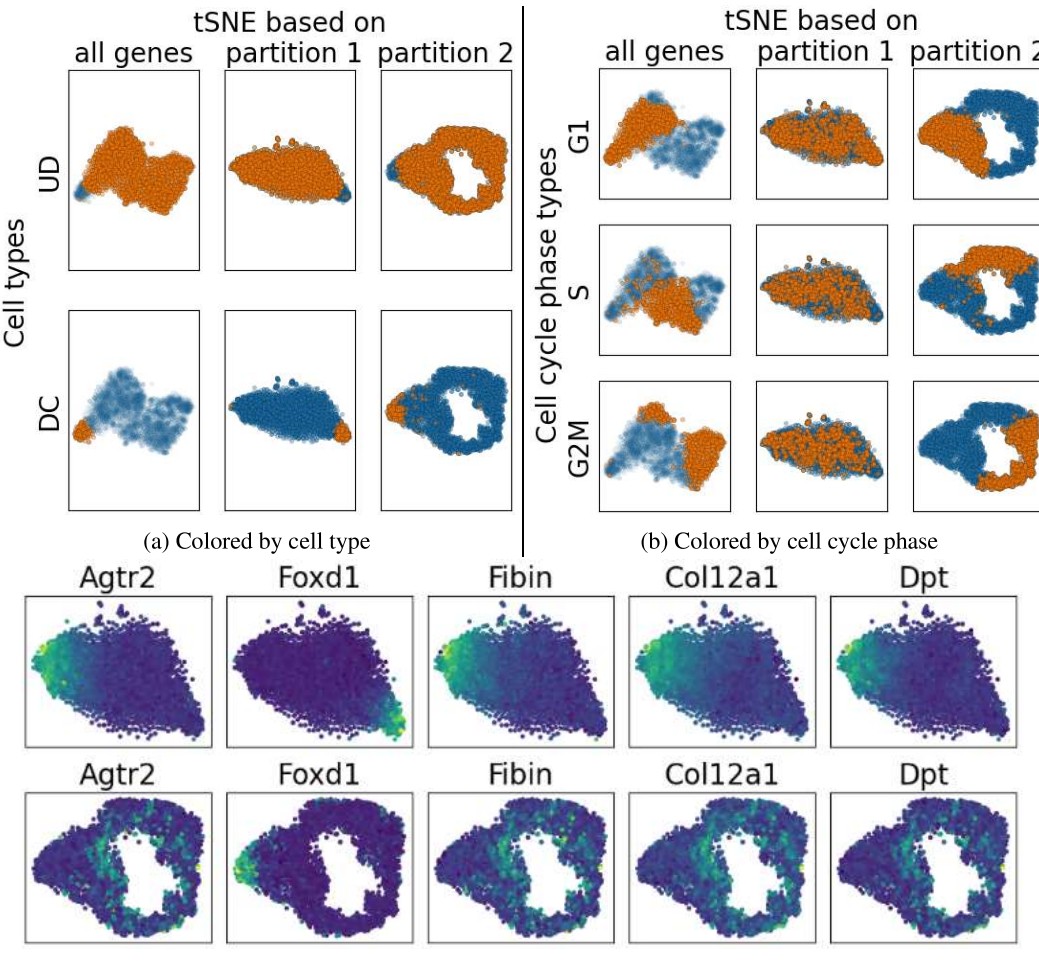

(a) Colored by cell type

(b) Colored by cell cycle phase

(c) The embeddings colored based on genes extracted from Partition 1 using Unsupervised Feature Selection

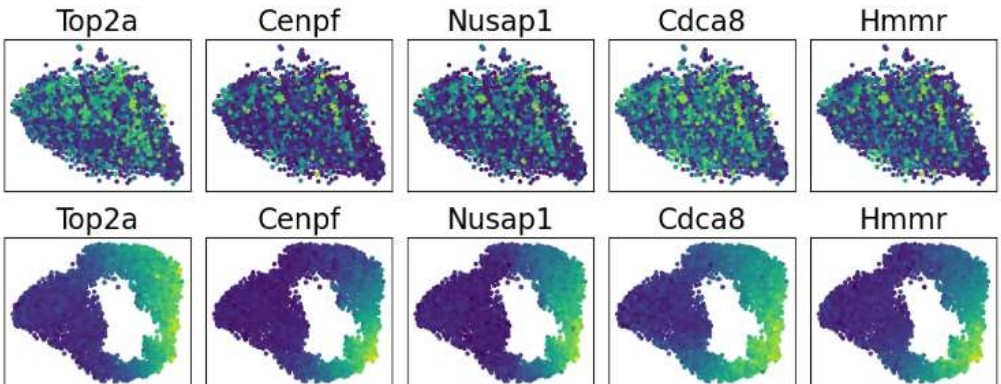

(d) Coloring the embeddings based on genes extracted from Partition 2 using Unsupervised Feature Selection

Figure 10: The tSNE embedding of $N = 3745$ cells characterized by their $D = 1708$ genes values as detailed in Section 3.3. The embeddings are colored by different attributes of the cells, and by the values of specific genes that were selected by an Unsupervised Feature selection method as described in Appendix D.4.

# E PROOFS

*Proofs overview.*

The proofs section is organized into three main parts. First, Appendix E.1 contains the proofs for the results in Section 1. Next, Appendix E.2 presents the proofs for the results in Section 2. Finally, Appendix E.3 includes the proofs of the results in Appendix C.

The proof outline for Appendix E.1 is structured as follows:

1. **Lemma 1**. This is an auxiliary lemma used in Proposition 1 that characterizes the entropy function of a discrete distribution.

2. **The proof of Proposition 1**. The proof begins by converting the optimization problem to a convex one, by relaxing an equality constraint. Then it derives an optimal solution to the optimization problem by using the Karush-Kuhn-Tucker(KKT) conditions for convex optimization problems (Boyd & Vandenberghe (2004)) and Lemma 1. Finally, we show that the solution satisfies the original equality constraint.

3. **Lemma 2**. This is an auxiliary lemma used in Proposition 2 that approximates specific integrals over a manifold in terms of the manifold's properties and density. The lemma uses results from Singer (2006).

4. **The proof of Proposition 2**. The proof begins by asymptotically approximating the function using results from Hein et al. (2005). Then, it derives its expression using Lemma 2 in terms of the manifold's characteristics and density. Finally, a further simplification of the expression is done by referring to results from Osher et al. (2017).

5. **Lemma 3**. This is an auxiliary lemma used in Proposition 3 that characterizes the optimal graphs in a Problem 1 with fixed partitions. The proof begins by converting the optimization problem to a convex one, by relaxing an equality constraint. Then it derives an optimal solution to the optimization problem by using the Karush-Kuhn-Tucker(KKT) conditions for convex optimization problems (Boyd & Vandenberghe (2004)) and Lemma 1. Finally, we show that the solution satisfies the original equality constraint.

6. **Lemma 4**. This is an auxiliary lemma used in Proposition 3 that characterizes the optimal partitions in a Problem 1 with fixed graphs.

7. **The proof of Proposition 3**. The proof divides the problem into two sub-problems, fixing either the feature partitions or their associated graphs in each case. Then it combines the results from Lemmas 3 and 4 to achieve a full characterization of optimal solutions to the problem.

The proof outline for Appendix E.2 is structured as follows:

1. **The proof of Corollary 1.** The proof begins by proving that the optimization problem is convex. Then it derives an optimal solution to the optimization problem by using the Karush-Kuhn-Tucker(KKT) conditions for convex optimization problems (Boyd & Vandenberghe (2004)).

2. **Lemma 5**. This is an auxiliary lemma used in Theorem 1 that characterizes a specific sum in terms of the data setup parameters and the overall dimension of it as defined in Section 2.

3. **Lemma 6**. This is an auxiliary lemma used in Lemma 7 that approximates specific integrals over a manifold in terms of the manifold's properties and density and some predefined parameters. Its proof uses results from Lemma 2.

4. **Lemma 7.** This is an auxiliary lemma used in Theorem 1 that approximates specific integrals over a manifold in terms of the manifold's properties, density and some predefined parameters. Each of the integrals under consideration includes a nested integral, with the inner integral being addressed in Lemma 6.

5. **The proof of Theorem 1.** The proof begins by solving the minimization problem over the graph parameters derived in Corollary 1. It then asymptotically approximates the solution as the dimension tends to infinity, utilizing Lemma 5. Next, it approximates this asymptotic value as the number of samples tends to infinity, applying Lemma 7 to express it in terms of the manifold's properties, density, and partition parameters.

6. **Lemma 8.** This auxiliary lemma, used in Theorem 2, considers the case of two partitions. It demonstrates that the analytical form derived in Theorem 1 is minimized when the partitioning is close to the true feature partitioning under a constraint on the possible partitioning solutions.

7. **The proof of Theorem 2.** The proof builds on Lemma 8 by considering a sequence of problems and demonstrates that the problem is minimized when the true partitioning solution is used.

8. **The proof of Problem 3**. The proof finds a parameter defining the regularization within Problem 3 that shifts the optimal solution towards the uniform partitioning solution.

The proof outline of Appendix C is structured as follows:

1. **Appendix E.3.** This is an auxiliary lemma used in Proposition 4 that characterizes the optimal graphs in a Problem 3 with fixed partitions. The proof begins by converting the optimization problem to a convex one, by relaxing an equality constraint. Then it derives an optimal solution to the optimization problem by using the Karush-Kuhn-Tucker(KKT) conditions for convex optimization problems (Boyd & Vandenberghe (2004)) and Lemma 1. Finally, we show that the solution satisfies the original equality constraint.

2. **The proof of Proposition 4.** The proof divides the problem into two sub-problems, fixing either the feature partitions or their associated graphs in each case. Then it combines the results from Lemma 3 from Section Appendix E.1 and Lemma 9 to achieve a full characterization of optimal solutions to the problem.

## E.1 Proofs of Section 1.1 and Section 1.2

**Lemma 1.** *Consider $\boldsymbol{y}_1, \ldots, \boldsymbol{y}_N \subset \mathbb{R}^D$, and define the function $f : (0, \infty) \to \mathbb{R}$ by*

$$f(\beta) = \sum_{j=1}^{N} \frac{exp(-\|\boldsymbol{y}_j\|^2\beta)}{\sum_{t=1}^{N} exp(-\|\boldsymbol{y}_t\|^2\beta)} \log \frac{exp(-\|\boldsymbol{y}_j\|^2\beta)}{\sum_{s=1}^{N} exp(-\|\boldsymbol{y}_s\|^2\beta)}. \tag{32}$$

*Then $f$ is monotonous non decreasing and continuous in $\beta$.*

*Proof.* We indicate that this proof is used in not new, and is shown here for completeness. We begin by rewriting the function $f$-

$$f(\beta) = \sum_{j=1}^{N} \left( \frac{exp(-\|\boldsymbol{y}_j\|^2\beta)}{\sum_{t=1}^{N} exp(-\|\boldsymbol{y}_t\|^2\beta)} \log \frac{exp(-\|\boldsymbol{y}_j\|^2\beta)}{\sum_{s=1}^{N} exp(-\|\boldsymbol{y}_s\|^2\beta)} \right) \tag{33}$$

$$= \left( \sum_{j=1}^{N} \frac{exp(-\|\boldsymbol{y}_j\|^2\beta)}{\sum_{t=1}^{N} exp(-\|\boldsymbol{y}_t\|^2\beta)} \log(exp(-\|\boldsymbol{y}_j\|^2\beta)) \right) \tag{34}$$

$$- \left( \sum_{j=1}^{N} \frac{exp(-\|\boldsymbol{y}_j\|^2\beta)}{\sum_{t=1}^{N} exp(-\|\boldsymbol{y}_t\|^2\beta)} log(\sum_{s=1}^{N} exp(-\|\boldsymbol{y}_s\|^2\beta)) \right)$$

$$= \left( \sum_{j=1}^{N} \frac{exp(-\|\boldsymbol{y}_j\|^2\beta)}{\sum_{t=1}^{N} exp(-\|\boldsymbol{y}_t\|^2\beta)} \log(exp(-\|\boldsymbol{y}_j\|^2\beta)) \right) \tag{35}$$

$$- \left( \frac{\sum_{j=1}^{N} exp(-\|\boldsymbol{y}_j\|^2\beta)}{\sum_{t=1}^{N} exp(-\|\boldsymbol{y}_t\|^2\beta)} \right) log(\sum_{s=1}^{N} exp(-\|\boldsymbol{y}_s\|^2\beta))$$

$$= -\beta \left( \sum_{j=1}^{N} \frac{exp(-\|\boldsymbol{y}_j\|^2\beta)}{\sum_{t=1}^{N} exp(-\|\boldsymbol{y}_t\|^2\beta)} \|\boldsymbol{y}_j\|^2 \right) - \log(\sum_{s=1}^{N} exp(-\|\boldsymbol{y}_s\|^2\beta)). \tag{36}$$

Now we will show that its first derivative is non negative:

$$
\begin{aligned}
\frac{d}{d\beta}f(\beta) \;=\;& -\left(\sum_{j=1}^{N}\frac{exp(-\|\boldsymbol{y}_j\|^2\beta)}{\sum_{t=1}^{N}exp(-\|\boldsymbol{y}_t\|^2\beta)}\|\boldsymbol{y}_j\|^2\right) \\
& -\beta\sum_{j=1}^{N}exp(-\|\boldsymbol{y}_j\|^2\beta)\|\boldsymbol{y}_j\|^2\cdot\frac{-\|\boldsymbol{y}_j\|^2\sum_{r=1}^{N}exp(-\|\boldsymbol{y}_r\|^2\beta)+\sum_{s=1}^{N}exp(-\|\boldsymbol{y}_s\|^2\beta)\|\boldsymbol{y}_s\|^2}{(\sum_{t=1}^{N}exp(-\|\boldsymbol{y}_t\|^2\beta))^2} \\
& +\frac{\sum_{r=1}^{N}exp(-\|\boldsymbol{y}_r\|^2\beta)\|\boldsymbol{y}_r\|^2}{\sum_{s=1}^{N}exp(-\|\boldsymbol{y}_s\|^2\beta)} & (37)\\
=\;& -\beta\sum_{j=1}^{N}exp(-\|\boldsymbol{y}_j\|^2\beta)\|\boldsymbol{y}_j\|^2\cdot\frac{-\|\boldsymbol{y}_j\|^2\sum_{r=1}^{N}exp(-\|\boldsymbol{y}_r\|^2\beta)+\sum_{s=1}^{N}exp(-\|\boldsymbol{y}_s\|^2\beta)\|\boldsymbol{y}_s\|^2}{(\sum_{t=1}^{N}exp(-\|\boldsymbol{y}_t\|^2\beta))^2} & (38)\\
=\;& \beta\left(\sum_{j=1}^{n}\|\boldsymbol{y}_j\|^4\frac{exp(-\|\boldsymbol{y}_j\|^2\beta)}{\sum_{t=1}^{n}exp(-\|\boldsymbol{y}_t\|^2\beta)}\right)-\beta\left(\sum_{j=1}^{N}\|\boldsymbol{y}_j\|^2\frac{exp(-\|\boldsymbol{y}_j\|^2\beta)}{\sum_{t=1}^{N}exp(-\|\boldsymbol{y}_t\|^2\beta)}\right)^2 & (39)\\
=\;& \beta\cdot\sum_{j=1}^{N}\frac{exp(-\|\boldsymbol{y}_j\|^2\beta)}{\sum_{t=1}^{n}exp(-\|\boldsymbol{y}_t\|^2\beta)}\cdot\left(\|\boldsymbol{y}_j\|^2-\left(\sum_{r=1}^{N}\|\boldsymbol{y}_r\|^2\frac{exp(-\|\boldsymbol{y}_r\|^2\beta)}{\sum_{s=1}^{N}exp(-\|\boldsymbol{y}_s\|^2\beta)}\right)\right)^2 & (40)\\
\geq\;& 0. & (41)
\end{aligned}
$$

Furthermore, we will show that the derivative is bounded and therefore it is continuous with respect to $\beta$. Based on Eq. 40-

$$
\begin{aligned}
\frac{d}{d\beta}f(\beta) \;=\;& \beta\cdot\sum_{j=1}^{N}\frac{exp(-\|\boldsymbol{y}_j\|^2\beta)}{\sum_{t=1}^{n}exp(-\|\boldsymbol{y}_t\|^2\beta)}\cdot\left(\|\boldsymbol{y}_j\|^2-\left(\sum_{r=1}^{N}\|\boldsymbol{y}_r\|^2\frac{exp(-\|\boldsymbol{y}_r\|^2\beta)}{\sum_{s=1}^{N}exp(-\|\boldsymbol{y}_s\|^2\beta)}\right)\right)^2 & (42)\\
=\;& \beta\left(\sum_{j=1}^{N}\frac{exp(-\|\boldsymbol{y}_j\|^2\beta)}{\sum_{t=1}^{n}exp(-\|\boldsymbol{y}_t\|^2\beta)}\|\boldsymbol{y}_j\|^4-\left(\sum_{j=1}^{N}\frac{exp(-\|\boldsymbol{y}_j\|^2\beta)}{\sum_{t=1}^{n}exp(-\|\boldsymbol{y}_t\|^2\beta)}\|\boldsymbol{y}_j\|^2\right)^2\right) & (43)\\
\leq\;& \beta\sum_{j=1}^{N}\frac{exp(-\|\boldsymbol{y}_j\|^2\beta)}{\sum_{t=1}^{n}exp(-\|\boldsymbol{y}_t\|^2\beta)}\|\boldsymbol{y}_j\|^4 & (44)\\
\leq\;& max_{j\in\{1,\dots N\}}\beta\|\boldsymbol{y}_j\|^4 & (45)
\end{aligned}
$$

where in the first derivation we use the variance decomposition into moments.

$\square$

**Proposition 1.** The matrix $\boldsymbol{W}\in[0,1]^{N\times N}$ defined in Eq. 1 is a solution to

$$
\underset{\tilde{\boldsymbol{W}}\in[0,1]^{N\times N}}{\operatorname{argmin}}\quad J(\tilde{\boldsymbol{W}},\{\boldsymbol{y}_1,\dots,\boldsymbol{y}_N\}), \tag{5}
$$

for $d=1,\dots,D$ and $i,j=1,\dots,N$, where $D_{i,i}^{(k)}\equiv\sum_{j=1}^{N}\exp(-\|\boldsymbol{y}_i-\boldsymbol{y}_j\|_{\boldsymbol{\omega}^{(k)}}^2/\epsilon_{k,i})$, the bandwidth parameters $\{\epsilon_{k,i}\}$ are set to satisfy the entropy constraints in Problem 1, and S is the Laplacian-type score defined in Eq. 3.

If there are $T>\alpha$ points which are the 1-nearest neighbors of point $i\in\{1,\dots,N\}$, then $W_{i,j}$ should be set to 1 only for a subset $A\subseteq\{j:j\text{ is among the T nearest neighbors of }i\}$ of size $\alpha$, and $W_{i,j}=0$ otherwise.

*Proof of Proposition 1.* In the following proof, we will solve the problem with an adjusted constraint using an inequality instead of an equality. However, the optimal result will satisfy the equality constraint. Specifically, we consider an inequality negative constraint

$$
\sum_{j=1}^{N}\tilde{W}_{i,j}\log\tilde{W}_{i,j}\leq-\log(\alpha) \tag{46}
$$

for all $i \in \{1, \ldots, N\}$.

Then by showing that the new problem includes convex inequality constraints, affine equality constraints and a convex objective function we can use the KKT conditions for convex problems (See *KKT conditions for convex problems* within Sec. 5.5.3 in Boyd & Vandenberghe (2004)) to find an optimal solution. Specifically, it states that any primal and dual sets parameters that satisfy the KKT conditions are the primal and dual optimal solution with zero duality gap. Finally, we will need to verify that the optimal solution satisfies the original negative entropy equality constraint.

We begin by showing that the new suggested problem satisfies the conditions of the KKT conditions for convex problems. The objective and the constraint that the sum over the rows of the graph should be 1 is convex as both are affine functions of the graphs. Finally, we will show that the inequality constraint in Eq. 46 is convex for all $j, i \in \{1, \ldots, N\}$ such that $i \neq j$, as $\tilde{W}_{i,i} = 0$. We will use second order condition that implies the convexity of the function (see Sec. 3.1.4 in Boyd & Vandenberghe (2004)). This is done by showing the that the Hessian of the function is positive semi-definite, which is demonstrated here by proving it is diagonal with positive values

$$\frac{d^2}{d^2\tilde{W}_{i,j}} \sum_{s,t=1}^{N} \tilde{W}_{s,t} \log(\tilde{W}_{s,t}) = \frac{d}{d\tilde{W}_{i,j}} \left(1 + \log(\tilde{W}_{i,j})\right) \tag{47}$$

$$= \frac{1}{\tilde{W}_{i,j}} \tag{48}$$

$$\geq 1. \tag{49}$$

$$\frac{d}{d\tilde{W}_{s,t}} \frac{d}{d\tilde{W}_{i,j}} \sum_{s,t=1}^{N} \tilde{W}_{s,t} \log(\tilde{W}_{s,t}) = \frac{d}{d\tilde{W}_{s,t}} \left(1 + \log(\tilde{W}_{i,j})\right) \tag{50}$$

$$= 0 \tag{51}$$

for all $i, j, s, t \in \{1, \ldots, N\}$ satisfying $i \neq j$ and $s \neq t$.

Next, we define the Lagrangian of the minimization problem over $\tilde{\boldsymbol{W}}$ by

$$J(\tilde{\boldsymbol{W}}, \{\boldsymbol{y}_i\}_{i=1}^{N}) \equiv \sum_{i,j=1}^{N} \tilde{W}_{i,j} \|\boldsymbol{y}_i - \boldsymbol{y}_j\|^2 \tag{52}$$

$$+ \sum_{i=1}^{N} \epsilon_i \left(\log(\alpha) + \sum_{j=1}^{N} \tilde{W}_{i,j} \log(\tilde{W}_{i,j})\right) \tag{53}$$

$$+ \sum_{i=1}^{N} \mu_i \left(\sum_{j=1}^{N} \tilde{W}_{i,j} - 1\right) \tag{54}$$

where $\epsilon_i \geq 0$ and $\mu_i \in \mathbb{R}$ for $i = 1, \ldots, N$.

We will find a solution $\boldsymbol{W}$ that attain the KKT conditions, as they are sufficient for optimality. We begin with the stationary condition-

$$0 = \frac{dJ(\boldsymbol{W}, \{\boldsymbol{y}_i\}_{i=1}^{N})}{dW_{i,j}} \tag{55}$$

$$= \|\boldsymbol{y}_i - \boldsymbol{y}_j\|^2 + \epsilon_i \left(1 + \log(W_{i,j})\right) + \mu_i \tag{56}$$

$$W_{i,j} = \exp\left(-\frac{\|\boldsymbol{y}_i - \boldsymbol{y}_j\|^2 + \epsilon_i + \mu_i}{\epsilon_i}\right). \tag{57}$$

The KKT condition primal feasibility condition on the on the equality constraint induces-

$$1 \quad = \quad \sum_{j=1}^{N} W_{i,j} \tag{58}$$

$$= \quad \sum_{j \neq i} \exp\left(-\frac{\|\boldsymbol{y}_i - \boldsymbol{y}_j\|^2 + \epsilon_i + \mu_i}{\epsilon_i}\right) \tag{59}$$

$$\exp\left(\frac{\mu_i}{\epsilon_i}\right) \quad = \quad \sum_{j \neq i} \exp\left(-\frac{\|\boldsymbol{y}_i - \boldsymbol{y}_j\|^2 + \epsilon_i}{\epsilon_i}\right). \tag{60}$$

By pushing it back into Eq. 57 we get that for all $j \neq i$

$$W_{i,j} \quad = \quad \frac{\exp\left(-\frac{\|\boldsymbol{y}_i - \boldsymbol{y}_j\|^2 + \epsilon_i}{\epsilon_i}\right)}{\sum_{t \neq i} \exp\left(-\frac{\|\boldsymbol{y}_i - \boldsymbol{y}_t\|^2 + \epsilon_i}{\epsilon_i}\right)} \tag{61}$$

$$= \quad \frac{\exp\left(-\frac{\|\boldsymbol{y}_i - \boldsymbol{y}_j\|^2}{\epsilon_i}\right)}{\sum_{t \neq i} \exp\left(-\frac{\|\boldsymbol{y}_i - \boldsymbol{y}_t\|^2}{\epsilon_i}\right)}. \tag{62}$$

Furthermore, to satisfy the KKT conditions on the inequality constraint, we define any $\epsilon_i$ for all $i \in \{1, \ldots, N\}$ by

$$\epsilon_i \quad s.t. \quad \sum_j W_{i,j} \log W_{i,j} = -\log(\alpha). \tag{63}$$

The function $\sum_j W_{i,j} \log W_{i,j}$ is a continuous, non increasing monotonous in $\epsilon_i$ according to Lemma 1. Now, under the reasonable assumption that $\{\boldsymbol{y}_i\}$ are $n$ distinct points such $\epsilon_i$ exists based on the intermediate value theorem (and can computationally be found up to the wanted accuracy using binary search). $\qquad\square$

**Lemma 2.** *Let $\mathcal{M} \subset \mathbb{R}^D$ be a smooth compact Riemannian manifold and let $g : \mathcal{M} \to \mathbb{R}_+$ be some smooth positive over it. Let $\Delta_{\mathcal{M}} : \mathcal{M} \to \mathbb{R}$ be the Laplace Beltrami operator over $\mathcal{M}$. Then, there exists $\tilde{\epsilon}(\mathcal{M})$ so that for any $\epsilon < \tilde{\epsilon}(\mathcal{M})$ and $\boldsymbol{x} \in \mathcal{M}$*

$$\frac{1}{C} \int_{\boldsymbol{x} \in \mathcal{M}} \exp\left(-\frac{\|\boldsymbol{x} - \boldsymbol{y}\|^2}{2\epsilon}\right) g(\boldsymbol{x}) d\boldsymbol{x} = g(\boldsymbol{y}) + \epsilon/2(E(\boldsymbol{y})g(\boldsymbol{y}) + \Delta_{\mathcal{M}}g(\boldsymbol{y})) + O(\epsilon^2) \tag{64}$$

$$= g(\boldsymbol{y})(1 + O(\epsilon)) \tag{65}$$

$$\frac{1}{C} \int_{\boldsymbol{x} \in \mathcal{M}} \exp\left(-\frac{\|\boldsymbol{x} - \boldsymbol{y}\|^2}{2\epsilon}\right) g(\boldsymbol{x})(x_d - y_d)^2 d\boldsymbol{x} = \epsilon g(\boldsymbol{y})\|\nabla_{\mathcal{M}} y_d\|^2 + O(\epsilon^2) \tag{66}$$

$$= \epsilon g(\boldsymbol{y})\|\nabla_{\mathcal{M}} y_d\|^2(1 + O(\epsilon)) \tag{67}$$

*for all $d \in \{1, \ldots, D\}$, where $E(\boldsymbol{x})$ is a smooth scalar function of the curvature of $\mathcal{M}$ at $\boldsymbol{x} \in \mathcal{M}$ and $C = (2\pi\epsilon)^{dim(\mathcal{M})/2}$.*

*Proof.* The proof of Eq. 64 is given in Singer (2006) (see Eq. 2.11). Now we begin proving Eq. 66. As $\boldsymbol{x}$ tends to $\boldsymbol{y}$

$$\Delta_{\mathcal{M}}g(\boldsymbol{x})(x_d - y_d)^2 \tag{68}$$

$$= g(\boldsymbol{x})(x_d - y_d)\Delta_{\mathcal{M}}(x_d - y_d) + (x_d - y_d)\Delta_{\mathcal{M}}g(\boldsymbol{x})(x_d - y_d) \tag{69}$$

$$+ 2\langle \nabla_{\mathcal{M}}g(\boldsymbol{x})(x_d - y_d), \nabla_{\mathcal{M}}(x_d - y_d)\rangle$$

$$= g(\boldsymbol{x})(x_d - y_d)\Delta_{\mathcal{M}}x_d + (x_d - y_d)\Delta_{\mathcal{M}}g(\boldsymbol{x})(x_d - y_d) \tag{70}$$

$$+ 2(x_d - y_d)\langle \nabla_{\mathcal{M}}g(\boldsymbol{x}), \nabla_{\mathcal{M}}x_d\rangle + 2g(\boldsymbol{x})\|\nabla_{\mathcal{M}}x_d\|^2$$

$$\to 2g(\boldsymbol{x})\|\nabla_{\mathcal{M}}x_d\|^2\big|_{\boldsymbol{x}=\boldsymbol{y}}, \tag{71}$$

where we use the identity $\Delta_{\mathcal{M}}\rho \cdot h = h\Delta_{\mathcal{M}}(\rho) + \rho\Delta_{\mathcal{M}}(h) + 2\langle\nabla_{\mathcal{M}}\rho, \nabla_{\mathcal{M}}h\rangle$ for smooth functions $\rho, h : \mathcal{M} \to \mathbb{R}$, based on the fact both $f$ and the restriction of the manifold onto a specific dimension is smooth with respect to them manifold $\mathcal{M}$. Furthermore we use the fact that by combining it with the fact that the manifold is compact, we get that the Laplace Beltrami operator and the gradients over $x_d, g(\boldsymbol{x}), g(\boldsymbol{x}) \cdot x_d$ are bounded.

Now, by combining it with Eq. 64 we get that for sufficiently small $\epsilon$

$$\frac{1}{C}\int_{\boldsymbol{x}\in\mathcal{M}}\exp\left(-\frac{\|\boldsymbol{x}-\boldsymbol{y}\|^2}{2\epsilon}\right)g(\boldsymbol{x})(x_d - y_d)^2 d\boldsymbol{x} = \epsilon/2(2g(\boldsymbol{y})\|\nabla_{\mathcal{M}}y_d\|^2) + O(\epsilon^2) \tag{72}$$

$$= \epsilon g(\boldsymbol{y})\|\nabla_{\mathcal{M}}y_d\|^2 + O(\epsilon^2). \tag{73}$$

$\square$

**Proposition 2.** Let $\mathcal{M} \subset \mathbb{R}^D$ be a smooth compact Riemannian manifold with an intrinsic dimension denoted by $dim(\mathcal{M}) < D$. Assume $\boldsymbol{y}_1, \dots, \boldsymbol{y}_N \in \mathcal{M}$ are independently sampled from a smooth non-vanishing distribution $f$ over $\mathcal{M}$, and that $\boldsymbol{W}$ is constructed as described in Eq. 1. Then, for all sufficiently small $\epsilon_1, \dots, \epsilon_N \in \mathbb{R}_+$, we have for all $i \in \{1, \dots, N\}$,

$$\sum_{j=1}^{N}W_{i,j}\|\boldsymbol{y}_i - \boldsymbol{y}_j\|^2 \overset{N\to\infty}{\underset{a.s.}{\longrightarrow}} \frac{\epsilon_i}{2} \cdot dim(\mathcal{M}) + O(\epsilon_i^2). \tag{6}$$

*Proof of Proposition 2.* For simplicity denote $\boldsymbol{x} = \boldsymbol{y}_i$. Let's examine the inquired term in Eq. 6

$$\sum_{j=1}^{N}W_{i,j}\|\boldsymbol{x} - \boldsymbol{y}_j\|^2 \tag{74}$$

$$= \sum_{j\in\{1,\dots,N\}/\{i\}}\frac{\exp(-\|\boldsymbol{x}-\boldsymbol{y}_j\|^2/\epsilon_i)}{\sum_{t\in\{1,\dots,N\}/\{i\}}\exp(-\|\boldsymbol{x}-\boldsymbol{y}_t\|^2/\epsilon_i)} \cdot \|\boldsymbol{x}-\boldsymbol{y}_j\|^2 \tag{75}$$

$$\overset{N\to\infty}{\underset{a.s.}{\longrightarrow}} \int_{\boldsymbol{y}\in\mathcal{M}}\frac{\exp(-\|\boldsymbol{x}-\boldsymbol{y}\|^2/\epsilon_i)\|\boldsymbol{x}-\boldsymbol{y}\|^2 f(\boldsymbol{y})d\boldsymbol{y}}{\int_{\boldsymbol{z}\in\mathcal{M}}\exp(-\|\boldsymbol{x}-\boldsymbol{z}\|^2/\epsilon_i)f(\boldsymbol{z})d\boldsymbol{z}}, \tag{76}$$

where the derivation made in the last line is based on Lemma 2 in Hein et al. (2005). Specifically by considering $A_{0,\epsilon_i,n-1}$ and the samples $\{\boldsymbol{y}_i\}_{i\in I}$ where $I = \{1, \dots, N\}/\{i\}$. To be explicit, in the context of the derivation this Lemma states that: Let $\boldsymbol{x} \in \mathcal{M}$ and $g$ be a continuous function on $\mathcal{M}$. Then, there exists a constant $C \geq 1$ so that for any $\epsilon, \delta \in (0, 1/C)$ such that

$$Pr\left(\left|\frac{\sum_{j\in I}exp(-\|\boldsymbol{x}-\boldsymbol{y}_j\|^2/\epsilon)g(\boldsymbol{y}_j)}{\sum_{j\in I}exp(-\|\boldsymbol{x}-\boldsymbol{y}_j\|^2/\epsilon)} - \frac{E_{\boldsymbol{y}\in\mathcal{M}}[exp(-\|\boldsymbol{x}-\boldsymbol{y}\|^2/\epsilon)g(\boldsymbol{y})]}{E_{\boldsymbol{z}\in\mathcal{M}}[exp(-\|\boldsymbol{x}-\boldsymbol{z}\|^2/\epsilon)]}\right| \leq \delta\right)$$
$$\geq 1 - CN \cdot exp(-Nh^{dim(\mathcal{M})\delta^2}/C).$$

Next, we decompose the Euclidean distance and apply Lemma 2 to approximate different terms within the integral

$$= \sum_{d=1}^{D}\int_{\boldsymbol{y}\in\mathcal{M}}\frac{\exp(-\|\boldsymbol{x}-\boldsymbol{y}\|^2/\epsilon_i)(x_d - y_d)^2 f(\boldsymbol{y})d\boldsymbol{y}}{\int_{\boldsymbol{z}\in\mathcal{M}}\exp(-\|\boldsymbol{x}-\boldsymbol{z}\|^2/\epsilon_i)f(\boldsymbol{z})d\boldsymbol{z}} \tag{77}$$

$$= \sum_{d=1}^{D}\frac{(\epsilon_i/2)f(\boldsymbol{x})\|\nabla_{\mathcal{M}}x_d\|^2(1 + O(\epsilon_i))}{f(\boldsymbol{x})(1 + O(\epsilon_i))} \tag{78}$$

$$= \sum_{d=1}^{D}(\epsilon_i/2)\|\nabla_{\mathcal{M}}x_d\|^2(1 + O(\epsilon_i)) \cdot \frac{1}{1 + O(\epsilon_i)} \tag{79}$$

$$= \sum_{d=1}^{D}(\epsilon_i/2)\|\nabla_{\mathcal{M}}x_d\|^2(1 + O(\epsilon_i)) \cdot (1 + O(\epsilon_i)) \tag{80}$$

$$= \left(\frac{\epsilon_i}{2}\sum_{d=1}^{D}\|\nabla_{\mathcal{M}}x_d\|^2\right) + O(\epsilon_i^2), \tag{81}$$

where we use the first-order approximation of $1/(1+a) = 1-a+O(a^2) = 1+O(a)$ for sufficiently small $a$. Finally, by using proposition 3.1 from Osher et al. (2017) we get

$$= \frac{\epsilon_i}{2} \cdot \dim(\mathcal{M}) + O\left(\epsilon^2\right), \tag{82}$$

and thus, the proof is complete. □

**Lemma 3.** *Let $\alpha \in (1, N-1)$ and let $\{\boldsymbol{\omega}^{(k)}\}_{k=1}^K, \subset [0,1]^D$ be a partition solution as defined in Problem 1. Define a set of $K$ affinity matrices $\{\boldsymbol{W}^{(k)}\} \subset [0,1]^{N \times N}$ by*

$$W_{i,j}^{(k)} = \begin{cases} \dfrac{\exp\left(-\frac{\|\boldsymbol{y}_i - \boldsymbol{y}_j\|_{\boldsymbol{\omega}^{(k)}}^2}{\epsilon_{k,i}}\right)}{\sum_{t=1}^n \exp\left(-\frac{\|\boldsymbol{y}_i - \boldsymbol{y}_t\|_{\boldsymbol{\omega}^{(k)}}^2}{\epsilon_{k,i}}\right)} & \text{for } j \neq i \\ 0 & \text{else} \end{cases} \tag{83}$$

*for all $i, j = 1, \dots, N$, and $K = 1, \dots, K$ where $\epsilon_{k,i}$ satisfies*

$$\sum_{j=1}^N W_{i,j}^{(k)} \log W_{i,j}^{(k)} = -\log(\alpha). \tag{84}$$

*Then, a minimizer of the function G (defined in Eq. 7) over matrices that satisfy the constraints in Problem 1 is*

$$\{\boldsymbol{W}^{(k)}\} = \underset{\{\tilde{\boldsymbol{W}}^{(k)}\}}{\arg\min} G(\{\tilde{\boldsymbol{W}}^{(k)}\}_{k=1}^K, \{\boldsymbol{\omega}^{(k)}\}_{k=1}^K, \{\boldsymbol{y}_1, \dots, \boldsymbol{y}_N\}). \tag{85}$$

*Proof.* In the following proof, we will solve the problem with an adjusted constraint using an inequality instead of an equality. Specifically, we consider an inequality negative constraint

$$\sum_{j=1}^N \tilde{W}_{i,j}^{(k)} \log \tilde{W}_{i,j}^{(k)} \leq -\log(\alpha) \tag{86}$$

for all $k \in \{1, \dots, K\}$ and $i \in \{1, \dots, N\}$. Then by showing that the new problem includes convex inequality constraints, affine equality constraints and a convex objective function we can use the KKT conditions for convex problems (See *KKT conditions for convex problems* within Sec. 5.5.3 in Boyd & Vandenberghe (2004)) to find optimal solutions. Specifically, it states that any primal and dual sets parameters that satisfy the KKT conditions are the primal and dual optimal solution with zero duality gap. Finally, we will need to verify that the optimal solution satisfies the original negative entropy equality constraint.

We begin by showing that the new suggested problem satisfies the conditions of the KKT conditions for convex problems. The objective and the constraint that the sum over the rows of the graph should be 1 is convex as both are affine functions of the graphs. Finally, we will show that the inequality constraint in Eq. 86 is convex for all $j, i \in \{1, \dots, N\}$ such that $i \neq j$, as $\tilde{W}_{i,i}^{(k)} = 0$, and for all $k \in \{1, \dots, K\}$. We will use second order condition that implies the convexity of the function (see Sec. 3.1.4 in Boyd & Vandenberghe (2004)). This is done by showing the that the Hessian of the function is positive semi-definite, which is demonstrated here by proving it is diagonal with positive values

$$\frac{d^2}{d^2 \tilde{W}_{i,j}^{(k)}} \sum_{s,t=1}^N \tilde{W}_{s,t}^{(k)} \log(\tilde{W}_{s,t}^{(k)}) = \frac{d}{d\tilde{W}_{i,j}^{(k)}} \left(1 + \log(\tilde{W}_{i,j}^{(k)})\right) \tag{87}$$

$$= \frac{1}{\tilde{W}_{i,j}^{(k)}} \tag{88}$$

$$\geq 1. \tag{89}$$

$$\frac{d}{d\tilde{W}_{s,t}^{(k)}} \frac{d}{d\tilde{W}_{i,j}^{(k)}} \sum_{s,t=1}^N \tilde{W}_{s,t}^{(k)} \log(\tilde{W}_{s,t}^{(k)}) = \frac{d}{d\tilde{W}_{s,t}^{(k)}} \left(1 + \log(\tilde{W}_{i,j}^{(k)})\right) \tag{90}$$

$$= 0 \tag{91}$$

for all $i, j, s, t \in \{1, \ldots, N\}$ satisfying $i \neq j$ and $s \neq t$.

Next, we define the Lagrangian of the minimization problem over $\{\tilde{\boldsymbol{W}}^{(k)}\}$ by

$$L(\{\tilde{\boldsymbol{W}}^{(k)}\}_{k=1}^K, \{\boldsymbol{\omega}^{(k)}\}_{k=1}^K) \equiv \sum_{k=1}^K \sum_{i,j=1}^N \tilde{W}_{i,j}^{(k)} \|\boldsymbol{y}_i - \boldsymbol{y}_j\|_{\boldsymbol{\omega}^{(k)}}^2 \tag{92}$$

$$+ \sum_{k=1}^K \sum_{i=1}^N \epsilon_{k,i} \left( \log(\alpha) + \sum_{j=1}^N \tilde{W}_{i,j}^{(k)} \log(\tilde{W}_{i,j}^{(k)}) \right) \tag{93}$$

$$+ \sum_{k=1}^K \sum_{i=1}^N \mu_{k,i} \left( \sum_{j=1}^N \tilde{W}_{i,j}^{(k)} - 1 \right) \tag{94}$$

where $\epsilon_{k,i} \geq 0$ and $\mu_{k,i} \in \mathbb{R}$ for $k = 1, \ldots, K$ and $i = 1, \ldots, N$.

We will find a solution $\{\boldsymbol{W}^{(k)}\}_{k=1}^K$ that attain the KKT conditions, as they are sufficient for optimality. We begin with the stationary condition-

$$0 = \frac{dL(\{\boldsymbol{W}^{(k)}\}, \{\boldsymbol{\omega}^{(k)}\})}{dW_{i,j}^{(k)}} \tag{95}$$

$$= \|\boldsymbol{y}_i - \boldsymbol{y}_j\|_{\boldsymbol{\omega}^{(k)}}^2 + \epsilon_{k,i} \left( 1 + \log(W_{i,j}^{(k)}) \right) + \mu_{k,i} \tag{96}$$

$$W_{i,j}^{(k)} = \exp\left( -\frac{\|\boldsymbol{y}_i - \boldsymbol{y}_j\|_{\boldsymbol{\omega}^{(k)}}^2 + \epsilon_{k,i} + \mu_{k,i}}{\epsilon_{k,i}} \right). \tag{97}$$

The KKT condition primal feasibility condition on the on the equality constraint induces-

$$1 = \sum_{j=1}^N W_{i,j}^{(k)} \tag{98}$$

$$= \sum_{j \neq i} \exp\left( -\frac{\|\boldsymbol{y}_i - \boldsymbol{y}_j\|_{\boldsymbol{\omega}^{(k)}}^2 + \epsilon_{k,i} + \mu_{k,i}}{\epsilon_{k,i}} \right) \tag{99}$$

$$\exp\left( \frac{\mu_{k,i}}{\epsilon_{k,i}} \right) = \sum_{j \neq i} \exp\left( -\frac{\|\boldsymbol{y}_i - \boldsymbol{y}_j\|_{\boldsymbol{\omega}^{(k)}}^2 + \epsilon_{k,i}}{\epsilon_{k,i}} \right). \tag{100}$$

By pushing it back into Eq. 97 we get that for all $j \neq i$ and $k \in \{1, \ldots, N\}$

$$W_{i,j}^{(k)} = \frac{\exp\left( -\frac{\|\boldsymbol{y}_i - \boldsymbol{y}_j\|_{\boldsymbol{\omega}^{(k)}}^2 + \epsilon_{k,i}}{\epsilon_{k,i}} \right)}{\sum_{t \neq i} \exp\left( -\frac{\|\boldsymbol{y}_i - \boldsymbol{y}_t\|_{\boldsymbol{\omega}^{(k)}}^2 + \epsilon_{k,i}}{\epsilon_{k,i}} \right)} \tag{101}$$

$$= \frac{\exp\left( -\frac{\|\boldsymbol{y}_i - \boldsymbol{y}_j\|_{\boldsymbol{\omega}^{(k)}}^2}{\epsilon_{k,i}} \right)}{\sum_{t \neq i} \exp\left( -\frac{\|\boldsymbol{y}_i - \boldsymbol{y}_t\|_{\boldsymbol{\omega}^{(k)}}^2}{\epsilon_{k,i}} \right)}. \tag{102}$$

Furthermore, to satisfy the KKT conditions on the inequality constraint we define any $\epsilon_{k,i}$ for all $i \in \{1, \ldots, N\}$ and $k \in \{1, \ldots, K\}$ by

$$\epsilon_{k,i} \quad s.t. \quad \sum_j W_{i,j}^{(k)} \log W_{i,j}^{(k)} = -\log(\alpha). \tag{103}$$

The function $\sum_j W_{i,j}^{(k)} \log W_{i,j}^{(k)}$ is a continuous, non increasing monotonous in $\epsilon_i$ according to Lemma 1. Now, under the reasonable assumption that $\{\boldsymbol{y}_i\}$ are $n$ distinct datapoints such $\epsilon_i$ exists

based on the intermediate value theorem (and can be found up to the wanted accuracy using binary search).

$\square$

**Lemma 4.** *Let $\{\boldsymbol{W}^{(k)}\} \in [0,1]^{n \times n}$ be affinity matrices under the constraints in Problem 1. Define a partitioning solution $\{\boldsymbol{\omega}^{(k)}\} \subset \{0,1\}^D$ by*

$$\omega_d^{(k)} = \begin{cases} 1 & \text{if } k = \tilde{k} \text{ for some } \tilde{k} \in \Omega(d) \\ 0 & \text{else} \end{cases}, \tag{104}$$

*for $d = 1, \ldots, D$, where $\Omega(d) = argmin_{k \in \{1,\ldots,K\}} S(\boldsymbol{W}^{(k)}, d)$ and S is defined in Eq. 3.*

*Then, a minimizer of the objective function in Problem 1 while fixing the graph parameters satisfy its constraints is*

$$\{\boldsymbol{\omega}^{(k)}\} = argmin_{\{\tilde{\boldsymbol{\omega}}^{(k)}\}} G(\{\boldsymbol{W}^{(k)}\}_{k=1}^K, \{\tilde{\boldsymbol{\omega}}^{(k)}\}_{k=1}^K, \{\boldsymbol{y}_1, \ldots, \boldsymbol{y}_N\}). \tag{105}$$

*Proof.* The minimization problem is

$$\underset{\{\tilde{\boldsymbol{\omega}}^{(k)}\}}{argmin} \sum_{d=1}^D \sum_{k=1}^K \tilde{\omega}_d^{(k)} \left( \sum_{i=1}^n \sum_{j=1}^n W_{i,j}^{(k)} ((\boldsymbol{y}_i)_d - (\boldsymbol{y}_j)_d)^2 \right). \tag{106}$$

As the constraints are $\sum_k \omega_d^{(k)} = 1$ for any $d = 1, \ldots, D$, the problem can be decomposed into $D$ independent problems. Meaning that for any $d \in \{1, \ldots, D\}$ we need to solve

$$\underset{\{\tilde{\omega}_d^{(k)}\}}{argmin} \sum_{k=1}^K \tilde{\omega}_d^{(k)} \left( \sum_{i=1}^n \sum_{j=1}^n W_{i,j}^{(k)} ((\boldsymbol{y}_i)_d - (\boldsymbol{y}_j)_d)^2 \right). \tag{107}$$

Hence, the solution to this problem is

$$\omega_d^{(k)} = \begin{cases} 1 & \text{if } k = \tilde{k} \text{ for some } \tilde{k} \in I(d) \\ 0 & \text{else} \end{cases} \tag{108}$$

for all $k \in \{1, \ldots, K\}$ and $d \in \{1, \ldots, D\}$, where

$$\Omega(d) = argmin_{k \in \{1,\ldots,K\}} S(\boldsymbol{W}^{(k)}, d). \tag{109}$$

$\square$

**Proposition 3** An optimal partitioning solution $\{\boldsymbol{\omega}^{(k)}\}_{k=1}^K$ and the corresponding affinity matrices $\{\boldsymbol{W}^{(k)}\}_{k=1}^K$ that solve Problem 1 must satisfy

$$\omega_d^{(k)} = \begin{cases} 1 & \text{if } k = \tilde{k} \text{ for some } \tilde{k} \in \Omega(d) \\ 0 & \text{else} \end{cases}, \tag{9}$$

$$W_{i,j}^{(k)} = \begin{cases} \exp\left(-\|\boldsymbol{y}_i - \boldsymbol{y}_j\|_{\boldsymbol{\omega}^{(k)}}^2 / \epsilon_{k,i}\right) / D_{i,i}^{(k)} & \text{if } i \neq j \\ 0 & \text{else} \end{cases}, \tag{10}$$

$$\Omega(d) = \underset{k \in \{1,\ldots,K\}}{\arg\min} \sum_{i,j=1}^n S(\boldsymbol{W}^{(k)}, d), \tag{11}$$

*Proof of Proposition 3.* The proposition aims to characterize optimal parameters of Problem 1. We begin by defining two sub-problems that are related to it, each focusing on minimizing one set of parameters while keeping the other set fixed:

$$\{\boldsymbol{W}^{(k)}\} = \underset{\{\tilde{\boldsymbol{W}}^{(k)}\}_k}{\arg\min} \ G(\{\tilde{\boldsymbol{W}}^{(k)}\}_{k=1}^K, \{\tilde{\boldsymbol{\omega}}^{(k)}\}_{k=1}^K, \{\boldsymbol{y}_1, \ldots, \boldsymbol{y}_N\}) \tag{110}$$

$$\{\boldsymbol{\omega}^{(k)}\} = \underset{\{\tilde{\boldsymbol{\omega}}^{(k)}\}_k}{\arg\min} \ G(\{\tilde{\boldsymbol{W}}^{(k)}\}_{k=1}^K, \{\tilde{\boldsymbol{\omega}}^{(k)}\}_{k=1}^K, \{\boldsymbol{y}_1, \ldots, \boldsymbol{y}_N\}). \tag{111}$$

where the parameters are limited to the constraints stated in Problem 1.

These two sub-problems are considered in Lemmas 3 and 4. Specifically, in Lemma 3 the optimal graph matrices are derived in the form of

$$
W_{i,j}^{(k)} \quad = \quad \begin{cases} \dfrac{\exp\left(-\dfrac{\|\boldsymbol{y}_i-\boldsymbol{y}_j\|_{\tilde{\boldsymbol{\omega}}^{(k)}}^2}{\epsilon_{k,i}}\right)}{\sum_{t=1}^{n}\exp\left(-\dfrac{\|\boldsymbol{y}_i-\boldsymbol{y}_t\|_{\tilde{\boldsymbol{\omega}}^{(k)}}^2}{\epsilon_{k,i}}\right)} & \text{for } j \neq i \\ 0 & \text{else} \end{cases}, \tag{112}
$$

$$\tag{113}$$

for $i,j = 1,\ldots,N$ and $k = 1,\ldots,K$, and $\epsilon_{k,i}$ satisfy

$$
\sum_{j=1}^{N} W_{i,j}^{(k)} \log W_{i,j}^{(k)} = -\log(\alpha). \tag{114}
$$

On the other hand, in Lemma 4 an optimal partitioning parameters are derived in the form of

$$
\omega_d^{(k)} \quad = \quad \begin{cases} 1 & \text{if } k = \tilde{k} \text{ for some } \tilde{k} \in \Omega(d) \\ 0 & \text{else} \end{cases} \tag{115}
$$

for $d = 1,\ldots,D$ and $k = 1,\ldots,K$, where $\Omega(d) = argmin_{k \in \{1,\ldots,K\}} S(\tilde{\boldsymbol{W}}^{(k)}, d)$ and $S$ is defined in Eq. 3.

Therefore there exists parameters of this form that minimizes Problem 1. $\qquad\square$

## E.2 Proofs of Section 2

**Corollary 1.** Let $\{\tilde{\boldsymbol{\omega}}^{(k)}\}$ be a partitioning that satisfies the constraints in Problem 2. Then, the graph affinity matrices that minimize Eq. 13 while fixing the partitioning parameters $\{\tilde{\boldsymbol{\omega}}^{(k)}\}$ are

$$W_{i,j}^{(k)} = \begin{cases} \exp\left(-\frac{\|\boldsymbol{y}_i - \boldsymbol{y}_j\|^2_{\tilde{\boldsymbol{\omega}}^{(k)}}}{\epsilon \cdot (1/D)\sum_{d=1}^{D}\tilde{\omega}_d^{(k)}}\right) / \sum_{t=1}^{N}\exp\left(-\frac{\|\boldsymbol{y}_i - \boldsymbol{y}_t\|^2_{\tilde{\boldsymbol{\omega}}^{(k)}}}{\epsilon \cdot (1/D)\sum_{d=1}^{D}\tilde{\omega}_d^{(k)}}\right) & \text{if } i \neq j \\ 0 & \text{else} \end{cases} . \quad (14)$$

*The Proof of Corollary 1.* For simplicity we denote the following term distance term

$$\gamma(\boldsymbol{y}_i, \boldsymbol{y}_j; \boldsymbol{\omega}^{(k)}) \equiv \frac{\|\boldsymbol{y}_i - \boldsymbol{y}_j\|^2_{\boldsymbol{\omega}^{(k)}}}{(1/D)\sum_{d=1}^{D}\omega_d^{(k)}}. \quad (116)$$

Then the problem we are trying to minimize is:

$$\min_{\left\{\tilde{\boldsymbol{W}}^{(k)}\right\}_k,} \tilde{G}\left(\{\tilde{\boldsymbol{W}}^{(k)}\}_{k=1}^{K}, \{\tilde{\boldsymbol{\omega}}\}_{k=1}^{K}, \{\boldsymbol{y}_1, \ldots, \boldsymbol{y}_N\}\right) + \epsilon\sum_{k=1}^{K}\sum_{i,j=1}^{N}\tilde{W}_{i,j}^{(k)}\log\left(\tilde{W}_{i,j}^{(k)}\right) \quad (117)$$

$$= \min_{\left\{\tilde{\boldsymbol{W}}^{(k)}\right\}_k,}\sum_{k=1}^{K}\sum_{i,j=1}^{N}\tilde{W}_{i,j}^{(k)}\cdot\frac{\|\boldsymbol{y}_i - \boldsymbol{y}_j\|^2_{\tilde{\boldsymbol{\omega}}^{(k)}}}{(1/D)\sum_d\tilde{\omega}_d^{(k)}} + \epsilon\sum_{k=1}^{K}\sum_{i,j=1}^{N}\tilde{W}_{i,j}^{(k)}\log\left(\tilde{W}_{i,j}^{(k)}\right) \quad (118)$$

$$= \min_{\left\{\tilde{\boldsymbol{W}}^{(k)}\right\}_k,}\sum_{k=1}^{K}\sum_{i,j=1}^{N}\tilde{W}_{i,j}^{(k)}\gamma(\boldsymbol{y}_i, \boldsymbol{y}_j; \boldsymbol{\omega}^{(k)}) + \epsilon\sum_{k=1}^{K}\sum_{i,j=1}^{N}\tilde{W}_{i,j}^{(k)}\log\left(\tilde{W}_{i,j}^{(k)}\right) \quad (119)$$

with the following constraints $\sum_{j=1}^{N}\tilde{W}_{i,j}^{(k)} = 1$, $\tilde{W}_{i,i}^{(k)} = 0$, for $i = 1, \ldots, N$, $k = 1, \ldots, K$ and $d = 1, \ldots, D$.

We begin by showing that the problem satisfies the conditions of the KKT conditions for convex problems. First, the constraints that constraint the sum over the rows of the graph to be 1 is convex as it is an affine function of the graphs. Second, we can see that the the first term in our problem is linear and therefore convex. By showing that the second sum term is convex as well, we can use the fact that sum of convex functions is a convex function Boyd & Vandenberghe (2004). To do so, we will use the second order condition that implies the convexity of the function (see Sec. 3.1.4 in Boyd & Vandenberghe (2004)). This is done by showing the that the Hessian of the function is positive semi-definite, which is demonstrated here by proving it is diagonal with positive values

$$\frac{d^2}{d^2\tilde{W}_{i,j}^{(k)}}\sum_{s,t=1}^{N}\tilde{W}_{s,t}^{(k)}\log(\tilde{W}_{s,t}^{(k)}) = \frac{d}{d\tilde{W}_{i,j}^{(k)}}\left(1 + \log(\tilde{W}_{i,j}^{(k)})\right) \quad (120)$$

$$= \frac{1}{\tilde{W}_{i,j}^{(k)}} \quad (121)$$

$$\geq 1. \quad (122)$$

$$\frac{d}{d\tilde{W}_{s,t}^{(k)}}\frac{d}{d\tilde{W}_{i,j}^{(k)}}\sum_{s,t=1}^{N}\tilde{W}_{s,t}^{(k)}\log(\tilde{W}_{s,t}^{(k)}) = \frac{d}{d\tilde{W}_{s,t}^{(k)}}\left(1 + \log(\tilde{W}_{i,j}^{(k)})\right) \quad (123)$$

$$= 0 \quad (124)$$

for all $i, j, s, t \in \{1, \ldots, N\}$ satisfying $i \neq j$ and $s \neq t$.

Next, we define the Lagrangian of the minimization problem over $\{\tilde{\boldsymbol{W}}^{(k)}\}$ by

$$L(\{\tilde{\boldsymbol{W}}^{(k)}\}_{k=1}^K, \{\boldsymbol{\omega}^{(k)}\}_{k=1}^K) \equiv \sum_{k=1}^K \sum_{i,j=1}^N \tilde{W}_{i,j}^{(k)} \gamma(\boldsymbol{y}_i, \boldsymbol{y}_j; \boldsymbol{\omega}^{(k)}) \tag{125}$$

$$+ \epsilon \sum_{k=1}^K \sum_{i,j=1}^N \tilde{W}_{i,j}^{(k)} \log(\tilde{W}_{i,j}^{(k)}) \tag{126}$$

$$+ \sum_{k=1}^K \sum_{i=1}^N \mu_{k,i} \left( \sum_{j=1}^N \tilde{W}_{i,j}^{(k)} - 1 \right) \tag{127}$$

where $\epsilon_{k,i} \geq 0$ and $\mu_{k,i} \in \mathbb{R}$ for $k = 1, \dots, K$ and $i = 1, \dots, N$.

We will find a solution $\{\boldsymbol{W}^{(k)}\}_{k=1}^K$ that attain the KKT conditions, as they are sufficient for optimality. We begin with the stationary condition-

$$0 = \frac{dL(\{\boldsymbol{W}^{(k)}\}, \{\boldsymbol{\omega}^{(k)}\})}{dW_{i,j}^{(k)}} \tag{128}$$

$$= \gamma(\boldsymbol{y}_i, \boldsymbol{y}_j; \boldsymbol{\omega}^{(k)}) + \epsilon \left( 1 + \log(W_{i,j}^{(k)}) \right) + \mu_{k,i} \tag{129}$$

$$W_{i,j}^{(k)} = \exp\left( -\frac{\gamma(\boldsymbol{y}_i, \boldsymbol{y}_j; \boldsymbol{\omega}^{(k)}) + \epsilon + \mu_{k,i}}{\epsilon} \right). \tag{130}$$

The KKT condition primal feasibility condition on the on the equality constraint induces-

$$1 = \sum_{j=1}^N W_{i,j}^{(k)} \tag{131}$$

$$= \sum_{j \neq i} \exp\left( -\frac{\gamma(\boldsymbol{y}_i, \boldsymbol{y}_j; \boldsymbol{\omega}^{(k)}) + \epsilon + \mu_{k,i}}{\epsilon} \right) \tag{132}$$

$$\exp\left( \frac{\mu_{k,i}}{\epsilon} \right) = \sum_{j \neq i} \exp\left( -\frac{\gamma(\boldsymbol{y}_i, \boldsymbol{y}_j; \boldsymbol{\omega}^{(k)}) + \epsilon}{\epsilon} \right). \tag{133}$$

By pushing it back into Eq. 130 we get that for all $j \neq i$ and $k \in \{1, \dots, N\}$

$$W_{i,j}^{(k)} = \frac{\exp\left( -\frac{\gamma(\boldsymbol{y}_i, \boldsymbol{y}_j; \boldsymbol{\omega}^{(k)}) + \epsilon}{\epsilon} \right)}{\sum_{t \neq i} \exp\left( -\frac{\gamma(\boldsymbol{y}_i, \boldsymbol{y}_y; \boldsymbol{\omega}^{(k)}) + \epsilon}{\epsilon} \right)} \tag{134}$$

$$= \frac{\exp\left( -\frac{\gamma(\boldsymbol{y}_i, \boldsymbol{y}_j; \boldsymbol{\omega}^{(k)})}{\epsilon} \right)}{\sum_{t \neq i} \exp\left( -\frac{\gamma(\boldsymbol{y}_i, \boldsymbol{y}_t; \boldsymbol{\omega}^{(k)})}{\epsilon} \right)}. \tag{135}$$

Therefore we can conclude the proof with the derived optimal solution form. $\qquad \square$

**Lemma 5.** *Assume the configuration described in Section 2, and let $\{\boldsymbol{\omega}^{(k)}\} \subset \{0,1\}^D$ be a partitioning solution that satisfies its conditions.*

*Let $i \neq j \in \{1, \dots, N\}$ be two samples, and $k \in \{1, \dots, K\}$ be a single partition then*

$$\sum_{d=1}^D \frac{\omega_d^{(k)} (\boldsymbol{y}_i - \boldsymbol{y}_j)_d^2}{(1/D) \sum_{\tilde{d}=1}^D \omega_{\tilde{d}}^{(k)}} = \sum_{s=1}^K \frac{p_s^{(k)} \left( \|\boldsymbol{x}_i^{(s)} - \boldsymbol{x}_j^{(s)}\|^2 + \|\boldsymbol{x}_i^{(K+1)} - \boldsymbol{x}_j^{(K+1)}\|^2 \right)}{\sum_{t=1}^K p_t^{(k)} D_t / D} \tag{136}$$

$$+ O\left( \sqrt{\frac{1}{D} \cdot \sum_{s=1}^K \frac{2 p_s^{(k)} \left( \|\boldsymbol{x}_i^{(s)} - \boldsymbol{x}_j^{(s)}\|^4 + \|\boldsymbol{x}_i^{(common)} - \boldsymbol{x}_j^{(common)}\|^4 \right)}{(D_s / D) \left( \sum_{t=1}^K p_t^{(k)} (D_t / D) \right)^2}} \right) \tag{137}$$

where we define each partition according to structure of the samples structure in Section 2, meaning $\boldsymbol{\omega}^{(k)} = ((\boldsymbol{\omega}^{(k,1)})^T, \ldots, (\boldsymbol{\omega}^{(k,K)})^T)^T$ for all $k \in \{1, \ldots, K\}$ and $p_s^{(k)} = \sum_{d=1}^{D_s} \omega_d^{(k,s)}/D_s$.

*Proof.* Rewrite the left hand side term in Eq. 136 results in

$$\sum_{d=1}^{D} \frac{\omega_d^{(k)}(\boldsymbol{y}_i - \boldsymbol{y}_j)_d^2}{(1/D) \sum_{\tilde{d}=1}^{D} \omega_{\tilde{d}}^{(k)}} = \sum_{s=1}^{K} \sum_{d=1}^{D_s} \frac{\omega_d^{(k,s)}(\boldsymbol{y}_i^{(s)} - \boldsymbol{y}_j^{(s)})_d^2}{(1/D) \sum_{d=1}^{D} \omega_{\tilde{d}}^{(k)}} \tag{138}$$

$$= \sum_{s=1}^{K} \sum_{d=1}^{D_s} \frac{\omega_d^{(k,s)} \left( \boldsymbol{P}^{(s)} \left( \begin{bmatrix} \boldsymbol{x}_i^{(s)} \\ \boldsymbol{x}_i^{(K+1)} \end{bmatrix} - \begin{bmatrix} \boldsymbol{x}_j^{(s)} \\ \boldsymbol{x}_j^{(K+1)} \end{bmatrix} \right) \right)_d^2}{(1/D) \sum_{t=1}^{K} \sum_{\tilde{d}=1}^{D_t} \omega_{\tilde{d}}^{(k)}}. \tag{139}$$

As shown, the examined function can be decomposed into different parts of $\boldsymbol{y}^{(s)}$, for $s = 1, \ldots, K$. For simplicity, we will the value for an arbitrary $s \in \{1, \ldots, K\}$ and the general result will follow. We denote $\boldsymbol{a} \equiv [\boldsymbol{x}_i^{(s)}; \boldsymbol{x}_i^{(K+1)}] - [\boldsymbol{x}_j^{(s)}; \boldsymbol{x}_j^{(K+1)}]$ and get that

$$\mathbb{E}\left[ \sum_{d=1}^{D_s} \frac{\omega_d^{(k,s)} \cdot \left( \boldsymbol{P}^{(s)} \boldsymbol{a} \right)_d^2}{(1/D) \sum_{\tilde{d}=1}^{D} \omega_{\tilde{d}}^{(k)}} \right] = \frac{\sum_{d=1}^{D_s} \omega_d^{(k,s)}}{(1/D) \sum_{\tilde{d}=1}^{D} \omega_{\tilde{d}}^{(k)}} \cdot \frac{1}{D_s} \|\boldsymbol{a}\|^2 \tag{140}$$

$$= \frac{\sum_{d=1}^{D_s} \omega_d^{(k,s)}}{(1/D) \sum_{t=1}^{T} \left( \sum_{\tilde{d}=1}^{D_t} \omega_{\tilde{d}}^{(k,t)} \right)} \cdot \frac{1}{D_s} \|\boldsymbol{a}\|^2 \tag{141}$$

$$= \frac{p_s^{(k)}}{\sum_{t=1}^{K} p_t^{(k)} D_t/D} \cdot \|\boldsymbol{a}\|^2 \tag{142}$$

$$Var\left[ \sum_{d=1}^{D_s} \frac{\omega_d^{(k,s)} \left( \boldsymbol{P}^{(s)} \boldsymbol{a} \right)_d^2}{(1/D) \sum_{\tilde{d}=1}^{D} \omega_{\tilde{d}}^{(k)}} \right] = \sum_{d=1}^{D_s} \left( \frac{\omega_d^{(k,s)}}{(1/D) \sum_{\tilde{d}=1}^{D} \omega_{\tilde{d}}^{(k)}} \right)^2 Var\left[ \left( \boldsymbol{P}^{(s)} \boldsymbol{a} \right)_d^2 \right] \tag{143}$$

$$= \sum_{d=1}^{D_s} \frac{\omega_d^{(k,s)}}{\left( (1/D) \sum_{\tilde{d}=1}^{D} \omega_{\tilde{d}}^{(k)} \right)^2} \cdot \frac{2\|\boldsymbol{a}\|^4}{D_s^2} \tag{144}$$

$$= \frac{(1/D_s) \sum_{d=1}^{D_s} \omega_d^{(k,s)}}{\left( (1/D) \sum_{t=1}^{K} \left( \sum_{\tilde{d}=1}^{D_t} \omega_{\tilde{d}}^{(k,t)} \right) \right)^2} \cdot \frac{2\|\boldsymbol{a}\|^4}{D_s}$$

$$= \frac{p_s^{(k)}}{\left( \sum_{t=1}^{K} p_t^{(k)} D_t/D \right)^2} \cdot \frac{2\|\boldsymbol{a}\|^4}{D_s} \tag{145}$$

as $\boldsymbol{P}^{(s)} \boldsymbol{a} \sim \mathcal{N}_{D_s}\left( \boldsymbol{0}, \frac{1}{D_s} \|\boldsymbol{a}\|^2 \boldsymbol{I} \right)$ following the fact that $P_{i,j}^{(s)} \sim \mathcal{N}(0, 1/D_s)$ for any $i, j \in \{1, \ldots, d_s + d_{common}\} \times \{1, \ldots, D_s\}$, and $p_t^{(k)} = (1/D_t) \sum_{d=1}^{D_t} \omega_d^{(k,t)}$ as defined in the lemma's statement for any $k, t \in \{1, \ldots, K\}$. $\square$

**Lemma 6.** *Let $f$ be a non-vanishing smooth distribution over a smooth compact Riemannian manifold $\mathcal{M}$ and let $\boldsymbol{y} \in \mathcal{M}$. Let $\beta_1, \ldots, \beta_K \in (0, 1)$ that satisfy $\sum_{k=1}^{K} \beta_k = 1$. There exists $\tilde{\epsilon}(\mathcal{M}, f)$ such that for any $\epsilon \le \tilde{\epsilon}^2/\sqrt{K}$ any $q_1, \ldots, q_K \in [\sqrt{\epsilon}, 1 - (K-1)\sqrt{\epsilon}]$ and any $s \in \{1, \ldots, K\}$ :*

$$\log\left( \int_{x \in \mathcal{M}} \exp\left( -\frac{\|\boldsymbol{x} - \boldsymbol{y}\|^2 q_s}{\epsilon \sum_{t=1}^{K} q_t \beta_t} \right) f(\boldsymbol{x}) d\boldsymbol{x} \right) \tag{146}$$

$$= \frac{dim(\mathcal{M})}{2} \log\left( \frac{\pi \epsilon \sum_{t=1}^{K} q_t \beta_t}{q_s} \right) + \log(f(\boldsymbol{y})) + O\left( \sqrt{\epsilon} \right),$$

*and*

$$\log \left( \int_{x \in \mathcal{M}} \exp \left( -\frac{\|\boldsymbol{x} - \boldsymbol{y}\|^2 \sum_{s=1}^{K} q_s}{\epsilon \sum_{t=1}^{K} q_t \beta_t} \right) f(\boldsymbol{x}) d\boldsymbol{x} \right) \tag{147}$$
$$= \frac{dim(\mathcal{M})}{2} \log \left( \frac{\pi \epsilon \sum_{t=1}^{K} q_t \beta_t}{\sum_{s=1}^{K} q_s} \right) + \log \left( f(\boldsymbol{y}) \right) + O\left( \sqrt{\epsilon} \right).$$

*Proof.* We begin by showing Eq. 146. Based on Lemma 2, there exists $\tilde{\epsilon}(\mathcal{M})$ so that for any $\epsilon < \tilde{\epsilon}(\mathcal{M})$

$$\int_{\boldsymbol{x} \in \mathcal{M}} \exp \left( -\frac{\|\boldsymbol{x} - \boldsymbol{y}\|^2}{\epsilon} \right) f(\boldsymbol{x}) d\boldsymbol{x} = (\pi \epsilon)^{dim(\mathcal{M})/2} f(\boldsymbol{y})(1 + O(\epsilon)). \tag{148}$$

Second, if $\epsilon \leq \tilde{\epsilon}^2/\sqrt{K}$ then

$$\frac{\epsilon \sum_{t=1}^{K} q_t \beta_t}{q_s} \leq \frac{\epsilon \sqrt{\sum_{t=1}^{K} \beta_t^2} \sqrt{\sum_{t=1}^{K} q_t^2}}{q_s} \leq \frac{\epsilon \sqrt{\sum_{t=1}^{K} \beta_t} \sqrt{\sum_{t=1}^{K} q_t}}{q_s} \leq \frac{\epsilon \sqrt{K}}{\sqrt{\epsilon}} = \sqrt{\epsilon K} \leq \tilde{\epsilon}, \tag{149}$$

by using the Cauchy-Schwartz inequality, the fact that $q_s, \beta_s \in [0, 1]$ for all $s \in \{1, \ldots, K\}$ and the assumption that $\sum_{s=1}^{K} \beta_s = 1$. Therefore

$$\log \left( \int_{x \in \mathcal{M}} \exp \left( -\frac{\|\boldsymbol{x} - \boldsymbol{y}\|^2 q_s}{\epsilon \sum_{t=1}^{K} q_t \beta_t} \right) f(\boldsymbol{y}) d\boldsymbol{y} \right) \tag{150}$$
$$= \log \left( \left( \frac{\pi \epsilon \sum_{t=1}^{K} q_t \beta_t}{q_s} \right)^{\dim(\mathcal{M})/2} f(\boldsymbol{x}) \left( 1 + O\left( \sqrt{\epsilon} \right) \right) \right) \tag{151}$$
$$= \frac{\dim(\mathcal{M})}{2} \log \left( \frac{\pi \epsilon \sum_{t=1}^{K} q_t \beta_t}{q_s} \right) + \log \left( f(\boldsymbol{x}) \right) + \log \left( 1 + O\left( \sqrt{\epsilon} \right) \right) \tag{152}$$
$$= \frac{\dim(\mathcal{M})}{2} \log \left( \frac{\pi \epsilon \sum_{t=1}^{K} q_t \beta_t}{q_s} \right) + \log \left( f(\boldsymbol{x}) \right) + O\left( \sqrt{\epsilon} \right). \tag{153}$$

where we used Lemma 2 in the derivation of Eq. 151, and the first order approximation of $log(1 + a) = a$ for small enough $a$.

Now, as for Eq. 147. If $\epsilon \leq \tilde{\epsilon}^2/\sqrt{K}$ then

$$\frac{\epsilon \sum_{t=1}^{K} q_t \beta_t}{\sum_{s=1}^{K} q_s} \leq \frac{\epsilon \sum_{t=1}^{K} q_t \beta_t}{q_r} \leq \tilde{\epsilon}, \tag{154}$$

for any $r \in \{1, \ldots, K\}$. Then, Eq. 147 follows using similar derivation.

$\square$

**Lemma 7.** *Let $f_1, \ldots, f_{K+1}$ be non-vanishing smooth distributions over smooth compact Riemannian manifolds $\mathcal{M}_1, \ldots, \mathcal{M}_{K+1}$, respectively. Define $f$ to be a non vanishing smooth distribution over the product manifold $\mathcal{M} = \mathcal{M}_1 \times \ldots \times \mathcal{M}_{K+1}$ by $f(\boldsymbol{x}^{(1)}, \ldots, \boldsymbol{x}^{(K+1)}) = \prod_{k=1}^{K+1} f_k(\boldsymbol{x}^{(k)})$ for any $(\boldsymbol{x}^{(1)}, \ldots, \boldsymbol{x}^{(K+1)}) \in \mathcal{M}$.*

*Let $\beta_1, \ldots, \beta_K \in (0, 1)$ that satisfy $\sum_k \beta_k = 1$. There exists $\tilde{\epsilon}(\mathcal{M}, f)$ such that for any $\epsilon \leq \tilde{\epsilon}^2/\sqrt{K}$ and any $q_1, \ldots, q_K \in [\sqrt{\epsilon}, 1 - (K - 1)\sqrt{\epsilon}]$:*

$$\int_{\boldsymbol{y} \in \mathcal{M}} f(\boldsymbol{y}) \log \Bigg( \tag{155}$$

$$\int_{\boldsymbol{x} \in \mathcal{M}} \left( \prod_{s=1}^{K} \exp \left( -\frac{\|\boldsymbol{x}^{(s)} - \boldsymbol{y}^{(s)}\|^2 q_s}{\epsilon \sum_{t=1}^{K} q_t \beta_t} - \frac{\|\boldsymbol{x}^{(K+1)} - \boldsymbol{y}^{(K+1)}\|^2 q_s}{\epsilon \sum_{t=1}^{K} q_t \beta_t} \right) \right) f(\boldsymbol{x}) d\boldsymbol{x} \Bigg) d\boldsymbol{y}$$

$$= \sum_{s=1}^{K} \frac{dim(\mathcal{M}_k)}{2} \log \left( \frac{\pi \epsilon \sum_{t=1}^{K} q_t \beta_t}{q_s} \right) \tag{156}$$

$$+ \frac{dim(\mathcal{M}_{K+1})}{2} \log \left( \frac{\pi \epsilon \sum_{t=1}^{K} q_t \beta_t}{\sum_{s=1}^{K} q_s} \right) - h(f) + O(\sqrt{\epsilon}),$$

*where $h(f)$ is the differential entropy of $f$ defined by $h(f) = -\int_{\boldsymbol{y} \in \mathcal{M}} f(\boldsymbol{y}) \log(f(\boldsymbol{y})) d\boldsymbol{y}$.*

*Proof.* We begin by rewriting the terms inside the logarithmic term in Eq. 155 using the separability properties of $\mathcal{M}$ and $f$

$$\log \int_{\boldsymbol{x} \in \mathcal{M}} \left( \prod_{s=1}^{K} \exp \left( -\frac{\|\boldsymbol{x}^{(s)} - \boldsymbol{y}^{(s)}\|^2 q_s}{\epsilon \sum_{t=1}^{K} q_t \beta_t} - \frac{\|\boldsymbol{x}^{(K+1)} - \boldsymbol{y}^{(K+1)}\|^2 q_s}{\epsilon \sum_{t=1}^{K} q_t \beta_t} \right) \right) f(\boldsymbol{x}) d\boldsymbol{x} \tag{157}$$

$$= \log \Bigg( \left( \prod_{s=1}^{K} \int_{\boldsymbol{x}^{(s)} \in \mathcal{M}_s} \exp \left( -\frac{\|\boldsymbol{x}^{(s)} - \boldsymbol{y}^{(s)}\|^2 q_s}{\epsilon \sum_{t=1}^{K} q_t \beta_t} \right) f_s(\boldsymbol{x}^{(s)}) d\boldsymbol{x}^{(s)} \right) \tag{158}$$

$$\cdot \left( \int_{\boldsymbol{x}^{(K+1)} \in \mathcal{M}_{K+1}} \exp \left( -\frac{\|\boldsymbol{x}^{(K+1)} - \boldsymbol{y}^{(K+1)}\|^2 \sum_{s=1}^{K} q_s}{\epsilon \sum_{t=1}^{K} q_t \beta_t} \right) f_{K+1}(\boldsymbol{x}^{(K+1)}) d\boldsymbol{x}^{(K+1)} \right) \Bigg)$$

$$= \sum_{s=1}^{K} \log \left( \int_{\boldsymbol{x}^{(s)} \in \mathcal{M}_s} \exp \left( -\frac{\|\boldsymbol{x}^{(s)} - \boldsymbol{y}^{(s)}\|^2 q_s}{\epsilon \sum_{t=1}^{K} q_t \beta_t} \right) f_k(\boldsymbol{x}^{(s)}) d\boldsymbol{x}^{(s)} \right) \tag{159}$$

$$+ \log \left( \int_{\boldsymbol{x}^{(K+1)} \in \mathcal{M}_{K+1}} \exp \left( -\frac{\|\boldsymbol{x}^{(K+1)} - \boldsymbol{y}^{(K+1)}\|^2 \sum_{s=1}^{K} q_s}{\epsilon \sum_{t=1}^{K} q_t \beta_t} \right) f_{K+1}(\boldsymbol{x}^{(K+1)}) d\boldsymbol{x}^{(K+1)} \right).$$

Based on Lemma 6 there exists $\tilde{\epsilon}(\mathcal{M}, f)$ such that for any $\epsilon \leq \tilde{\epsilon}^2 / \sqrt{K}$ and $q_1, \ldots, q_K \in [\sqrt{\epsilon}, 1 - (K-1)\sqrt{\epsilon}]$ then the equation above is equal to

$$= \sum_{s=1}^{K} \left( \frac{\dim(\mathcal{M}_s)}{2} \log \left( \frac{\pi \epsilon \sum_{t=1}^{K} q_t \beta_t}{q_s} \right) + \log \left( f_s(\boldsymbol{y}) \right) \right) \tag{160}$$

$$+ \frac{\dim(\mathcal{M}_{K+1})}{2} \log \left( \frac{\pi \epsilon \sum_{t=1}^{K} q_t \beta_t}{\sum_{s=1}^{K} q_s} \right) + \log \left( f_{K+1}(\boldsymbol{y}) \right) + O(\sqrt{\epsilon})$$

$$= \sum_{s=1}^{K} \left( \frac{\dim(\mathcal{M}_s)}{2} \log \left( \frac{\pi \epsilon \sum_{t=1}^{K} q_t \beta_t}{q_s} \right) \right) \tag{161}$$

$$+ \frac{\dim(\mathcal{M}_{K+1})}{2} \log \left( \frac{\pi \epsilon \sum_{t=1}^{K} q_t \beta_t}{\sum_{s=1}^{K} q_s} \right) + \log(f(\boldsymbol{y})) + O(\sqrt{\epsilon}).$$

Now, we can push this term inside Eq. 155 and derive that it can be rewritten by

$$\int_{\boldsymbol{y}\in\mathcal{M}} f(\boldsymbol{y})\left(\sum_{s=1}^{K}\left(\frac{\dim(\mathcal{M}_s)}{2}\log\left(\frac{\pi\epsilon\sum_{t=1}^{K}q_t\beta_t}{q_s}\right)\right)\right) \tag{162}$$

$$+\frac{\dim(\mathcal{M}_{K+1})}{2}\log\left(\frac{\pi\epsilon\sum_{t=1}^{K}q_t\beta_t}{\sum_{s=1}^{K}q_s}\right)+\log(f(\boldsymbol{y}))+O(\sqrt{\epsilon})\right)d\boldsymbol{y}$$

$$=\sum_{s=1}^{K}\left(\frac{\dim(\mathcal{M}_s)}{2}\log\left(\frac{\pi\epsilon\sum_{t=1}^{K}q_t\beta_t}{q_s}\right)\right) \tag{163}$$

$$+\frac{\dim(\mathcal{M}_{K+1})}{2}\log\left(\frac{\pi\epsilon\sum_{t=1}^{K}q_t\beta_t}{\sum_{s=1}^{K}q_s}\right)-h(f)+O(\sqrt{\epsilon}).$$

$\square$

**Theorem 1.** Let $\{\boldsymbol{\omega}^{(k)}\}$ be a partitioning that satisfies the constraints in Problem 2. Then, there exists $\tilde{\epsilon}(\mathcal{M}, f) \leq 1$ such that for any $\epsilon \leq \tilde{\epsilon}$ and $p_s^{(k)} \in [\sqrt{\epsilon}, 1-(K-1)\sqrt{\epsilon}]$ we have

$$\min_{\{\tilde{\boldsymbol{W}}^{(k)}\}}\frac{1}{\epsilon N}\left(\tilde{G}\left(\left\{\tilde{\boldsymbol{W}}^{(k)}\right\}, \left\{\boldsymbol{\omega}^{(k)}\right\}, \{\boldsymbol{y}_i\}\right)+\epsilon\left(\sum_{i,j=1}^{N}\tilde{W}_{i,j}^{(k)}\log\left(\tilde{W}_{i,j}^{(k)}\right)\right)\right)+K\log(N) \tag{16}$$

$$\xrightarrow{N,D\to\infty}\sum_{k=1}^{K}\frac{\dim(\mathcal{M}_{K+1})}{2}\log\left(\frac{\sum_{s=1}^{K}p_s^{(k)}}{\sum_{t=1}^{K}p_t^{(k)}\beta_t}\right)+\sum_{k,s=1}^{K}\frac{\dim(\mathcal{M}_s)}{2}\log\left(\frac{p_s^{(k)}}{\sum_{t=1}^{K}p_t^{(k)}\beta_k}\right) \tag{17}$$

$$-K\sum_{s=1}^{K+1}\frac{dim(\mathcal{M}_s)}{2}\log\left(\pi\epsilon\right)+K\sum_{s=1}^{K+1}h_s(f_s)+O(\sqrt{\epsilon}),$$

where the convergence is in probability and $h_k(f_k) = -\int_{\boldsymbol{z}\in\mathcal{M}_k} f_k(\boldsymbol{z})\log f_k(\boldsymbol{z})d\boldsymbol{z}$ is the differential entropy of the density $f_k$ over $\mathcal{M}_k$.

*Proof of Theorem 1.*
We denote the entire latent data manifold by $\mathcal{M} = \mathcal{M}_1 \times \ldots \times \mathcal{M}_{K+1}$, and its distribution by $f$ defined by $f(\boldsymbol{x}^{(1)}, \ldots, \boldsymbol{x}^{(K+1)}) \equiv \prod_{k=1}^{K+1} f_k(\boldsymbol{x}^{(k)})$. As can be understood, this definition complies with the data definition of Section 2.
Based on Corollary 1, a set of affinity matrices $\{\boldsymbol{W}^{(k)}\}_{k=1}^{K}$ that minimize Eq. 16 is of the form:

$$W_{i,j}^{(k)} = \frac{A_{i,j}^{(k)}}{\sum_t A_{i,t}^{(k)}} \tag{164}$$

for all $k \in \{1, \ldots, K\}$ and $i, j \in \{1, \ldots, N\}$ and

$$A_{i,j}^{(k)} = \begin{cases} \exp\left(-\frac{\|\boldsymbol{y}_i-\boldsymbol{y}_j\|_{\tilde{\boldsymbol{\omega}}^{(k)}}^2}{\epsilon\cdot(1/D)\sum_d\tilde{\omega}_d^{(k)}}\right) & \text{if } i\neq j \\ 0 & \text{else} \end{cases}. \tag{165}$$

By plugging these affinity matrices in to Eq. 16 we get

$$\frac{1}{\epsilon N}\sum_{k=1}^{K}\sum_{i,j=1}^{N}\frac{A_{i,j}^{(k)}}{\sum_{t=1}^{n}A_{i,t}^{(k)}}\cdot\frac{\|\boldsymbol{y}_i-\boldsymbol{y}_j\|_{\boldsymbol{\omega}^{(k)}}^2}{\sum_{d=1}^{D}\omega_d^{(k)}} \tag{166}$$

$$+ \frac{1}{N} \sum_{k=1}^{K} \sum_{i,j=1}^{N} \frac{A_{i,j}^{(k)}}{\sum_{t=1}^{n} A_{i,t}^{(k)}} \log \left( \frac{A_{i,j}^{(k)}}{\sum_{t=1}^{n} A_{i,t}^{(k)}} \right) + K \log(N)$$

$$= -\frac{1}{N} \sum_{k=1}^{K} \sum_{i,j=1}^{N} \frac{A_{i,j}^{(k)}}{\sum_{t=1}^{n} A_{i,t}^{(k)}} \cdot \log(A_{i,j}^{(k)}) \tag{167}$$

$$+ \frac{1}{N} \sum_{k=1}^{K} \sum_{i=1}^{N} \left( \log(N) + \sum_{j=1}^{N} \frac{A_{i,j}^{(k)}}{\sum_{t=1}^{n} A_{i,t}^{(k)}} \log \left( \frac{A_{i,j}^{(k)}}{\sum_{t=1}^{N} A_{i,t}^{(k)}} \right) \right)$$

$$= \frac{1}{N} \sum_{k=1}^{K} \sum_{i=1}^{N} \left( \log(N) - \sum_{j=1}^{n} \frac{A_{i,j}^{(k)}}{\sum_{t=1}^{n} A_{i,t}^{(k)}} \log \left( \sum_{t=1}^{N} A_{i,t}^{(k)} \right) \right) \tag{168}$$

$$= \frac{1}{N} \sum_{k=1}^{K} \sum_{i=1}^{N} \left( \log(N) - \log \left( \sum_{t=1}^{n} A_{i,t}^{(k)} \right) \right) \tag{169}$$

$$= -\frac{1}{N} \sum_{k=1}^{K} \sum_{i=1}^{N} \left( \log \left( \frac{1}{N} \sum_{t=1}^{N} A_{i,t}^{(k)} \right) \right). \tag{170}$$

Now it will be easier to derive Eq. 17. For the asymptotic convergence in the next lines we employ Lemmas 5 to 7. Specifically, we use Lemma 7 that proves that there exists $\tilde{\epsilon}(\mathcal{M}, f)$ such that for any $\epsilon \leq \tilde{\epsilon}^2 / \sqrt{K}$ and any $p_s^{(k)} \in [\sqrt{\epsilon}, 1 - (K-1)\sqrt{\epsilon}]]$, where $s, k = 1, \dots, K$, we get

$$\frac{-1}{N} \sum_{k=1}^{K} \sum_{i=1}^{N} \log \frac{1}{n} \sum_{j=1}^{N} \exp \left( - \frac{-\|\boldsymbol{y}_i - \boldsymbol{y}_j\|_{\boldsymbol{\omega}^{(k)}}^2}{\epsilon \cdot (1/D) \sum_d \omega_d^{(k)}} \right) \tag{171}$$

$$\xrightarrow[p]{D \to \infty} \frac{-1}{N} \sum_{k=1}^{K} \sum_{i=1}^{N} \log \frac{1}{N} \sum_{j=1}^{N} \exp \left( - \frac{\sum_{s=1}^{K} \left\| \begin{bmatrix} \boldsymbol{x}_i^{(s)} \\ \boldsymbol{x}_i^{(K+1)} \end{bmatrix} - \begin{bmatrix} \boldsymbol{x}_j^{(s)} \\ \boldsymbol{x}_j^{(K+1)} \end{bmatrix} \right\|^2 p_s^{(k)}}{\epsilon \sum_{t=1}^{K} p_t^{(k)} \beta_t} \right) \tag{172}$$

$$= \frac{-1}{N} \sum_{k=1}^{K} \sum_{i=1}^{N} \log \frac{1}{N} \sum_{j=1}^{N} \exp \left( - \frac{\sum_{s=1}^{K} \left\| \boldsymbol{x}_i^{(s)} - \boldsymbol{x}_j^{(s)} \right\|^2 p_s^{(k)}}{\epsilon \sum_{t=1}^{K} p_t^{(k)} \beta_t} - \frac{\left\| \boldsymbol{x}_i^{(K+1)} - \boldsymbol{x}_j^{(K+1)} \right\|^2 \sum_{s=1}^{K} p_s^{(k)}}{\epsilon \sum_{t=1}^{K} p_t^{(k)} \beta_t} \right) \tag{173}$$

$$\xrightarrow[a.s.]{N \to \infty} - \sum_{k=1}^{K} \int_{\boldsymbol{x} \in \mathcal{M}} \log \int_{\boldsymbol{z} \in \mathcal{M}} \exp \left( - \frac{\sum_{s=1}^{K} \|\boldsymbol{x}^{(s)} - \boldsymbol{z}^{(s)}\|^2 p_s^{(k)}}{\epsilon \sum_{t=1}^{K} p_t^{(k)} \beta_t} - \frac{\left\| \boldsymbol{x}_i^{(K+1)} - \boldsymbol{z}^{(K+1)} \right\|^2 \sum_{s=1}^{K} p_s^{(k)}}{\epsilon \sum_{t=1}^{K} p_t^{(k)} \beta_t} \right) \tag{174}$$

$$f(\boldsymbol{z}) d\boldsymbol{z} f(\boldsymbol{x}) d\boldsymbol{x}$$

$$= \sum_{k=1}^{K} \frac{\dim(\mathcal{M}_{K+1})}{2} \log \left( \frac{\sum_{s=1}^{K} p_s^{(k)}}{\sum_{t=1}^{K} p_t^{(k)} \beta_t} \right) - K \frac{\dim(\mathcal{M}_{K+1})}{2} \log(\pi \epsilon) \tag{175}$$

$$+ \sum_{k,s=1}^{K} \frac{\dim(\mathcal{M}_s)}{2} \log \left( \frac{p_s^{(k)}}{\sum_{t=1}^{K} p_t^{(k)} \beta_t} \right) - K \sum_{s=1}^{K} \frac{\dim(\mathcal{M}_s)}{2} \log (\pi \epsilon)$$

$$+ K h(f) \log + O(\sqrt{\epsilon})$$

$\square$

**Lemma 8.** *Let $C_1, C_2, C_3 > 0$, and $\beta_1, \beta_2 \in (0,1)$ where $\beta_1 + \beta_2 = 1$. Define the function $f : (0,1)^2 \to \mathbb{R}$ by*

$$
\begin{aligned}
f(p_1, p_2) \quad = \quad & C_1 \log\left(\frac{p_1(1-p_1)}{(p_1\beta_1 + p_2\beta_2)(1 - p_1\beta_1 - p_2\beta_2)}\right) \\
& + C_2 \log\left(\frac{p_2(1-p_2)}{(p_1\beta_1 + p_2\beta_2)(1 - p_1\beta_1 - p_2\beta_2)}\right) \\
& + C_3 \log\left(\frac{(p_1 + p_2)(2 - p_1 - p_2)}{(p_1\beta_1 + p_2\beta_2)(1 - p_1\beta_1 - p_2\beta_2)}\right).
\end{aligned}
\tag{176}
$$

*Let $\alpha \geq max_{t \in \{0,1\}} 0.5 \cdot (4\beta_2(1-\beta_2))^{-(C_1 + C_2 + C_3)/(C_t)}$ be a neighborhood constant. Then, for any $\delta \in (0, min(\beta_1, \beta_2)/\alpha)$ the minimizers $p_1^*, p_2^*$ defined by*

$$
p_1^*, p_2^* = \underset{(p_1, p_2) \in [\delta, 1-\delta]^2}{argmin} f(p_1, p_2)
\tag{177}
$$

*should sastisfy $(p_1^*, p_2^*) \in L \times U$ or $(p_1^*, p_2^*) \in U \times L$. where $L = [\delta, \alpha\delta]$ and $U = [1 - \alpha\delta, 1 - \delta]$.*

*Proof.* We begin with going through the properties of $\rho : [0,1] \to \mathbb{R}$ defined by $\rho(p) = p(1-p)$ for any $p \in [0,1]$. The function is concave, maximized when $p = 1/2$ and is symmetric around it.

For any $(p_1, p_2) \in [\alpha\delta, 1 - \alpha\delta] \times [\delta, 1 - \delta]$ we begin by analyzing the value of the first two terms with respect to their value in $(\delta, 1 - \delta)$

$$
\sum_{i=1}^{2} C_i \log\left(\frac{p_i(1-p_i)}{(p_1\beta_1 + p_2\beta_2)(1 - p_1\beta_1 - p_2\beta_2)}\right)
\tag{178}
$$

$$
- \sum_{i=1}^{2} C_i \log\left(\frac{\delta(1-\delta)}{(\delta\beta_1 + (1-\delta)\beta_2)(1 - \delta\beta_1 - (1-\delta)\beta_2)}\right)
$$

$$
= \sum_{i=1}^{2} C_i \log\left(\frac{p_i(1-p_i)}{\delta(1-\delta)}\right)
\tag{179}
$$

$$
+ \sum_{i=1}^{2} C_i \log\left(\frac{(\delta\beta_1 + (1-\delta)\beta_2)(1 - \delta\beta_1 - (1-\delta)\beta_2)}{(p_1\beta_1 + p_2\beta_2)(1 - p_1\beta_1 - p_2\beta_2)}\right)
$$

$$
\geq C_1 \log\left(\frac{\alpha\delta(1-\alpha\delta)}{\delta(1-\delta)}\right) + C_2 \log\left(\frac{\delta(1-\delta)}{\delta(1-\delta)}\right) + \sum_{i=1}^{2} C_i \log\left(\frac{\beta_2(1-\beta_2)}{1/4}\right)
\tag{180}
$$

$$
\geq C_1 \log\left(\alpha(1-\alpha\delta)\right) + \sum_{i=1}^{2} C_i \log\left(4\beta_2(1-\beta_2)\right)
\tag{181}
$$

$$
\geq C_1 \log\left(\alpha/2\right) + \sum_{i=1}^{2} C_i \log\left(4\beta_2(1-\beta_2)\right)
\tag{182}
$$

based on the assumption on the properties of $\rho$, and that for $\beta = min(\beta_1, \beta_2)$ the identity $\delta(1 - \beta) + (1-\delta)\beta \in [\beta, 1/2]$ is sastified as

$$
\begin{aligned}
\delta(1-\beta) + (1-\delta)\beta \quad &= \quad \beta + \delta(1-2\beta) & \tag{183} \\
&\leq \quad \beta + \beta(1-2\beta) & \tag{184} \\
&\leq \quad 2\beta(1-\beta) & \tag{185} \\
&\leq \quad 1/2 & \tag{186} \\
\delta(1-\beta) + (1-\delta)\beta - \beta \quad &\geq \quad \delta(1-2\beta) & \tag{187} \\
&\geq \quad 0. & \tag{188}
\end{aligned}
$$

Using the same derivations mentioned above we can derive the difference between third terms -

$$C_3 \log\left(\frac{(p_1 + p_2)(2 - p_1 - p_2)}{(p_1\beta_1 + p_2\beta_2)(1 - p_1\beta_1 - p_2\beta_2)}\right) \qquad (189)$$

$$-C_3 \log\left(\frac{(\delta + 1 - \delta)(2 - \delta - (1 - \delta))}{(\delta\beta_1 + (1 - \delta)\beta_2)(1 - \delta\beta_1 - (1 - \delta)\beta_2)}\right)$$

$$\geq C_3 \log\left(\frac{p_1 + p_2}{p_1\beta_1 + p_2\beta_2}\right) + C_3 \log\left(\frac{2 - p_1 - p_2}{1 - p_1\beta_1 - p_2\beta_2}\right) - C_3 \log\left(\frac{1}{\beta_2(1 - \beta_2)}\right) \qquad (190)$$

$$\geq C_3 \log\left(\frac{p_1\beta_1 + p_2\beta_2}{p_1\beta_1 + p_2\beta_2}\right) + C_3 \log\left(\frac{1 - p_1 + 1 - p_2}{1 - min(p_1, p_2)}\right) + C_3 \log\left(4\beta_2(1 - \beta_2)\right) \qquad (191)$$

$$\geq C_3 \log\left(4\beta_2(1 - \beta_2)\right). \qquad (192)$$

By combining these two statements we get that $f(p_1, p_2) \geq f(\delta, 1 - \delta)$ -

$$f(p_1, p_2) - f(\delta, 1 - \delta) \qquad (193)$$

$$\geq C_1 \log\left(\alpha/2\right) + \sum_{i=1}^{3} C_i \log\left(4\beta_2(1 - \beta_2)\right) \qquad (194)$$

$$= C_1 \log\left(\alpha \cdot \frac{(4\beta_2(1 - \beta_2))^{\sum_{i=1}^{3} C_i/C_t}}{2}\right) \qquad (195)$$

$$\geq C_1 \log\left(max_{t \in \{0,1\}} \frac{2}{(4\beta_2(1 - \beta_2))^{\sum_{i=1}^{3} C_i/C_t}} \cdot \frac{(4\beta_2(1 - \beta_2))^{\sum_{i=1}^{3} C_i/C_1}}{2}\right) \qquad (196)$$

$$\geq C_1 \log\left(\frac{2}{(4\beta_2(1 - \beta_2))^{\sum_{i=1}^{3} C_i/C_1}} \cdot \frac{(4\beta_2(1 - \beta_2))^{\sum_{i=1}^{3} C_i/C_1}}{2}\right) \qquad (197)$$

$$\geq 0. \qquad (198)$$

The proof for $(p_1, p_2) \in [\delta, 1 - \delta] \times [\alpha\delta, 1 - \alpha\delta]$ can be derived from the above proof. Now we are left to show that this inequality stands for any $(p_1, p_2) \in [\delta, \alpha\delta]^2$. As before, we begin with the first two terms-

$$\sum_{i=1}^{2} C_i \log\left(\frac{p_i(1 - p_i)}{(p_1\beta_1 + p_2\beta_2)(1 - p_1\beta_1 - p_2\beta_2)}\right) \qquad (199)$$

$$-\sum_{i=1}^{2} C_i \log\left(\frac{\delta(1 - \delta)}{(\delta\beta_1 + (1 - \delta)\beta_2)(1 - \delta\beta_1 - (1 - \delta)\beta_2)}\right)$$

$$= \sum_{i=1}^{2} C_i \log\left(\frac{p_i(1 - p_i)}{\delta(1 - \delta)}\right) + \sum_{i=1}^{2} C_i \log\left(\frac{(\delta\beta_1 + (1 - \delta)\beta_2)(1 - \delta\beta_1 - (1 - \delta)\beta_2)}{(p_1\beta_1 + p_2\beta_2)(1 - p_1\beta_1 - p_2\beta_2)}\right) \qquad (200)$$

$$\geq \sum_{i=1}^{2} C_i \log\left(\frac{\delta(1 - \delta)}{\delta(1 - \delta)}\right) + \sum_{i=1}^{2} C_i \log\left(\frac{(\delta\beta_1 + (1 - \delta)\beta_2)(1 - \delta\beta_1 - (1 - \delta)\beta_2)}{(\alpha\delta\beta_1 + \alpha\delta\beta_2)(1 - \alpha\delta\beta_1 - \alpha\delta\beta_2)}\right) \qquad (201)$$

$$\geq \sum_{i=1}^{2} C_i \log\left(\frac{\beta_2(1 - \beta_2)}{\alpha\delta(1 - \alpha\delta)}\right) \qquad (202)$$

$$\geq 0 \qquad (203)$$

based on the assumption on the properties of $g$, and that $\delta(1 - \beta) + (1 - \delta)\beta \in [\beta, 1/2]$. $\qquad \square$

**Theorem 2.** Define $f : (0, 1)^2 \to \mathbb{R}$ by

$$f(p_1, p_2) = \sum_{k=1}^{K} \frac{\dim(\mathcal{M}_{K+1})}{2} \log\left(\frac{\sum_{s=1}^{K} p_s^{(k)}}{\sum_{t=1}^{K} p_t^{(k)}\beta_t}\right) + \sum_{k,s=1}^{K} \frac{\dim(\mathcal{M}_s)}{2} \log\left(\frac{p_s^{(k)}}{\sum_{t=1}^{K} p_t^{(k)}\beta_k}\right), \qquad (204)$$

where $p_1^{(1)} = p_1$ and $p_2^{(1)} = p_2$ and therefore $p_2^{(1)} = 1 - p_1^{(1)}, p_2^{(2)} = 1 - p_2^{(2)}$.
The minimizers $p_1^*, p_2^* = \lim_{\epsilon \to 0} \text{argmin}_{p_1, p_2 \in [\sqrt{\epsilon}, 1 - \sqrt{\epsilon}]^2} f(p_1, p_2)$ exist and satisfy $(p_1^*, p_2^*) \in \{(0,1), (1,0)\}$.

*Proof of Theorem 2.* Based on Lemma 8, we can see that $\alpha$ does not depends on $\delta$. Meaning that if we take a sequence of $\delta$s that converges to 0, the minimizers should converge to $(0,1)$ or $(1,0)$. Now, by defining $\delta = \sqrt{\epsilon}$ we attain the theorem's statement. □

### E.3 Proofs of Appendix C

**Lemma 9.** *Let $\{\boldsymbol{W}^{(k)}\} \subset [0,1]^{n \times n}$ be affinity matrices under the constraints in Problem 3. Define a partitioning solution $\{\boldsymbol{\omega}^{(k)}\} \subset [0,1]^D$ by*

$$\omega_d^{(k)} = \frac{\exp\left(-\sum_{i,j=1}^n W^{(k)}\left((\boldsymbol{y}_i)_d - (\boldsymbol{y}_j)_d\right)^2 / \delta\right)}{\sum_{\tilde{k}=1}^K \exp\left(-\sum_{i,j=1}^n W^{(\tilde{k})}\left((\boldsymbol{y}_i)_d - (\boldsymbol{y}_j)_d\right)^2 / \delta\right)} \qquad k = 1, \ldots, K. \quad (205)$$

*Then, a minimizer of the optimization problem suggested in Problem 3 among partitioning solutions that satisfy its constraints is*

$$\boldsymbol{\omega}^{(k)*} = \underset{\{\tilde{\boldsymbol{\omega}}^{(k)}\}}{argmin} \, G_{reg}(\delta, \{\boldsymbol{W}^{(k)}\}_{k=1}^K, \{\tilde{\boldsymbol{\omega}}^{(k)}\}_{k=1}^K, \{\boldsymbol{y}_i\}_{i=1}^N) \qquad (206)$$

*Proof.* The minimization problem with respect to partitioning solutions can be written by

$$\underset{\{\tilde{\boldsymbol{\omega}}^{(k)}\}}{argmin} \sum_{d=1}^D \sum_{k=1}^K \tilde{\omega}_d^{(k)} \left(\sum_{i=1}^n \sum_{j=1}^n W_{i,j}^{(k)}((\boldsymbol{y}_i)_d - (\boldsymbol{y}_j)_d)^2\right) + \delta \sum_{d=1}^D \sum_{k=1}^K \tilde{\omega}_d^{(k)} \log\left(\tilde{\omega}_d^{(k)}\right), \quad (207)$$

where $\sum_{k=1}^K \omega_d^{(k)} = 1$ for any $d = 1, \ldots, D$. As the constraints are $\sum_{k=1}^K \omega_d^{(k)} = 1$ for all $d \in \{1, \ldots, D\}$, the above problem can be decomposed into $D$ independent problems. Meaning that for and $d \in \{1, \ldots, D\}$ we need to solve

$$\underset{\{\tilde{\boldsymbol{\omega}}_d^{(k)}\}}{argmin} \sum_{k=1}^K \tilde{\omega}_d^{(k)} \left(\sum_{i=1}^n \sum_{j=1}^n W_{i,j}^{(k)}((\boldsymbol{y}_i)_d - (\boldsymbol{y}_j)_d)^2\right) + \delta \sum_{k=1}^K \tilde{\omega}_d^{(k)} \log\left(\tilde{\omega}_d^{(k)}\right). \quad (208)$$

Then by showing that the new problem includes affine equality constraints and a convex objective function we can use the KKT conditions for convex problems (See *KKT conditions for convex problems* within Sec. 5.5.3 in Boyd & Vandenberghe (2004)) to find an optimal solution. Specifically, it states that any primal and dual sets parameters that satisfy the KKT conditions are the primal and dual optimal solution with zero duality gap.

We begin by showing that the new suggested problem satisfies the conditions of the KKT conditions for convex problems. The constraint the sums the partitions up to one in every dimension is indeed linear. Now we will show that the objective function is linear by using second order condition that implies the convexity of the function (see Sec. 3.1.4 in Boyd & Vandenberghe (2004)). This is done by showing the that the Hessian of the function is positive semi-definite, which is demonstrated here by proving it is diagonal with positive values

$$\frac{d^2}{d^2 \tilde{\omega}_d^{(k)}} \left(\sum_{k=1}^K \tilde{\omega}_d^{(k)} \left(\sum_{i=1}^n \sum_{j=1}^n W_{i,j}^{(k)}((\boldsymbol{y}_i)_d - (\boldsymbol{y}_j)_d)^2\right) + \delta \sum_{k=1}^K \tilde{\omega}_d^{(k)} \log\left(\tilde{\omega}_d^{(k)}\right)\right) \quad (209)$$

$$= \frac{d}{d\tilde{\omega}_d^{(k)}} \left(\left(\sum_{i=1}^n \sum_{j=1}^n W_{i,j}^{(k)}((\boldsymbol{y}_i)_d - (\boldsymbol{y}_j)_d)^2\right) + \delta(1 + log(\tilde{\omega}_d^{(k)}))\right) \quad (210)$$

$$= \frac{\delta}{\tilde{\omega}_d^{(k)}} \quad (211)$$

$$\geq 1. \quad (212)$$

$$\frac{d}{d\tilde{\omega}_d^{(s)}} \frac{d}{d\tilde{\omega}_d^{(k)}} \left(\left(\sum_{i=1}^n \sum_{j=1}^n W_{i,j}^{(k)}((\boldsymbol{y}_i)_d - (\boldsymbol{y}_j)_d)^2\right) + \delta \sum_{k=1}^K \tilde{\omega}_d^{(k)} \log\left(\tilde{\omega}_d^{(k)}\right)\right) \quad (213)$$

$$= \frac{d}{d\tilde{\omega}_d^{(s)}} \left(\left(\sum_{i=1}^n \sum_{j=1}^n W_{i,j}^{(k)}((\boldsymbol{y}_i)_d - (\boldsymbol{y}_j)_d)^2\right) + \delta(1 + log(\tilde{\omega}_d^{(k)}))\right) \quad (214)$$

$$= 0 \quad (215)$$

for all $s, k \in \{1, \ldots, K\}$ such that $s \neq k$.

We define the Lagrangian of the minimization problem over $\{\tilde{\boldsymbol{\omega}}_d^{(k)}\}$ for all $d \in \{1, \ldots, D\}$ by

$$\tilde{L}_d(\boldsymbol{W}^{(k)}, \tilde{\omega}_d^{(k)}) \equiv \sum_{k=1}^{K} \tilde{\omega}_d^{(k)} \left( \sum_{i=1}^{n} \sum_{j=1}^{n} W_{i,j}^{(k)} ((\boldsymbol{y}_i)_d - (\boldsymbol{y}_j)_d)^2 \right) \tag{216}$$

$$+\delta \sum_{k=1}^{K} \tilde{\omega}_d^{(k)} \log \left( \tilde{\omega}_d^{(k)} \right) \tag{217}$$

$$+\mu_d (\sum_{\tilde{k}=1}^{K} \tilde{\omega}_d^{(\tilde{k})} - 1) \tag{218}$$

where $\mu_d \in \mathbb{R}$.

Next, we will find solutions $\{\omega^{(k)}\}$ that attain the KKT conditions, as they are sufficient for optimality. We begin with the stationary condition

$$0 = \frac{d\tilde{L}_d}{d\omega_d^{(k)}} \tag{219}$$

$$= \sum_{i=1}^{n} \sum_{j=1}^{n} W_{i,j}^{(k)} ((\boldsymbol{y}_i)_d - (\boldsymbol{y}_j)_d)^2 + \delta(1 + \log(\omega_d^{(k)})) + \mu_d \tag{220}$$

$$\omega_d^{(k)} = exp \left( -\frac{\sum_{i,j=1}^{n} W_{i,j}^{(k)} ((\boldsymbol{y}_i)_d - (\boldsymbol{y}_j)_d)^2 + \delta + \mu_d}{\delta} \right). \tag{221}$$

Now, the KKT condition primal feasibility condition on the equality constraints induces

$$1 = \sum_{k=1}^{K} \omega_d^{(k)} \tag{222}$$

$$= \sum_{k=1}^{K} exp \left( -\frac{\sum_{i=1}^{n} \sum_{j=1}^{n} W_{i,j}^{(k)} ((\boldsymbol{y}_i)_d - (\boldsymbol{y}_j)_d)^2 + \delta + \mu_d}{\delta} \right) \tag{223}$$

$$exp \left( \frac{\mu_d}{\delta} \right) = \sum_{k=1}^{K} exp \left( -\frac{\sum_{i=1}^{n} \sum_{j=1}^{n} W_{i,j}^{(k)} ((\boldsymbol{y}_i)_d - (\boldsymbol{y}_j)_d)^2 + \delta}{\delta} \right) \tag{224}$$

$$\mu_d = \delta \cdot \log \left( \sum_{k=1}^{K} exp \left( -\frac{\sum_{i=1}^{n} \sum_{j=1}^{n} W_{i,j}^{(k)} ((\boldsymbol{y}_i)_d - (\boldsymbol{y}_j)_d)^2 + \delta}{\delta} \right) \right). \tag{225}$$

By pushing it back into our explicit stationary condition we get

$$\omega_d^{(k)} = \frac{exp \left( -\frac{\sum_{i=1}^{n} \sum_{j=1}^{n} W_{i,j}^{(k)} ((\boldsymbol{y}_i)_d - (\boldsymbol{y}_j)_d)^2 + \delta}{\delta} \right)}{\sum_{k=1}^{K} exp \left( -\frac{\sum_{i=1}^{n} \sum_{j=1}^{n} W_{i,j}^{(k)} ((\boldsymbol{y}_i)_d - (\boldsymbol{y}_j)_d)^2 + \delta}{\delta} \right)} \tag{226}$$

$$= \frac{exp \left( -\frac{\sum_{i=1}^{n} \sum_{j=1}^{n} W_{i,j}^{(k)} ((\boldsymbol{y}_i)_d - (\boldsymbol{y}_j)_d)^2}{\delta} \right)}{\sum_{k=1}^{K} exp \left( -\frac{\sum_{i=1}^{n} \sum_{j=1}^{n} W_{i,j}^{(k)} ((\boldsymbol{y}_i)_d - (\boldsymbol{y}_j)_d)^2}{\delta} \right)}. \tag{227}$$

This concludes the proof. $\qquad \square$

**Proposition 4.** Let $\delta \geq 0$, $\{\boldsymbol{\omega}^{(k)}\}_{k=1}^{K} \subset [0, 1]^D$ be a partitioning weights and $\{\boldsymbol{W}^{(k)}\}_{k=1}^{K}, \subset [0, 1]^{N \times N}$ be affinity matrices that satisfy the constraints of Problem 3.

Define $\{\boldsymbol{W}^{(k)*}\} \subset [0,1]^{N \times N}$ as in Eq. 10 and $\{\boldsymbol{\omega}^{(k)*}\}_{k=1}^{K}$ by

$$\omega_d^{(k)*} = \exp\left(-\frac{\sum_{i,j} W_{i,j}^{(k)}\left((\boldsymbol{y}_i)_d - (\boldsymbol{y}_j)_d\right)^2}{\delta}\right) \Big/ \sum_{s=1}^{K} \exp\left(-\frac{\sum_{i,j} W_{i,j}^{(s)}\left((\boldsymbol{y}_i)_d - (\boldsymbol{y}_j)_d\right)^2}{\delta}\right). \tag{23}$$

Then, a set of parameters that minimize the objective function and are with the domain of Problem 3 are

$$\{\boldsymbol{W}^{(k)*}\} = \underset{\{\tilde{\boldsymbol{W}}^{(k)}\}}{argmin}\, G_{reg}(\delta, \{\tilde{\boldsymbol{W}}^{(k)}\}_{k=1}^{K}, \{\boldsymbol{\omega}^{(k)}\}_{k=1}^{K}, \{\boldsymbol{y}_i\}_{i=1}^{N}), \tag{24}$$

$$\{\boldsymbol{\omega}^{(k)*}\} = \underset{\{\tilde{\boldsymbol{\omega}}^{(k)}\}}{argmin}\, G_{reg}(\delta, \{\boldsymbol{W}^{(k)}\}_{k=1}^{K}, \{\tilde{\boldsymbol{\omega}}^{(k)}\}_{k=1}^{K}, \{\boldsymbol{y}_i\}_{i=1}^{N}). \tag{25}$$

*Proof of Proposition 4.* The proposition aims to characterize optimal parameters of Problem 3. We begin by defining two sub-problems that are related to it, each focusing on minimizing one set of parameters while keeping the other set fixed:

$$\{\boldsymbol{W}^{(k)*}\} = \underset{\{\tilde{\boldsymbol{W}}^{(k)}\}}{\arg\min}\, G_{reg}(\delta, \{\tilde{\boldsymbol{W}}^{(k)}\}_{k=1}^{K}, \{\boldsymbol{\omega}^{(k)}\}_{k=1}^{K}, \{\boldsymbol{y}_i\}_{i=1}^{N}), \tag{228}$$

$$\{\boldsymbol{\omega}^{(k)*}\} = \underset{\{\tilde{\boldsymbol{\omega}}^{(k)}\}}{\arg\min}\, G_{reg}(\delta, \{\boldsymbol{W}^{(k)}\}_{k=1}^{K}, \{\tilde{\boldsymbol{\omega}}^{(k)}\}_{k=1}^{K}, \{\boldsymbol{y}_i\}_{i=1}^{N}). \tag{229}$$

where the parameters are limited to the constraints stated in Problem 3. Interestingly, the suggested optimization problem in Eq. 229 can be rewritten by

$$\{\boldsymbol{W}^{(k)*}\} = \underset{\{\tilde{\boldsymbol{W}}^{(k)}\}}{\arg\min}\, G_{reg}(\delta, \{\tilde{\boldsymbol{W}}^{(k)}\}_{k=1}^{K}, \{\boldsymbol{\omega}^{(k)}\}_{k=1}^{K}, \{\boldsymbol{y}_i\}_{i=1}^{N} \tag{230}$$

$$= \underset{\{\tilde{\boldsymbol{W}}^{(k)}\}}{\arg\min}\, G(\{\tilde{\boldsymbol{W}}^{(k)}\}, \{\tilde{\boldsymbol{\omega}}^{(k)}\}, \{\boldsymbol{y}_i\}_{i=1}^{N}) + \delta\left(D\log(K) + \sum_{d=1}^{D}\sum_{k=1}^{K} \tilde{\omega}_d^{(k)}\log(\tilde{\omega}_d^{(k)})\right) \tag{231}$$

$$= \underset{\{\tilde{\boldsymbol{W}}^{(k)}\}}{\arg\min}\, G(\{\tilde{\boldsymbol{W}}^{(k)}\}, \{\tilde{\boldsymbol{\omega}}^{(k)}\}, \{\boldsymbol{y}_i\}_{i=1}^{N}). \tag{232}$$

These two sub-problems are considered in Lemmas 3 and 9. Specifically, in Lemma 3 the optimal graph matrices are derived in the form of

$$W_{i,j}^{(k)*} = \begin{cases} \dfrac{\exp\left(-\dfrac{\|\boldsymbol{y}_i - \boldsymbol{y}_j\|_{\tilde{\boldsymbol{\omega}}^{(k)}}^2}{\epsilon_{k,i}}\right)}{\sum_{t=1}^{n} \exp\left(-\dfrac{\|\boldsymbol{y}_i - \boldsymbol{y}_t\|_{\tilde{\boldsymbol{\omega}}^{(k)}}^2}{\epsilon_{k,i}}\right)} & \text{for } j \neq i \\[2em] 0 & \text{else} \end{cases}, \tag{233}$$

$$\tag{234}$$

for $i, j = 1, \ldots, N$ and $k = 1, \ldots, K$, and $\epsilon_{k,i}$ satisfy

$$\sum_{j=1}^{N} W_{i,j}^{(k)*} \log W_{i,j}^{(k)*} = -\log(\alpha). \tag{235}$$

On the other hand, in Lemma 9 an optimal partitioning parameters are derived in the form of

$$\omega_d^{(k)} = \frac{\exp\left(-\sum_{i,j=1}^{n} W^{(k)}\left((\boldsymbol{y}_i)_d - (\boldsymbol{y}_j)_d\right)^2/\delta\right)}{\sum_{\tilde{k}=1}^{K} \exp\left(-\sum_{i,j=1}^{n} W^{(\tilde{k})}\left((\boldsymbol{y}_i)_d - (\boldsymbol{y}_j)_d\right)^2/\delta\right)} \tag{236}$$

for $d = 1, \ldots, D$ and $k = 1, \ldots, K$.

Therefore there exists parameters of this form that minimizes Problem 3. $\square$

**Proposition 5.** Let $\{\overline{\boldsymbol{\omega}}^{(k)}\}_{k=1}^K \subset [0,1]^D$ be a uniform partitioning, i.e. $\overline{\omega}_d^{(k)} = 1/K$, and the corresponding affinity matrices $\{\overline{\boldsymbol{W}}^{(k)}\}_{k=1}^K$ defined as in Proposition 4 based on these partitions. Let $\{\boldsymbol{\omega}^{(k)}\}_{k=1}^K \subset \{0,1\}^D$ and $\{\boldsymbol{W}^{(k)}\}_{k=1}^K$ be the optimal partitioning solution as discussed in Proposition 3.

Define

$$\delta_{init} \equiv \frac{G(\{\overline{\boldsymbol{W}}^{(k)}\}_{k=1}^K, \{\overline{\boldsymbol{\omega}}^{(k)}\}_{k=1}^K, \{\boldsymbol{y}_i\}_{i=1}^N)}{D \cdot \log(K)}. \tag{26}$$

Then,

$$G_{reg}(\delta_{init}, \{\overline{\boldsymbol{W}}^{(k)}\}_{k=1}^K, \{\overline{\boldsymbol{\omega}}^{(k)}\}_{k=1}^K, \{\boldsymbol{y}_i\}_{i=1}^N) \leq G_{reg}(\delta_{init}, \{\boldsymbol{W}^{(k)}\}_{k=1}^K, \{\boldsymbol{\omega}^{(k)}\}_{k=1}^K, \{\boldsymbol{y}_i\}_{i=1}^N) \tag{27}$$

*Proof of Proposition 5.* We can see that the inequality stands based on the following derivation-

$$G_{reg}(\delta_{init}, \{\overline{\boldsymbol{W}}^{(k)}\}_{k=1}^K, \{\overline{\boldsymbol{\omega}}^{(k)}\}_{k=1}^K, \{\boldsymbol{y}_i\}_{i=1}^N) \tag{237}$$

$$= G(\{\overline{\boldsymbol{W}}^{(k)}\}_{k=1}^K, \{\overline{\boldsymbol{\omega}}^{(k)}\}_{k=1}^K, \{\boldsymbol{y}_i\}_{i=1}^N) + \delta_{init}\left(D\log(K) + \sum_{d=1}^D \sum_{k=1}^K \overline{\omega}_d^{(k)} \log(\overline{\omega}_d^{(k)})\right) \tag{238}$$

$$= G(\{\overline{\boldsymbol{W}}^{(k)}\}_{k=1}^K, \{\overline{\boldsymbol{\omega}}^{(k)}\}_{k=1}^K, \{\boldsymbol{y}_i\}_{i=1}^N) \tag{239}$$

$$= \delta_{init}(D \cdot \log(K)) \tag{240}$$

$$\leq G(\{\boldsymbol{W}^{(k)}\}_{k=1}^K, \{\boldsymbol{\omega}^{(k)}\}_{k=1}^K, \{\boldsymbol{y}_i\}_{i=1}^N) + \delta_{init}(D \cdot \log(K)) \tag{241}$$

$$= G(\{\boldsymbol{W}^{(k)}\}_{k=1}^K, \{\boldsymbol{\omega}^{(k)}\}_{k=1}^K, \{\boldsymbol{y}_i\}_{i=1}^N) + \delta_{init}\left(D\log(K) + \sum_{d=1}^D \sum_{k=1}^K \omega_d^{(k)} \log(\omega_d^{(k)})\right) \tag{242}$$

$$= G_{reg}(\delta_{init}, \{\boldsymbol{W}^{(k)}\}_{k=1}^K, \{\boldsymbol{\omega}^{(k)}\}_{k=1}^K, \{\boldsymbol{y}_i\}_{i=1}^N), \tag{243}$$

where we use the fact that the function $g$ is non-negative, and that negative entropies satisfy $\sum_{k=1}^K \omega_d^{(k)} \log \omega_d^{(k)} = 0$ and $\sum_{k=1}^K \overline{\omega}_d^{(k)} \log \overline{\omega}_d^{(k)} = -\log(K)$ for all $d \in \{1, \ldots, D\}$. $\qquad\square$

# F  ADDITIONAL PROOFS

**Proposition 6.** Let the data consist of $N$ data points in $\mathbb{R}^D$. Suppose the data is given in the form of a singular value decomposition (SVD) approximation of rank $S \ll N, D$. Then, the computational complexity of the updates suggested in Proposition 4 is $O(K(S^2 N^2 + S^2 D))$.

*Proof.* Let $\boldsymbol{Y} \in \mathbb{R}^{N \times D}$ represent the data matrix, with the points embedded as rows. The SVD approximation is given by $\boldsymbol{Y} = \boldsymbol{U}\boldsymbol{E}\boldsymbol{V}^T$, where $\boldsymbol{U} \in \mathbb{R}^{N \times S}$ and $\boldsymbol{V} \in \mathbb{R}^{D \times S}$ are the left and right singular vector matrices, respectively, and $\boldsymbol{E} \in \mathbb{R}^{S \times S}$ is the diagonal matrix of the leading singular values.

The update rule for the feature partitioning weights, which is described in Eq. 23, includes computation of

$$\sum_{i,j=1}^N W_{i,j}^{(k)}((\boldsymbol{y}_i)_d - (\boldsymbol{y}_j)_d)^2 \tag{244}$$

for every $d \in \{1, \ldots, D\}$ and $k \in \{1, \ldots, K\}$. The other operations used within this update rule includes taking the exponential of this value and normalization it across the different partitions which result in a computational complexity of $O(DK)$. In the next lines we will rewrite the equation above and compute the computational complexity associated with it.

We begin with rewriting the Eq. 244 by

$$\sum_{i,j=1}^{N} W_{i,j}^{(k)}((\boldsymbol{y}_i)_d - (\boldsymbol{y}_j)_d)^2 \tag{245}$$

$$= \sum_{j=1}^{N}(\sum_{i=1}^{N} W_{i,j}^{(k)})(\boldsymbol{y}_j)_d^2 + \sum_{i=1}^{N}(\sum_{j=1}^{N} W_{i,j}^{(k)})(\boldsymbol{y}_i)_d^2 - 2\sum_{i,j=1}^{N} W_{i,j}^{(k)}(\boldsymbol{y}_i)_d(\boldsymbol{y}_j)_d \tag{246}$$

$$= \sum_{j=1}^{N}(\sum_{i=1}^{N} W_{i,j}^{(k)} + W_{j,i}^{(k)})(\boldsymbol{y}_j)_d^2 - \sum_{i,j=1}^{N}(W_{i,j}^{(k)} + W_{j,i}^{(k)})(\boldsymbol{y}_i)_d(\boldsymbol{y}_j)_d \tag{247}$$

$$= (\boldsymbol{Y}^T(diag(\boldsymbol{W}^{(k)}\boldsymbol{1} + (\boldsymbol{W}^{(k)})^T\boldsymbol{1}))\boldsymbol{Y})_{d,d} - (\boldsymbol{Y}^T(\boldsymbol{W}^{(k)} + (\boldsymbol{W}^{(k)})^T)\boldsymbol{Y})_{d,d} \tag{248}$$

$$= (\boldsymbol{Y}^T\boldsymbol{L}\boldsymbol{Y})_{d,d} \tag{249}$$

where $\boldsymbol{L}^{(k)} \in \mathbb{R}^{N \times N}$ is defined by $\boldsymbol{L} = diag((\boldsymbol{W}^{(k)} + (\boldsymbol{W}^{(k)})^T)\boldsymbol{1}) - \boldsymbol{W}^{(k)} - (\boldsymbol{W}^{(k)})^T$ for any $k \in \{1, \ldots, K\}$, the vector $\boldsymbol{1} \in \mathbb{R}^N$ is an all ones vector, and $diag : \mathbb{R}^N \to \mathbb{R}^{N \times N}$ generates a diagonal matrix from a given vector. Next we are going to use the SVD decomposition within this equation:

$$(\boldsymbol{V}\boldsymbol{E}\boldsymbol{U}^T\boldsymbol{L}^{(k)}\boldsymbol{U}\boldsymbol{E}\boldsymbol{V}^T)_{d,d}. \tag{250}$$

The computational complexity of $\boldsymbol{A}^{(k)} \equiv \boldsymbol{U}^T\boldsymbol{L}^{(k)}\boldsymbol{U} \in \mathbb{R}^{S \times S}$ is $O(SN^2 + S^2N)$ as $\boldsymbol{U} \in \mathbb{R}^{N \times S}$ and $\boldsymbol{L}^{(k)} \in \mathbb{R}^{N \times N}$. The computational complexity of $\boldsymbol{B}^{(k)} \equiv \boldsymbol{E}\boldsymbol{A}^{(k)}\boldsymbol{E} \in \mathbb{R}^{S \times S}$ is $O(S^2)$ as $\boldsymbol{E}$ is a diagonal matrix. Therefore we are left with computing

$$(\boldsymbol{V}\boldsymbol{B}^{(k)}\boldsymbol{V}^T)_{d,d}. \tag{251}$$

The computational complexity of each such element is $O(S^2)$, and the computational complexity for all $d \in \{1, \ldots, D\}$ is $O(S^2D)$. Therefore the computational complexity of this step for a single $k \in \{1, \ldots, K\}$ is $O(SN^2 + S^2D)$, and over all $k$s $O(K(SN^2 + S^2D))$.

Now, we are left with deriving the computational complexity of the graph weights. The update rule for the graph weights that is described in Eq. 14 contains the computation of

$$\|\boldsymbol{y}_i - \boldsymbol{y}_j\|_{\boldsymbol{\omega}^{(k)}}^2 \tag{252}$$

for all $i, j \in \{1, \ldots, N\}$ and $k \in \{1, \ldots, K\}$. The other operations used within this update rule includes taking its exponential values and normalizing its values based on the values across $j \in \{1, \ldots, N\}$ for any $i \in \{1, \ldots, N\}$ and $k \in \{1, \ldots, K\}$. The latter operations has has a computational complexity of $O(N^2K)$, and now we can rewrite Eq. 252 and derive its computational complexity.

Eq. 252 can be rewritten by

$$\|\boldsymbol{y}_i - \boldsymbol{y}_j\|_{\boldsymbol{\omega}^{(k)}}^2 \tag{253}$$

$$= (\boldsymbol{Y}diag(\boldsymbol{\omega}^{(k)})\boldsymbol{Y}^T)_{i,i} + (\boldsymbol{Y}diag(\boldsymbol{\omega}^{(k)})\boldsymbol{Y}^T)_{j,j} - 2(\boldsymbol{Y}diag(\boldsymbol{\omega}^{(k)})\boldsymbol{Y}^T)_{i,j} \tag{254}$$

$$= (\boldsymbol{U}\boldsymbol{E}\boldsymbol{V}^T diag(\boldsymbol{\omega}^{(k)})\boldsymbol{V}\boldsymbol{E}\boldsymbol{U}^T)_{i,i} + (\boldsymbol{U}\boldsymbol{E}\boldsymbol{V}^T diag(\boldsymbol{\omega}^{(k)})\boldsymbol{V}\boldsymbol{E}\boldsymbol{U}^T)_{j,j} \tag{255}$$

$$-2(\boldsymbol{U}\boldsymbol{E}\boldsymbol{V}^T diag(\boldsymbol{\omega}^{(k)})\boldsymbol{V}\boldsymbol{E}\boldsymbol{U}^T)_{i,j} \tag{256}$$

for all $i, j \in \{1, \ldots, N\}$ and $k \in \{1, \ldots, K\}$, where $diag : \mathbb{R}^D \to \mathbb{R}^{D \times D}$ generates a diagonal matrix from a given vector. The computational complexity of $\boldsymbol{C}^{(k)} = \boldsymbol{V}^T diag(\boldsymbol{\omega}^{(k)})\boldsymbol{V} \in \mathbb{R}^{S \times S}$ is $O(S^2D)$, and of $\boldsymbol{F}^{(k)} = \boldsymbol{E}\boldsymbol{C}^{(k)}\boldsymbol{E} \in \mathbb{R}^{S \times S}$ is $O(S^2)$ as $\boldsymbol{E}$ is a diagonal matrix. Finally, the computational complexity of $(\boldsymbol{V}^T\boldsymbol{F}^{(k)}\boldsymbol{V})_{i,j}$ is $O(S^2)$, and the computational complexity of deriving it for all entrees is $O(S^2N^2)$. Therefore the total computational complexity for this step including over all partitions $k \in \{1, \ldots, K\}$ results in $O(K(S^2N^2 + KS^2D))$.

Now, by combining the computational complexities for both parameter updates results in $O(K(SN^2 + S^2D) + K(S^2N^2 + KS^2D)) = O(S^2N^2 + KS^2D)$.

$\square$

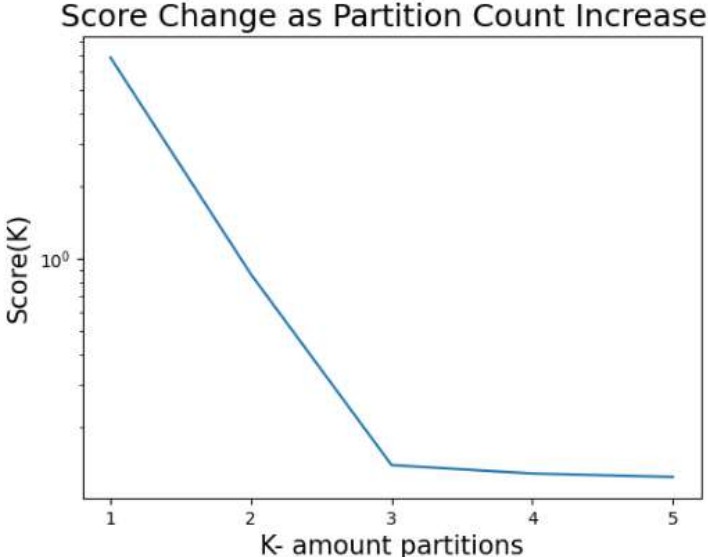

Figure 11: The score (Eq. 8) of our algorithm when using an increasing number of partitions ($K$) on the dataset with the three rotating figurines, used in Section 3.2.

## G   CHOOSING THE NUMBER OF PARTITIONS ($K$) AND VALIDATING THE DATA SETUP ASSUMPTION

The selection of the number of partitions ($K$) is a practical consideration that directly affects the outcome of our algorithm and therefore the visualizations generated based on it. A sub-optimal choice of $K$ can result in visualizations that are either overly complex or redundant. In this section, we discuss how to determine $K$ using our algorithm and examine the implications of selecting a sub-optimal number of partitions. Finally, we propose a procedure to verify whether the data is indeed composed of subsets of features that are partially independent as our approach assumes.

In order to determine $K$ we propose to inspect the behavior of the smoothness score from our approach (Eq. 8) as a function of the number of partitions $K$. As long as $K$ increases but remains lower than the true number of partitions, we expect to see a rapid decay of the score. Then, when $K$ reaches the true number of partitions, we expect to see an 'elbow', after which the score saturates or decays very slowly. This phenomenon is reminiscent of analogous ones when manually selecting the number of clusters in k-means or the number of components in PCA. We demonstrate this behavior of the smoothness score in Figure 11.

Next, we discuss the expected impact of over-selecting or under-selecting $K$ on the resulting representation and embedding. If the number of partitions is lower than the number of true partitions in the data, we expect the partitions to separate the features into super-groups that correspond to the most distinct geometric structures, even if some of these groups could be further subdivided. In this case, each group of features describes a geometry that is simpler than that of the original data, i.e., it has a lower intrinsic dimension. Hence, the outcome of our procedure in this case still improves the ability to embed and visualize the data in a low-dimensional space, albeit sub-optimally – where the embedding dimension may need to be higher than it would be if the optimal choice of $K$ was used.

If the number of partitions is slightly larger than the true number of partitions in the data, we expect that one or more of the true partitions will be arbitrarily subdivided. This will introduce redundancy into the representation, where some feature subsets will exhibit similar geometric structures. The user should take into account this possibility and inspect the resulting embeddings for potential redundancy. Nonetheless, the partitioning is still beneficial in this case, since the data for each partition can be easily embedded and visualized in a low-dimensional space. If the number of partitions is grossly overestimated, then in addition to redundancy, some of the feature subsets may not reliably represent the underlying geometry of any of the true feature subsets.

When the data's assumptions are not satisfied, we expect that applying our algorithm to partition the features into $K$ groups will result in partitions with similar or nearly identical underlying structures. Consequently, the graphs that correspond to the partitions will also exhibit similar characteristics, leading to embeddings (such as tSNE) that closely resemble each other.

To determine if the features can be partitioned into meaningful subsets, we propose a type of a permutation test. Specifically, we propose to compare the smoothness score produced by our method from the original data to the analogous score obtained from manipulated versions of the data. The manipulated versions are obtained by applying a random orthogonal transformation to the features, in which case the data does not satisfy our underlying assumption by design. The different steps of this procedure and their justification are detailed below:

1. Apply our procedure with $K = 2$ to the given data matrix and store the associated score ( Eq. 8).

2. Generate multiple modified versions of the dataset by applying random orthogonal transformation to the features of the data. Each orthogonal transformation randomly mixes all the features in the dataset. If distinct feature partitions existed in the original data, they will be completely mixed after the transformation. Hence, our underlying assumption does not hold for these manipulated datasets.

3. Apply our procedure with $K = 2$ to each transformed dataset and store the associated scores ( Eq. 8).

4. Compare the score from step 1 to the distribution of the scores from step 3. If the score from step 1 is smaller than a chosen percentile of the scores from step 3 (e.g., 0.01), we conclude that the original data contains feature partitions with significantly distinct structures, suggesting that our assumption holds, at least to some extent.

## H   ADDITIONAL EXPERIMENTS

We conducted an additional validation experiment using a subset of the COIL-20 dataset Nene et al. (1996), consisting of images of three different cars captured at varying azimuths. While this dataset may not explicitly satisfy our method's theoretical assumptions, it provides an interesting test case for our approach.

We applied our algorithm to partition the pixels into $K = 2$ groups and analyzed the resulting t-SNE visualizations (Figure 12). The baseline t-SNE visualization using all pixels (left) reveals a closed loop structure corresponding to the object's azimuth. Notably, partition 2 (right) produces a more coherent representation of the azimuth, demonstrating also enhanced stability across different perplexity values. Interestingly, partition 1 effectively separates the images based on car identity, suggesting our method naturally discovered semantically meaningful features. The actual pixel partitions are shown in Figure 12c.

The experiment details are - The car objects that were used are objects number $2, 5, 18$. The dataset was standardized using the z-score transformation on each pixel across all observations. Then, we approximated the data using its 20 leading singular value component to get rid of noise. Finally, we ran our algorithm using a perplexity parameter of 10 in order to divide the features into two subsets ($K = 2$).

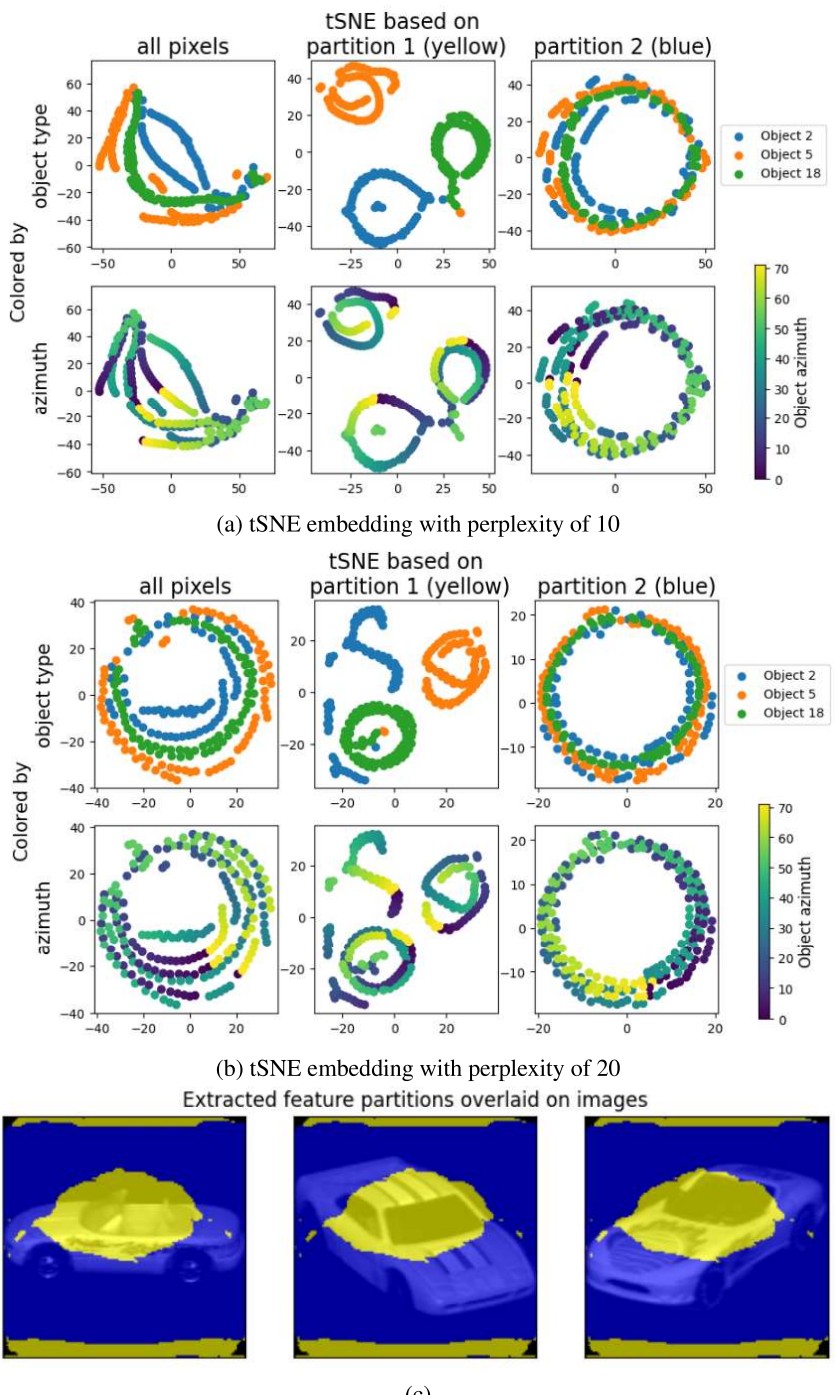

Figure 12: The experiment detailed in Appendix H. The dataset includes $N = 216$ gray-scale images with $D = 16217$ pixels from Coil-20. Each image contains a single rotated object, and the considered objects are three types of cars. (a)- tSNE embedding of the data perplexity parameter of 10, (b)- tSNE embedding of the data perplexity parameter of 20. (c)- Three sample images from the dataset, overlaid by the feature partitions of our method.