# OpenReview forum: "Partition First, Embed Later: Laplacian-Based Feature Partitioning for Refined Embedding and Visualization of High-Dimensional Data"
_ICLR.cc/2025/Conference — Submitted to ICLR 2025_

### Official Review · Reviewer_HqPj · 2024-10-24

**Soundness:** 2
**Presentation:** 1
**Contribution:** 2
**Rating:** 5
**Confidence:** 4

**Summary:**

This paper proposes a technique for separating the data features into several disjoint sets of features (dimensions) and to run tSNE on each of these independently. By doing this, their method learns multiple embeddings simultaneously which can more accurately represent the structure of the features. Much of the paper is devoted to deriving the corresponding optimization problem.

**Strengths:**

I have changed the score from a 3 to a 5

---------------------

This paper sets out to handle an interesting problem: that the embedding of datasets can be performed better if we decompose the dataset into disjoint groups of features. This is a fundamentally interesting problem that lends itself to robust analysis and principled insights. Within this, I really appreciate the development of a dataset which explicitly showcases the objective that the authors are seeking to optimize.

I also applaud the paper's theoretical approach to the topic. Many works in modern dimensionality reduction simply state an objective and optimize it and I find this paper's theoretical derivations to be the exact kind of work that the field is in need of right now.

Furthermore, the paper is aesthetically pleasing -- the images in particular are well-made.

**Weaknesses:**

I believe this paper has three major weaknesses. Although I give a lengthy description of these weaknesses, I do not mean it as a condemnation of the paper -- simply constructive criticism. I believe that if these issues are addressed then this would be a strong paper.

## Presentation and Theoretical Rigor

The theoretical results in this paper are messy. For context, I am very familiar with the related work and come from a theoretical background, so my concern is not that the topic itself is difficult. Instead, many theoretical ideas are presented with little context and the proofs seem to be hand-wavy at times. I include below an inexhaustive list of examples:

 - There are missing definitions. For example, the Laplace-Beltrami operator in equation 32 is never defined and it took me a while to recognize that this is what it represented. Similarly, page 22 references "Slater's conditions for strong duality" which are, to my knowledge, not introduced elsewhere. The same proof also mentions the KKT conditions. What are these?
 - The proof of Proposition 1 references the proof of Proposition 3, which appears 5 pages later in the appendix and one page later in the main body. In an ideal world, theoretical derivations should start at first principles and build up to things in a linear order without requiring one to jump around. However, more worryingly, the proof is simply "[the proof] can be derived from Proposition 3 by... " . But the proof of Proposition 3 simply says "this can be obtained by combining Lemmas 3 and 4." On its surface, it is not clear to me how this should be accomplished (and I've spent 5+ hours now trying to parse the derivations in the appendix). If Prop. 3 can be proven from these lemmas, then it should be shown how this is done. Thus, not only is the proof of Proposition 3 not clear, the proof of Proposition 1 is difficult to check by extension as well.
 - This difficulty in parsing the proofs is partially due to the fact that the lemmas in the appendix appear with no motivation or context. For example, Lemmas 1, 2, 3, and 4 all appear spontaneously and it is unclear to what end they will be used. It would be extremely helpful to have a "proof overview" as is often done in learning theory papers which describes how the lemmas come together to give the ideas.
 - Furthermore, many steps in the proofs are not explained. For example, equations 43-48 are... unclear to me. Specifically, we say equation 43 tends to equation 44 as $N \rightarrow \infty$, where the difference is that we replace the summation by an integral and multiply by $f(y)$ in the integral. This can't be true for all functions $f$, though, right? For example, let $f(y) = ||y||$. This is smooth and positive over the manifold and satisfies the conditions in the lemma statement. Then we have $$ \int_{y \in M} \exp(-||x - y||^2 / \epsilon_i) ||x - y||^2 ||y|| dy $$ in the numerator. Now let $f(y) = ||y||^{100}$. Then our numerator becomes $$ \int_{y \in M} \exp(-||x - y||^2 / \epsilon_i) ||x - y||^2 ||y||^{100} dy. $$ Clearly, these are different. So how can equation 43 tend to 44 under any $f$ if equation 44 depends on the $f$?
    - Similarly, it is unclear how we go from equation 45 to 46. How does the denominator get simplified out?
    - Lastly, there is a reference to Osher et. al between equations 47 and 48. This reference does not appear in the references of the main body and therefore I could not find the corresponding paper.
 - Furthermore, there are cases where the proofs seem to have mistakes? For example, equation 50 to equation 51 is incorrect. As I understand it, the log of a fraction has been separated into the difference of two logs and the distributive property should now be applied to this difference. However, the multiplication is only applied to one of the terms in the difference of the logs. If the distributive property is applied correctly, then the second term will also depend on $x_j$ and the derivative of the second term will not be 0. Thus, the proof is incorrect as it is currently written. Since Proposition 3 depends on Lemma 2 via Lemma 3 (and, subsequently, Proposition 1, Corollary 1, etc.), the remaining theoretical results of the paper are technically unsubstantiated as well.

## Experiments

Unfortunately, it is similarly difficult to understand what the experiments are showing and the experiments feel very incomplete to me.

 - How are the standard tSNE plots obtained in Figures 1 and 4? As described, the dataset has images which include all the pixels (not just those corresponding to a single figurine). So, to be clear, the left column of Figure 1C is the tSNE plot of *all* the pixels and the right columns are the tSNE plots of just subsets of the pixels? If so, then this took quite a while to understand as the color-coding implies that the top-left image in Figure 1C is the tSNE of the yoda figurine under some other, arbitrary partition while the bottom-left is the tSNE of the bulldog figure under the same arbitrary partition.
 - I'm confused what point Figures 5, 8 and 9 are trying to make. Is the premise that, under the partitioning approach proposed by the authors, the dataset's cyclic nature is evidenced when embedding with tSNE on partition 2? This seems poorly motivated to me. If the goal is to find cycles in the data, then I would guess that methods in topological data analysis are more effective and interpretable than partitioning + tSNE, so a comparison should be made there to point out why one should use the author's proposed method. If the point is simply that the visualization is better under partition 2 then I am unconvinced that this is the case. The tSNE embeddings based on "all genes" in figures 5 and 9 look rather good to me. What does partition 2's tSNE embedding provide which quantitatively improves over the "all genes" one? While I agree that the loop "looks" better, is this actually more representative of the input or just representative of what we would like the embedding to look like?
 - Since the authors' primary contribution is a method is for data segmentation with an application to visualization, then they have not conducted sufficient enough experiments to verify the necessity or validity of this approach. With regards to validity, it would be nice to measure against competitors. For example, they analyzed spectral clustering for the ellipsoid dataset but nowhere else. I would hypothesize that an equally valid competitor to the authors' work would be the following algorithm: Step 1: find a good partition of the images over all the data samples using spectral clustering. Step 2: apply this partition to the full dataset. Step 3: visualize each partitioned dataset using tSNE. I would be very interested to see how this performs compared to the authors' approach. Furthermore, if the authors believe that their approach outperforms my suggested competitor algorithm, then they should provide quantitative measures which emphasize this.
     - There is similarly an absent set of references to image segmentation techniques using neural networks. Would those not work for getting the partitions? They consistently outperform all laplacian based methods...
 - This paper is missing quantitative measures of the embedding quality more generally. It is difficult to say one embedding is "better" than another when there is not a quantitative evaluation of this fact.
 - Is there a reason that the authors did not use the coil-20 and coil-100 datasets? These are essentially identical to the authors' proposed figurines and standard datasets for evaluating tSNE, UMAP, and other dim. reduction approaches. Perhaps the coil dataset could help in providing quantitative comparisons of manifold preservation?

## Motivation

Some of the motivation for the technique seems artificial to me. Specifically, entire fields of related work are discarded as "inapplicable" with insufficient description/motivation.
 - On page 2 the authors state that their approach "does not require the observations in each feature partition to be clusterable". But that is required of their method, no? Problem 1 and Problem 2 explicitly define clustering objectives where each feature is placed into a disjoint partition (cluster). Thus, there is an absent set of comparisons to other clustering techniques. For example, the technique in [1] seems particularly relevant.
 - Similarly, page 2 states that subspace clustering (an entire field attempting to do the partitioning that the authors are performing) approaches are "less interpretable and the analytical properties of the solutions are not well understood". Could the authors explain how their techniques are more interpretable than subspace clustering [2]?

[1]: Spectral Clustering Based on Local PCA. Arias-Castro, Lerman and Zhang. 2017.
[2]: Subspace Clustering for High Dimensional Data: A Review. Parsons, Haque and Liu. 2004.

**Questions:**

I will select a few questions from the above weaknesses as the ones which can be responded to in the short reviewer-response period. The numbered elements are those which are more immediately pressing I list some additional things which would be nice-to-have but likely outside the scope of a reviewer-response period.
 1. Can the authors explain how Lemmas 1-4 come together to prove Propositions 3 and 1? A roadmap of the proofs would go a long way to making it easy to follow.
 2. Furthermore, could the authors address the issue in equations 50 -> 51 and verify that Lemma 2 still holds?
    - Similarly, I would appreciate an explanation of steps 43-48. Currently, these steps are completely unclear to me.
 3. For the experiments, could the authors compare against other approaches for partitioning the datasets before doing the tSNE embeddings? I am under the impression that other partitioning approaches (using spectral clustering or neural nets) should give equally good segmentations and therefore lead to comparable embedding quality.
    - Some quantitative measures of the embedding quality would go a long way. When comparing the embeddings visually, I am uncertain what I am looking for in a formal sense.
 4. A more detailed evaluation of the related work is in order. Specifically, the topics of subspace clustering seem very relevant but have been brushed off in the introduction.

---

> ### Author Response · Authors · 2024-11-22
> **Q1, Q2-part I**
>
> Just updated the page numbers.
> ____
> We appreciate the time and effort the reviewer invested in thoroughly reading the paper and the proofs. We are grateful for the acknowledgment that the problem we address is "fundamentally interesting." Additionally, we value the reviewer’s positive feedback on our theoretical approach, which highlights that the theoretical derivation presented in this paper is the type of work that the field currently needs.
>
> Next, we will answer the points the reviewer raised:
> # Questions
>
> **Q1)** Can the authors explain how Lemmas 1-4 come together to prove Propositions 3 and 1? A roadmap of the proofs would go a long way to making it easy to follow.
>
> **Response:** Proposition 1 and 3: We included a "proof overview" section within the revised manuscript on pages 21-22, outlining the proofs within the proofs section as the reviewer suggested. Furthermore, we reorganized the proof section, including the proofs for Sections 1.1 and 1.2, to align more closely with the order in which the material is presented in the main text.
>
> The proof of Proposition 1 has been revised to be almost self-contained relying only on Lemma 1 (Lemma 2 in the original manuscript). The proof follows the steps outlined in Lemma 3 which solves a more general problem.
>
> Regarding the proof of Proposition 3, we removed the text “this can be obtained by combining Lemmas 3 and 4” and added a detailed explanation of how Lemmas 3 and 4 are applied to obtain the required result. Specifically, the revised proof of Proposition 3 begins with the definition of two subproblems derived from Problem 1. Each subproblem focuses on minimizing the objective function with respect to one set of variables (either the graph matrices or the partitioning variables) while keeping the other fixed. Then, Lemmas 3 and 4 are applied to characterize the minimizers of these subproblems. By combining the results of these lemmas, we obtain a full characterization of the minimizers of Problem 1. Lemma 1 (which was Lemma 2 in the original manuscript) is an auxiliary result used for the proof of Lemma 3.
>
> **Q2- part I** Furthermore, could the authors address the issue in equations 50 -> 51 and verify that Lemma 2 still holds?
>
> **Response**
> We appreciate the reviewer's careful examination of the derivations, and we added text and intermediate steps within the proofs to make them more clear. In the next lines we hope to clear the misunderstanding.
> The derivation between Equation 50 to Equation 51 in the original manuscript is correct. However, a step that was omitted might have caused the confusion. In the revised manuscript, this derivation now spans Equation 33 to 36 on page 22, with additional intermediate steps included for clarity. To address the specific concern of the reviewer we provide further elaboration below.
>
> Below is the derivation made:
> $$
> \sum_{j=1}^N \left(\frac{exp(-\|  y_j\|^2 \beta)}{
>         \sum_{t=1}^N exp(-\| y_t\|^2 \beta)
>         }
>         \log
>         \frac{exp(-\|y_j\|^2 \beta)}{
>         \sum_{s=1}^N exp(-\|y_s\|^2 \beta)
>         } \right)
> $$
> $$
> =  \left(\sum_{j=1}^N \frac{exp(-\| y_j\|^2 \beta)}{
>         \sum_{t=1}^N exp(-\| y_t \|^2 \beta)
>         }
>         \log ( exp(-\|  y_j\|^2 \beta) )\right)  - \left(  \sum_{j=1}^N \frac{exp(-\| y_j\|^2 \beta)}{ \sum_{t=1}^N exp(-\| y_t\|^2 \beta)} log(
>         \sum_{s=1}^N exp(-\| y_s\|^2 \beta))
>          \right)
> $$
> $$
>          =  -\beta \left(\sum_{j=1}^N \frac{exp(-\| y_j\|^2 \beta)}{ \sum_{t=1}^N exp(-\| y_t\|^2 \beta) }
>         \| y_j\|^2  )\right)    - \left(
>         \frac{\sum_{j=1}^Nexp(-\| y_j\|^2 \beta)}{
>         \sum_{t=1}^N exp(-\| y_t\|^2 \beta)
>         }\right) log(
>         \sum_{s=1}^N exp(-\| y_s\|^2 \beta))
> $$
> $$
>           =  -\beta \left(\sum_{j=1}^N \frac{exp(-\| y_j\|^2 \beta)}{ \sum_{t=1}^N exp(-\| y_t\|^2 \beta) }
>         \| y_j\|^2  )\right)   -  log(
>         \sum_{s=1}^N exp(-\| y_s\|^2 \beta))
> $$

---

> ### Author Response · Authors · 2024-11-22
> **Q2 part II, Q3**
>
> just updated the page numbers
> ____
>
> **Q2 -part II** Similarly, I would appreciate an explanation of steps 43-48. Currently, these steps are completely unclear to me.
> **Response**  Equation 43-48  appear now as Eq. 74-82 on page 26 in the revised manuscript. We added more steps and text explaining them within the revision. We will now provide an explanation for each step below (the old equation number will appear in the brackets).
> *  Transition from Eq. 74(42) to 75(43): In this transition, we plug in the expression of $W_{i,j}$ from Eq. 1 into the equation.
> * Transition from Eq. 75(43) to 76(44): In this step, we let n tend to infinity, causing the  empirical mean to converge almost surely to its analytical value (based on Lemma 2 of [B] ). The function f here is the density over the manifold from which the data points $y_1,..,y_n$ were sampled. We inadvertently also used f to denote a general function in Lemma 2 (Lemma 1 in the old manuscript). To avoid confusion, we changed the notation in Lemma 2.
> * Transition from Eq. 76(44) to 77?(45): In this transition, the squared Euclidean distance is expressed as the sum of squared differences across the coordinates, i.e., $\|x- y\|^2 = \sum_{d=1}^D (x_d-y_d)^2$.
> * Transition from Eq. 77(45) to 78(46): In this step, we approximate the integrals in the numerator and denominator of the equation using the asymptotic approximation stated in Lemma 2.
> * Transition from Eq. 78(46) to 81(47): Here, we use the first order Taylor approximation of the term $1/(1+O(\epsilon_i))$ in the equation as $1/(1+O(\epsilon_i))= 1+O(\epsilon_i)$. Then, we can write
> $$(1+O(\epsilon_i))/(1+O(\epsilon_i)) = (1+O(\epsilon_i))^2 = 1+O(\epsilon_i).$$
> * Transition from Eq. 81(47) to 82(48): In this step we apply proposition 3.1 from ref. [1].
> [A] - Osher, S., Shi, Z., & Zhu, W. (2017). Low dimensional manifold model for image processing. SIAM Journal on Imaging Sciences, 10(4), 1669–1690.
> [B]- Hein, M., Audibert, J.Y., & Von Luxburg, U. (2005). From graphs to manifolds–weak and strong pointwise consistency of graph Laplacians. In International Conference on Computational Learning Theory (pp. 470–485).
>
> **Q3)** For the experiments, could the authors compare against other approaches for partitioning the datasets before doing the tSNE embeddings? I am under the impression that other partitioning approaches (using spectral clustering or neural nets) should give equally good segmentations and therefore lead to comparable embedding quality.
> Some quantitative measures of the embedding quality would go a long way. When comparing the embeddings visually, I am uncertain what I am looking for in a formal sense.
>
> **Response:** In the revised manuscript we included the results of K-means and spectral clustering on the pixels to Experiment 2. To assess the quality of the embeddings, we examined the overlap between the 50-nearest neighbors of each point defined by the extracted pixel partitions with the neighbors determined by the latent parameters that govern the data, specifically the ground truth angle of the rotating figurines. On page 16-18, we provide figures illustrating the pixel partitions obtained by the different methods and a table including a performance comparison table. Furthermore, on the bottom of page 9 of the main text we included a brief discussion on the performance comparison.
>
> Below we present the average results obtained by each partition method, highlighting the degree of overlap between the nearest neighbors (using 50 neighbors) in the partitioned space and the latent parameter space, which is the true angle of the corresponding figurine.
> |All features|FP(Ours)| K-means | Spectral Clustering |
> |---|---|---|---|
> |11.74%|85.1%|25.9%|24.7%|
>
> K-means and spectral clustering provide a slight improvement over the use of all the pixels altogether. However, our approach significantly outperforms these methods for partitioning the features. The fundamental reason is that the pixels are not clusterable according to standard distances. We illustrate the pixel partitions obtained by K-means and spectral clustering in Fig 8 (page 18), showing that they do not capture the figurines accurately and also mix between the pixels of the different figurines.
>
> Image segmentation: We believe that traditional image segmentation does not address our problem setting. Such techniques are usually referred to as methods that segment the pixels for each image independently, while our method partitions the features while considering all images together. Moreover, traditional image segmentation techniques rely on the spatial continuity of neighboring pixels within/outside the segments. In that sense, our method tackles a more general problem where we do not assume such prior knowledge.
> We note that the purpose of using images in Fig 1 and Experiment 2 is to demonstrate our algorithm intuitively, however our goal is to use this approach on complex high-dimensional data, such as single-cell data in Experiment 3.

---

> > ### Comment · Reviewer_HqPj · 2024-11-22
> >
> > Thank you for the improved theoretical descriptions and analysis. The proofs are significantly easier to follow.

---

> ### Author Response · Authors · 2024-11-22
> **Q4**
>
> **Q4)** A more detailed evaluation of the related work is in order. Specifically, the topics of subspace clustering seem very relevant but have been brushed off in the introduction.
>
> **Response:** We added a discussion about clustering on page 2 in the Introduction, a point that was specifically raised in one of the comments. However, subspace clustering typically focuses on techniques that cluster the samples and associate each cluster with a specific subspace. These methods assume that the data points can be partitioned into subsets with distinct geometric structure (a linear subspace), whereas our setting does not make such an assumption. As illustrated in the motivating example of Figure 1, all the samples (images) belong to a continuous geometric structure, which is the product of two closed curves in the pixel space. Thus, it is not possible to partition these samples into meaningful groups. Moreover, the output of our method is  fundamentally different from subspace clustering. Our method outputs a simple subset affiliation for each feature, whereas subspace clustering methods output subspace principal directions. These include a lot more parameters and are much less interpretable in high-dimensional space.

---

> ### Comment · Reviewer_HqPj · 2024-11-22
> **Regarding Spectral clustering and Subspace clustering**
>
> Forgive the original version of this comment -- I had not seen the figures on page 17. The original comment is below, but I believe the figures on page 17 address my concerns.
>
> ---------------- Original comment ------------------
>
> The points on subspace clustering are well-taken. However, while I completely agree that spectral clustering or image segmentation in its default form will not work in the given setting, I think it would be very easy to extend to the setting in the paper.
>
> Specifically, let each image be a point in $\mathbb{R}^d$, so that we have a dataset $X \in \mathbb{R}^{n \times d}$, where $n$ is the number of images and $d$ represents the numbers of pixels (if we are in the grayscale setting). Now consider that the rows of $X^\top \in \mathbb{R}^{d \times n}$ contain each pixel's values across the entire dataset. I argue that finding two clusters in this dataset ($X^\top$) is similar to what the authors propose.
>
> Thus, I disagree with the premise that spectral clustering or image segmentation techniques do not apply in this setting -- one can easily perform clustering or image segmentation across an entire dataset and find a consistent separation. I provide just three options for how one could do this:
> - The naive method would be to take a segmentation on one image's pixels and to apply it blindly to all the remaining images. One could use spectral clustering for this, as in Figure 1 of [1]. Since spectral clustering can (famously) give the partitions in Figure 1 of that paper and one can apply such a partition over all the images, I fail to see why this would struggle in the Yoda/Dog dataset the authors rely on.
> - A more sophisticated approach is to take two clusters (with, for example, spectral clustering) over the entire dataset $X^\top$.
> - A third method would be to find one partition for each image and to then use cluster ensembling techniques to find the best partition over all the images.
>
> Using any of the above techniques, one could then train tSNE on the two partitions. If one wanted to solve the problem posed by the authors, this would be the first thing one would try and it has not been made clear that it cannot work. Indeed, I would expect it to perform consistently with the authors' proposed approach on the chosen datasets. I am happy to be proven wrong.
>
> For full disclosure, I do not think that this is nearly as principled as the method the authors propose and I am sure one can find cases where it breaks (for example, if choosing a bad single image to apply across the remaining cases in the naive use case). My concern is more that the authors do not compare to any methods for segmenting a dataset of images and then running tsne on the segmented subsets. Thus, I am not asking for "state-of-the-art" against these methods. The alternative techniques I've proposed are unprincipled so, even if they perform equally to the authors' algorithm, I could be convinced that the authors' techniques are superior. But the fact that one can solve the paper's primary task in other ways, an analysis of these other techniques and their tradeoffs is missing from the paper in its current form.
>
> Given the improved readability of the theoretical derivations, if such an analysis is included (and done in good faith), then I would be willing to change my score for the paper.
>
> [1]: https://proceedings.mlr.press/v162/macgregor22a/macgregor22a.pdf (For clarity, I am *not* one of the authors of this paper and do not know the authors. This was just the first example of spectral clustering for image segmentation I could think of.)

---

> ### Author Response · Authors · 2024-11-23
> **Presentation and Theoretical Rigor- 1**
>
> We are pleased to have addressed the reviewer's concerns so far. If we understand correctly, we have responded to Questions 1 through 4. We would like to confirm that the reviewer has seen the following responses, as they appeared as the second part of our official replies:
> *  Q2- part I: regarding the validity of derivation of Eq 50 to Eq. 51, shown in the response titled “Q1, Q2-part I”.
> * Q3: regarding the quantitative measures we added to Experiment 2, shown within the response titled “Q2 part II, Q3”
>
>
> **P1)** There are missing definitions. For example, the Laplace-Beltrami operator in Equation 32 is never defined and it took me a while to recognize that this is what it represented. Similarly, page 22 references "Slater's conditions for strong duality" which are, to my knowledge, not introduced elsewhere. The same proof also mentions the KKT conditions. What are these?
>
> **Response:** We acknowledge the oversight regarding the missing definitions and citations within Lemmas 2 on page 25 (it was Lemma 1 in the original manuscript), Lemma 3 (on page 26-28) and Lemma 9 (on page 41-42), which created confusion. In the revised version, we have incorporated the definition of the Laplace-Beltrami operator into Lemma 2 and added the relevant citations for Slater's condition and the Karush-Kuhn-Tucker (KKT) conditions within Lemma 3 and 9. Additionally, we have included further explanations to clarify the application and context of these conditions.

---

> ### Author Response · Authors · 2024-11-23
> **Presentation and Theoretical Rigor- 2**
>
> **P2)** The proof of Proposition 1 references the proof of Proposition 3, which appears 5 pages later in the appendix and one page later in the main body. In an ideal world, theoretical derivations should start at first principles and build up to things in a linear order without requiring one to jump around. However, more worryingly, the proof is simply "[the proof] can be derived from Proposition 3 by... " . But the proof of Proposition 3 simply says "this can be obtained by combining Lemmas 3 and 4." On its surface, it is not clear to me how this should be accomplished (and I've spent 5+ hours now trying to parse the derivations in the appendix). If Prop. 3 can be proven from these lemmas, then it should be shown how this is done. Thus, not only is the proof of Proposition 3 not clear, the proof of Proposition 1 is difficult to check by extension as well.
>
> **Response:** We appreciate the reviewer's detailed feedback and the time invested in examining the proofs of Proposition 1 and Proposition 3.  We updated the proofs to be in a linear order so that Proposition 1 will appear before Proposition 3. The proof Proposition 1 and 3 were updated and now include much more text. See response to Q1 above for further details.

---

> ### Author Response · Authors · 2024-11-23
> **Presentation and Theoretical Rigor- 3**
>
> **P3)** This difficulty in parsing the proofs is partially due to the fact that the lemmas in the appendix appear with no motivation or context. For example, Lemmas 1, 2, 3, and 4 all appear spontaneously and it is unclear to what end they will be used. It would be extremely helpful to have a "proof overview" as is often done in learning theory papers which describes how the lemmas come together to give the ideas.
> **Response:** We added “proof overview” overview on pages 21-22 as suggested and organized the order of the proofs to be more linear. See response to Q1 above for more details.

---

> ### Author Response · Authors · 2024-11-23
> **Presentation and Theoretical Rigor- 4**
>
> **P4)** Furthermore, many steps in the proofs are not explained. For example, equations 43-48 are... unclear to me. Specifically, we say equation 43 tends to equation 44 as n-> infinity, where the difference is that we replace the summation by an integral and multiply by f(y)  in the integral. This can't be true for all functions f though, right? For example, let f(y)= ||y||^. This is smooth and positive over the manifold and satisfies the conditions in the lemma statement. Then we have
> \int_{y\in M} exp(-||x-y||^2/epsilon_i) ||x-y||^2 ||y|| dy
> in the numerator. Now let f(y)= ||y||^100. Then our numerator becomes
> \int_{y\in M} exp(-||x-y||^2/epsilon_i) ||x-y||^2 ||y||^100 dy
> Clearly, these are different. So how can equation 43 tend to 44 under any f  if equation 44 depends on the  f?
> Similarly, it is unclear how we go from equation 45 to 46. How does the denominator get simplified out?
> Lastly, there is a reference to Osher et. al between equations 47 and 48. This reference does not appear in the references of the main body and therefore I could not find the corresponding paper.
>
> **Response:** We note that f represents the density of the manifold from which the data points are sampled. For more details, refer to the answer to Q2 part II.

---

> ### Author Response · Authors · 2024-11-23
> **Presentation and Theoretical Rigor- 5**
>
> **P5)** Furthermore, there are cases where the proofs seem to have mistakes? For example, equation 50 to equation 51 is incorrect. As I understand it, the log of a fraction has been separated into the difference of two logs and the distributive property should now be applied to this difference. However, the multiplication is only applied to one of the terms in the difference of the logs. If the distributive property is applied correctly, then the second term will also depend on x_j  and the derivative of the second term will not be 0. Thus, the proof is incorrect as it is currently written. Since Proposition 3 depends on Lemma 2 via Lemma 3 (and, subsequently, Proposition 1, Corollary 1, etc.), the remaining theoretical results of the paper are technically unsubstantiated as well.
>
> **Response:** The proof is correct, and to make it easier to follow we added intermediate steps. For more details, refer to the answer to Q2 above.

---

> ### Author Response · Authors · 2024-11-23
> **Experiments -1**
>
> **E1)** How are the standard tSNE plots obtained in Figures 1 and 4? As described, the dataset has images which include all the pixels (not just those corresponding to a single figurine). So, to be clear, the left column of Figure 1C is the tSNE plot of all the pixels and the right columns are the tSNE plots of just subsets of the pixels? If so, then this took quite a while to understand as the color-coding implies that the top-left image in Figure 1C is the tSNE of the yoda figurine under some other, arbitrary partition while the bottom-left is the tSNE of the bulldog figure under the same arbitrary partition.
>
> **Response.**  To clarify the point raised by the reviewer, both plots in the left column show the same tSNE embedding based on all the pixels. The upper embedding is colored by the Yoda figurine’s angle (top left of Fig. 1C) and the lower by the Bulldog figurine’s angle (bottom left of Fig. 1C). In contrast, the right column displays the tSNE plots based on  the pixel partitions extracted using our approach, as visualized in Figure 1B. Specifically, the tSNE embedding based on the grey pixel partition (shown in Fig. 1B) is colored by the Yoda figurine’s angle (top right of Fig. 1C), while the tSNE embedding based on the yellow pixel partition is colored by the Bulldog figurine’s angle (bottom right of Fig. 1C).
> We recognize that this may have been confusing, and we have revised the figure caption to clarify this distinction (the same revisions were made to Figure 4B). This adjustment should resolve the misunderstanding and make the visualization more intuitive. Furthermore, we changed the colors used for the extracted partitions illustrated in Figure 1B  from blue and green to yellow and gray so that there will be no overlap between these and the color indicating the figurine angles (blue and green). Thank you for pointing this out!

---

> ### Author Response · Authors · 2024-11-23
> **Experiment 2**
>
> **E2)** I'm confused what point Figures 5, 8 and 9 are trying to make. Is the premise that, under the partitioning approach proposed by the authors, the dataset's cyclic nature is evidenced when embedding with tSNE on partition 2? This seems poorly motivated to me. If the goal is to find cycles in the data, then I would guess that methods in topological data analysis are more effective and interpretable than partitioning + tSNE, so a comparison should be made there to point out why one should use the author's proposed method. If the point is simply that the visualization is better under partition 2 then I am unconvinced that this is the case. The tSNE embeddings based on "all genes" in figures 5 and 9 look rather good to me. What does partition 2's tSNE embedding provide which quantitatively improves over the "all genes" one? While I agree that the loop "looks" better, is this actually more representative of the input or just representative of what we would like the embedding to look like?
>
>
> Response:
> The motivation for using our feature partitioning approach in Experiment 3 on single-cell data comes from the literature [C], which suggests that the genes can be partitioned into subsets that describe decoupled cellular processes. In particular, for this dataset, we expect one group of genes to describe a cell cycle (which has circular structure), and another group to describe dermal cellular development (which has a linear structure).
> When we apply our feature partitioning approach, we obtain one partition that corresponds almost perfectly to the cell cycle and another partition that corresponds to the cellular development (according to the ground truth labels). Moreover, this second partition is almost independent of the cell cycle phase (as shown in top plots of Figure 10 in page 20 within the revised manuscript, which was Figure 9 in the original). We included additional information in the text on page 10 to enhance clarity.
> Topological Data Analysis (TDA), in contrast, offers a global characterization of the data's structure, focusing on topological features like connected components, loops, and voids, without reference to specific data points or features. Furthermore, when applied to data with a product manifold structure (since different feature groups correspond to independent manifolds), TDA will provide a complex characterization of the entire dataset that will be difficult to interpret. By first applying our feature partitioning approach and then applying TDA to each feature partitioning, we can enable each TDA to uncover clearer and more interpretable insights. Thus, TDA is not a competing alternative to our approach but rather a complementary one that benefits from our feature partitioning, making its results more meaningful and easier to interpret.
>
> [C] - Tirosh, I., Izar, B., Prakadan, S., Wadsworth, M., Treacy, D., Trombetta, J., Rotem, A., Rodman, C., Lian, C., Murphy, G., & others (2016). Dissecting the multicellular ecosystem of metastatic melanoma by single-cell RNA-seq. Science, 352(6282), 189–196.

---

> ### Author Response · Authors · 2024-11-23
> **Experiment 3**
>
> **E3)** Since the authors' primary contribution is a method is for data segmentation with an application to visualization, then they have not conducted sufficient enough experiments to verify the necessity or validity of this approach. With regards to validity, it would be nice to measure against competitors. For example, they analyzed spectral clustering for the ellipsoid dataset but nowhere else. I would hypothesize that an equally valid competitor to the authors' work would be the following algorithm: Step 1: find a good partition of the images over all the data samples using spectral clustering. Step 2: apply this partition to the full dataset. Step 3: visualize each partitioned dataset using tSNE. I would be very interested to see how this performs compared to the authors' approach. Furthermore, if the authors believe that their approach outperforms my suggested competitor algorithm, then they should provide quantitative measures which emphasize this.
> There is similarly an absent set of references to image segmentation techniques using neural networks. Would those not work for getting the partitions? They consistently outperform all laplacian based methods..
>
> **Response:**  We refer the reviewer to the answer to Q3.

---

> ### Author Response · Authors · 2024-11-28
> **Experiment 5**
>
> **E5)** Is there a reason that the authors did not use the coil-20 and coil-100 datasets? These are essentially identical to the authors' proposed figurines and standard datasets for evaluating tSNE, UMAP, and other dim. reduction approaches. Perhaps the coil dataset could help in providing quantitative comparisons of manifold preservation?
>
> **Response.** Following the reviewer's suggestion, we have added an experiment using a subset of the COIL-20 dataset in Section H of the Appendix. In this experiment, we compare t-SNE applied to all features with t-SNE applied to each of our extracted partitions.
>
> Our results demonstrate that the t-SNE obtained from all features captures the circular structure of all objects in an overlayed embedding. In contrast, our method partitions the data such that one partition is better for separating the clusters while the other captures the underlying angles of the objects.
>
> Nonetheless, it is important to note that the COIL-20 dataset does not fully align with our data assumptions. Our method assumes that data can be divided into subsets of features that are either partially dependent or independent. In our original datasets, as shown in Fig. 1 and Experiment 3.2 (Fig. 4), the same figurines consistently appear in the same pixel regions across all images, which satisfies our data setup assumptions. Conversely, in the COIL-20 dataset, different objects occupy the same spatial location at various azimuthal angles, making the general structure of the data less clear. It is not evident that one can divide the features into meaningful subsets.

---

> ### Author Response · Authors · 2024-11-28
> **Motivation**
>
> # Motivation
>
> **M1)** On page 2 the authors state that their approach "does not require the observations in each feature partition to be clusterable". But that is required of their method, no? Problem 1 and Problem 2 explicitly define clustering objectives where each feature is placed into a disjoint partition (cluster). Thus, there is an absent set of comparisons to other clustering techniques. For example, the technique in [1] seems particularly relevant.
> [1]: Spectral Clustering Based on Local PCA. Arias-Castro, Lerman and Zhang. 2017.
>
> **Response.** We would like to explain the distinction between our approach and the traditional notion of clustering, especially when applied to the features (rather than the samples). In our framework, we propose to group features together if their values (across the samples) correspond to the same low-dimensional latent geometry, i.e., if they are determined by the same latent variables. For instance, in the experiments with the three rotating figurines (Figure 4), pixels that belong to figurines that share the same angle are grouped together, even if they correspond to visually different figurines (the bulldog figurine in Figure 4, which appears in two distinct views). Therefore, the features in our setup *do not* need to have similar values to be associated with the same partition, while it is *required* for traditional clustering. In fact, in our setup, one can add an arbitrary constant to each feature (the same value across all samples but different values for different features), and our approach would perform exactly the same (since pairwise distances between samples remain unchanged), while the features will no longer be clusterable in any traditional sense. We incorporated this discussion into the introduction section in the revised manuscript (see page 2).
>
> As mentioned in our previous response to the reviewer’s question 3 (Q3), we added a new experiment comparing our approach to traditional clustering methods when these are utilized for partitioning the features into disjoint sets (i.e., applied to the features rather than the samples). We show in Figure 8 and Table 1 that these traditional clustering methods substantially underperform compared to our approach. In particular, the resulting feature partitions from these alternative methods mix between the different figurines, and moreover, the accuracy of the recovered rotation angle for each figurine is substantially lower than the one obtained from our approach.

---

> ### Author Response · Authors · 2024-12-02
>
> As the discussion period is nearing its end, we would like to check if there are any remaining concerns regarding our paper. We hope these revisions address all concerns raised, and we would appreciate any further feedback before the discussion ends.

---

### Official Review · Reviewer_Sx3T · 2024-10-30

**Soundness:** 3
**Presentation:** 3
**Contribution:** 3
**Rating:** 8
**Confidence:** 4

**Summary:**

The authors propose an approach for dimensionality reduction that optimize a combined objective function from multiple similarity matrices/graphs that result from partitions the features. This eases known problems with classical techniques for embedding where extreme dimension reduction can distort results/large embedding dimension is required for faithful representation. They provide theoretical results characterizing the solution of their stated optimization problem as well as related asymptotic analysis. They provided experiments that examine the efficacy of their approach in real data.

**Strengths:**

originality: The approach of partitioning features first, and then embedding, is certainly original in the context of dimensionality reduction. (As the authors mention, feature partitioning was considered in the spectral biclustering literature, but the authors' version is more general)

quality: The quality of the mathematical derivations, the exposition, the motivation, and the experiments, are sufficient and satisfactory. The mathematical results are sufficiently deep and relevant, and the application use cases are clearly motivated and illustrated. The experimental results and comparisons are relevant and convincing.

 clarity: This is perhap the biggest strength of the paper. The mathematical content and derivations involving multi graph setting and Riemannian manifolds naturally look complicated, but the authors did a good job of exposition by first explaining the tradition construction and graph score and then generalizing it to the feature partitioning case. I enjoyed reading the paper and followed it without problem due to the exposition.

 significance: dimensionality reduction is an important area of study, and the practical technique introduced here (with theoretical analysis) is a welcome and substantial contribution to the area.

**Weaknesses:**

I list my questions and comments here:

1. How to choose K is an important consideration in this problem, but the authors do not seem to sufficiently address this. I do not expect any theory/simulations etc, but I would like to see a high level discussion that addresses choosing K, and how that could relate to dimensionality of the embedding space etc.

2. What is the practical computaitonal complexity tradeoff between the feature partition approach vs the traditional approach?

**Questions:**

n/a

---

> ### Author Response · Authors · 2024-11-28
> **W1**
>
> We thank the reviewer for your thoughtful and positive review of our paper. We greatly appreciate your recognition that our approach is “original in the context of dimensionality reduction” and more general than other feature partitioning methods. We are also grateful for your encouraging feedback on the quality of our mathematical derivations, exposition, and experiments, as well as your acknowledgment of the clarity of our presentation. Lastly, we are delighted that you enjoyed reading the paper and appreciate your kind words that our “practical technique … with (the supporting) theoretical analysis … is a welcome and substantial contribution to the area.”
>
> **W1** How to choose K is an important consideration in this problem, but the authors do not seem to sufficiently address this. I do not expect any theory/simulations etc, but I would like to see a high level discussion that addresses choosing K, and how that could relate to dimensionality of the embedding space etc.
>
> **Response.** To address the reviewer's concern, we added a discussion on choosing K in Section G within the Appendix. Specifically, we propose to inspect the behavior of the smoothness score from our approach (eq. (8)) as a function of the number of partitions K. As long as K increases but remains lower than the true number of partitions, we expect to see a rapid decay of the score. Then, when K reaches the true number of partitions, we expect to see an ‘elbow’, after which the score saturates or decays very slowly. This phenomenon is reminiscent of analogous ones when manually selecting the number of clusters in k-means or the number of components in PCA. We demonstrate this behavior of the smoothness score in Figure 11 of the revised manuscript (within section G). Automated techniques for selecting K will be developed in future work.
>
> In Section G within the Appendix we included a high-level discussion of the expected impact of over-selecting or under-selecting K on the resulting representation and embedding. If the number of partitions is lower than the number of true partitions in the data, we expect the partitions to separate the features into super-groups that correspond to the most distinct geometric structures, even if some of these groups could be further subdivided. In this case, each group of features describes a geometry that is simpler than that of the original data, i.e., it has a lower intrinsic dimension. Hence, the outcome of our procedure in this case still improves the ability to embed and visualize the data in a low-dimensional space, albeit suboptimally – where the embedding dimension may need to be higher than it would  be if the optimal choice of $K$ was used.
>
> If the number of partitions is slightly larger than the true number of partitions in the data, we expect that one or more of the true partitions will be arbitrarily subdivided. This will introduce redundancy into the representation, where some feature subsets will exhibit similar geometric structures. The user should take into account this possibility and inspect the resulting embeddings for potential redundancy. Nonetheless, the partitioning is still beneficial in this case, since the data for each partition can be easily embedded and visualized in a low-dimensional space. If the number of partitions is grossly overestimated, then in addition to redundancy, some of the feature subsets may not reliably represent the underlying geometry of any of the true feature subsets. However, we believe that this issue can be circumvented by following the guidance we provide in the revised manuscript for choosing K.
>
> Lastly, we included Section G at the end of the document so that any references made within this rebuttal will stay the same. In the camera-ready version it will appear after Section C.

---

> ### Author Response · Authors · 2024-11-28
> **W2**
>
> **W2)** What is the practical computational complexity tradeoff between the feature partition approach vs the traditional approach?
>
> **Response.** To address the reviewer's concern, we revised the section discussing the computational complexity of our approach (end of Section C) to include a comparison with the traditional approach where t-SNE is used to embed all data features.
> The computational complexity of constructing the similarity graph in t-SNE for N observation in dimension D is $O(N^2D)$. The computational complexity of obtaining K partitions and their corresponding similarity graphs using our approach is $O(KN^2D)$. Hence, the computational complexity of our approach incurs an additional factor of K, which is typically small. Nonetheless, our approach can be significantly slower due to the iterative procedure we employ for solving our optimization problem. To enhance scalability, we also describe an implementation that exploits a low-rank approximation of the data, which can considerably reduce the running time for large datasets.

---

### Official Review · Reviewer_GDbR · 2024-11-06

**Soundness:** 3
**Presentation:** 3
**Contribution:** 3
**Rating:** 6
**Confidence:** 3

**Summary:**

This paper addresses the problem of manifold learning and dimensionality reduction, specifically examining cases where the observed features can be divided into several disjoint subsets. The authors propose an approach that begins by partitioning the features into subsets that minimize a graph-Laplacian-based smoothness score within each partition. Subsequently, traditional manifold learning techniques are applied separately to each subset. As a result, this scheme captures the underlying structure of the observations more effectively than traditional methods that analyze the features as a whole. The method's effectiveness is demonstrated through several examples involving simulated and real datasets, also by presenting some theoretical analysis that supports the method's optimality.

**Strengths:**

- This paper is well-written and easy to follow.
- The problem of manifold learning when the features falls onto several disjoint groups is important and interesting, this is also well-motivated by an example shown in Figure 1.
- The proposed method that aims to finding the partitions by mimimize the Laplacian scores is straight-forward, and appears to be techniqually correct. Especially, I like the result in Proposition 1 that shows how the Laplacian score (Eq. 3) is connected with the Gaussian kernel. Besides, the theretical analysis is a good plus that consolidates the efficacy of the method.

**Weaknesses:**

My primary concern is with the experimental section. Since this paper relies on the assumption that the observed features comprise several disjoint subsets, it would be beneficial to address, or at least discuss, the following questions:

- How can we validate the existence of this assumption given a set of observed features? Answering this question could guide users in selecting the appropriate method for manifold learning.

- What are the results when this assumption does not hold? This is crucial for practical applications, as it is often challenging to confirm this assumption in real datasets (unlike the illustrative example with figurines shown in Figure 1).

- How does the method perform when the number of partitions is not optimal? In real-world applications, finding the ideal number of partitions may not be feasible, so it would be valuable to understand if sub-optimal choices still yield meaningful results. Although it may not be necessary to devise an automatic technique to determine the optimal number, a clear understanding of this hyperparameter would make the proposed method more accessible for practical use.

**Questions:**

N/A

---

> ### Author Response · Authors · 2024-11-28
> **W1**
>
> We thank the reviewer for their time, thoughtful feedback, and for reading our paper. We greatly appreciate their recognition of the partitioning problem as “important and interesting” and are pleased that they found the paper to be “well-written and easy to follow” as well as “well motivated by … Figure 1.” Additionally, we are grateful for their acknowledgment of our formulation as “original” and their appreciation of our efforts to “develop theoretical insight in terms of manifold approximation.”
>
>
> **W1)** How can we validate the existence of this assumption given a set of observed features? Answering this question could guide users in selecting the appropriate method for manifold learning.
>
> **Response.** To address the reviewer’s question, we added a discussion in Section G within the Appendix and a proposed procedure for validating our underlying model assumption. For simplicity, we focus on the case of K=2 (two partitions). To determine if the data features can be partitioned into meaningful subsets, we propose a type of a permutation test. Specifically, we propose to compare the smoothness score produced by our method from the original data to the analogous score obtained from manipulated versions of the data. The manipulated versions are obtained by applying a random orthogonal transformation to the features, in which case the data does not satisfy our underlying assumption by design. The different steps of this procedure and their justification are detailed below:
>
> 1. Apply our procedure with K=2 to the given data matrix and store the associated score (eq. (8)).
>
> 2. Generate multiple modified versions of the dataset by applying random orthogonal transformation to the features of the data. Each orthogonal transformation randomly mixes all the features in the dataset. If distinct feature partitions existed in the original data, they will be completely mixed after the transformation. Hence, our underlying assumption does not hold for these manipulated datasets.
>
> 3. Apply our procedure with K=2 to each transformed dataset and store the associated scores (eq. (8)).
>
> 4. Compare the score from step 1 to the distribution of the scores from step 3. If the score from step 1 is smaller than a chosen percentile of the scores from step 3 (e.g., 0.01), we conclude that the original data contains feature partitions with significantly distinct structures, suggesting that our assumption holds, at least to some extent.
>
>
> Lastly, we included Section G at the end of the document so that any references made within this rebuttal will stay the same. In the camera-ready version it will appear after Section C.

---

> ### Author Response · Authors · 2024-11-28
> **W2**
>
> **W2)** What are the results when this assumption does not hold? This is crucial for practical applications, as it is often challenging to confirm this assumption in real datasets (unlike the illustrative example with figurines shown in Figure
>
> **Response.** When the data's assumptions are not satisfied, we expect that applying our algorithm to partition the features into K groups will result in partitions with similar or nearly identical underlying structures. Consequently, the graphs that correspond to the partitions will also exhibit similar characteristics, leading to embeddings (such as t-SNE) that closely resemble each other. We discuss this point in G within the Appendix of the revised manuscript.
>
> Lastly, we included Section G at the end of the document so that any references made within this rebuttal will stay the same. In the camera-ready version it will appear after Section C.

---

> ### Author Response · Authors · 2024-11-28
> **W3**
>
> **W3)** How does the method perform when the number of partitions is not optimal? In real-world applications, finding the ideal number of partitions may not be feasible, so it would be valuable to understand if sub-optimal choices still yield meaningful results. Although it may not be necessary to devise an automatic technique to determine the optimal number, a clear understanding of this hyperparameter would make the proposed method more accessible for practical use.
>
> **Response.** We added a discussion to address this question in Section G within the Appendix. Sub-optimal choices for the number of partitions could be categorized into too few or too many partitions. If the number of partitions is lower than the number of true partitions in the data, we expect the partitions to separate the features into super-groups that correspond to the most distinct geometric structures, even if some of these groups could be further subdivided. In this case, each group of features describes a geometry that is simpler than that of the original data, i.e., it has a lower intrinsic dimension. Hence, the outcome of our procedure in this case still improves the ability to embed and visualize the data in a low-dimensional space, albeit suboptimally – where the embedding dimension may need to be higher than it would  be if the optimal choice of $K$ was used.
>
> When the number of partitions is slightly larger than the true number of partitions in the data, we expect that one or more of the true partitions will be arbitrarily subdivided. This will introduce redundancy into the representation, where some feature subsets will exhibit similar geometric structures. The user should take into account this possibility and inspect the resulting embeddings for potential redundancy. Nonetheless, the partitioning is still beneficial in this case, since the data for each partition can be easily embedded and visualized in a low-dimensional space. If the number of partitions is grossly overestimated, then in addition to redundancy, some of the feature subsets may not reliably represent the underlying geometry of any of the true feature subsets.
>
> To avoid the above issues, we provide guidance on how to choose K in Section G within the Appendix. Specifically, we propose to inspect the behavior of the smoothness score from our approach (eq. (8)) as a function of increasing numbers of partitions K. As long as K increases but remains lower than the true number of partitions, we expect to see a rapid decay of the score. Then, when K reaches the true number of partitions, we expect to see an ‘elbow’, after which the score saturates or decays very slowly. This phenomenon is reminiscent of analogous ones when manually selecting the number of clusters in k-means or the number of components in PCA. We demonstrate this behavior of the smoothness score in Figure 11 of the revised manuscript (In section G). Automated techniques for selecting K will be developed in future work.
>
> Lastly, we included Section G at the end of the document so that any references made within this rebuttal will stay the same. In the camera-ready version it will appear after Section C.

---

> ### Author Response · Authors · 2024-12-02
>
> As the discussion period is nearing its end, we would like to check if there are any remaining concerns regarding our paper. We hope these revisions address all concerns raised, and we would appreciate any further feedback before the discussion ends.

---

### Official Review · Reviewer_Zpyn · 2024-11-09

**Soundness:** 3
**Presentation:** 2
**Contribution:** 2
**Rating:** 5
**Confidence:** 3

**Summary:**

This paper introduces a new two-step approach that involves partitioning followed by dimensionality reduction. The partitioning is performed using a graph-based approach, drawing on the construction of the affinity graph from t-SNE.

**Strengths:**

The formulation is original and builds upon t-SNE, which is a widely used state-of-the-art dimensionality reduction approach. It relies on an adaptive kernel bandwidth, enabling it to capture heteroscedastic noise in the data.

Some efforts are made to develop theoretical insights in terms of manifold approximation. While I find it challenging to fully assess the relevance of this, the proofs appear to be sound.

**Weaknesses:**

I see two main weaknesses in this work that I believe the authors should address for the paper to meet the required standard:

1- A similar result to Proposition 1 already exists, specifically Proposition 1 in reference (a). This previous result is actually stronger than the one presented here, as it demonstrates that the entropy constraint can be relaxed into a lower bound while still achieving equivalence with the affinity matrix of t-SNE. Although the authors of (a) do not impose a null diagonal, one can simply add the null diagonal in the constraints to recover the matrix W tilde (the t-SNE affinity matrix). Considering such a relaxation in the present paper could add significant value by converting the problem into a convex one.

Reference (a):
Van Assel, H., Vayer, T., Flamary, R., & Courty, N. (2024). SNEkhorn: Dimension Reduction with Symmetric Entropic Affinities. Advances in Neural Information Processing Systems, 36.

2 - The experimental section is insufficient, and the datasets considered are too small. The authors only examine a dataset of 3,745 cells. Larger and more diverse datasets should be considered to strengthen the experimental results in this paper.

**Questions:**

- Can you relate the clustering mechanisms presented here to any known algorithms, such as kernel k-means with the kernel matrix W tilde ? This comparison could provide insights into the quality of the resulting clustering.

- If I understand correctly, Problem 2 not only includes entropy in the objective but also changes the adaptive bandwidth to a global scalar bandwidth. Does this not diverge too much from the original problem?

I believe it would be beneficial to include the algorithm in the main text rather than in the appendix.

---

> ### Author Response · Authors · 2024-11-23
> **Q1**
>
> We thank the reviewer for their time and thoughtful feedback, and we appreciate their recognition of our formulation as “original” and the acknowledgment of our efforts to “develop theoretical insight in terms of manifold approximation.”
>
> **Q1)** Can you relate the clustering mechanisms presented here to any known algorithms, such as kernel k-means with the kernel matrix W tilde ? This comparison could provide insights into the quality of the resulting clustering.
>
> **Response**  First, we want to emphasize that our approach solves a fundamentally different problem than clustering. Specifically, our approach aims to partition the features into K mutually exclusive subsets, such that in each partition, all observations (data points) exhibit a distinct smooth geometric structure. To this end, our approach learns multiple graphs over all observations, one based on each feature partition (see Problem 1 and Proposition 3 in Section 1.2). This is in stark contrast to traditional clustering techniques, which partition the samples into distinct groups using all features simultaneously. Since our approach involves forming multiple graphs over the observations, it is not directly comparable to any approach that uses a single affinity matrix or a kernel matrix (such as W tilde) over the samples. In our numerical examples (including Figure 1), the data is not clusterable in the traditional sense. In particular, all samples belong to a single connected component, where different feature subsets correspond to different geometric structures (which are continuous across all samples). Thus, traditional clustering will not work in these settings.
>
> A naive alternative approach to ours is to apply clustering on the features while treating the samples as coordinates, i.e., transposing the data matrix before feeding it to a traditional clustering algorithm. However, such an approach would substantially underperform compared to our method, as we exemplify in the revised manuscript; see Figure 7,8 and Table 1 in Section D.2 in the supplement. The fundamental reason is that the features in our example are not clusterable according to standard distances (e.g. Euclidean distance). Instead, our approach partitions the features by identifying which features share a smooth geometric structure. Previously, specialized tools have been developed to address this problem in domain-specific applications [A]. Here, we develop a principled framework to solve this problem in general settings with rigorous theoretical guarantees.
>
> [A] Qu, R., Cheng, X., Sefik, E., Stanley III, J., Landa, B., Strino, F., Platt, S., Garritano, J., Odell, I., Coifman, R., & others (2024). Gene trajectory inference for single-cell data by optimal transport metrics. Nature Biotechnology, 1–11.

---

> ### Author Response · Authors · 2024-11-23
> **Q2**
>
> **Q2)** If I understand correctly, Problem 2 not only includes entropy in the objective but also changes the adaptive bandwidth to a global scalar bandwidth. Does this not diverge too much from the original problem?
>
> **Response:** First, we want to clarify that while we use a global bandwidth parameter in Problem 2, the effective bandwidth of the resulting kernel is adaptive to the number of features in each partition (see Corollary 1). Indeed, the bandwidth here does not change across the data points within a given partition. However, this problem formulation provides a simpler surrogate that is easier to analyze than Problem 1 while approximating its landscape in the case of uniform (or approximately uniform) sampling density. Analyzing this surrogate problem allows us to gain important theoretical insights about the consistency of our approach in challenging high-dimensional settings. Moreover, we demonstrate in Figure 2 that both the analytic and empirical landscapes of Problem 2 closely mimic that of Problem 1.

---

> ### Author Response · Authors · 2024-11-23
> **Q3**
>
> **Q3)** I believe it would be beneficial to include the algorithm in the main text rather than in the appendix.
>
> **Response** We agree with the reviewer that including the algorithm within the main text would improve clarity. Due to lack of space, we included the pseudocode both in the Appendix and in the Github page of our software and refer to it within the main text.

---

> ### Author Response · Authors · 2024-11-23
> **Weakness 1**
>
> **W1)**  A similar result to Proposition 1 already exists, specifically Proposition 1 in reference (a). This previous result is actually stronger than the one presented here, as it demonstrates that the entropy constraint can be relaxed into a lower bound while still achieving equivalence with the affinity matrix of t-SNE. Although the authors of (a) do not impose a null diagonal, one can simply add the null diagonal in the constraints to recover the matrix W tilde (the t-SNE affinity matrix). Considering such a relaxation in the present paper could add significant value by converting the problem into a convex one.
> Reference (a): Van Assel, H., Vayer, T., Flamary, R., & Courty, N. (2024). SNEkhorn: Dimension Reduction with Symmetric Entropic Affinities. Advances in Neural Information Processing Systems, 36.
>
>
> **Response** Response: We would like to thank the reviewer for pointing out this reference. We agree that results similar to those obtained in Proposition 1 were obtained for related settings [1,2,3]. To address the reviewer’s concern, we added the suggested references as well as a remark about existing results on page 4 (in red). We would like to emphasize that we do not consider Proposition 1 as a primary novel contribution. The role of Proposition 1 is to justify the construction of the standard affinity matrix in t-SNE and then motivate our proposed problem formulation for feature partitioning and multi-graph learning.  Our main contribution lies in this extended problem formulation and its theoretical analysis in Section 2.
>
> We also appreciate the reviewer’s suggestion regarding convex relaxation. We highlight that the proposed relaxation cannot convexify our current problem formulation for feature partitioning stated in Problem 1. The proposed relaxation can only convexify the subproblem of optimizing over the graph affinity matrices while keeping the partition parameters fixed. However, this step already admits a closed-form solution, given by the graph construction step of t-SNE for each feature partition separately (see Proposition 3). Nonetheless, we consider the convexification of our feature partitioning problem as a promising future direction, which we mention in the discussion section of the revised version.
>
>
> [1] - Marco Cuturi. Sinkhorn distances: Lightspeed computation of optimal transport. In Advances in neural information processing systems, pages 2292–2300, 2013.
> [2]- Peyre, G., Cuturi, M., & others (2019). Computational optimal transport: With applications to data science. Foundations and Trends® in Machine Learning, 11(5-6), 355–607.
> [3] - Zass, R., & Shashua, A. (2006). Doubly stochastic normalization for spectral clustering. Advances in neural information processing systems, 19.

---

> ### Author Response · Authors · 2024-11-23
> **Weakness 2**
>
> **W2)** The experimental section is insufficient, and the datasets considered are too small. The authors only examine a dataset of 3,745 cells. Larger and more diverse datasets should be considered to strengthen the experimental results in this paper.
>
> **Response:** The main challenge with experimental data is to find datasets that are sufficiently rich and contain multiple latent processes on the one hand, but are also accompanied with high-quality ground truth information on the other hand. Without such ground truth, we cannot validate the results obtained from our approach. Finding large datasets of this type that are publicly available is not a trivial task. Nonetheless, we are currently engaged in exploring various options that might address the reviewer’s concern, and we will provide further updates in this regard.

---

> ### Author Response · Authors · 2024-12-02
>
> As the discussion period is nearing its end, we would like to check if there are any remaining concerns regarding our paper. We hope these revisions address all concerns raised, and we would appreciate any further feedback before the discussion ends.

---

### Author Response · Authors · 2024-12-04

We sincerely thank the area chair and all the reviewers for their valuable time and insightful feedback. Many of your suggestions have been thoughtfully incorporated into this revised version. Key revisions made:


1. **Presentation.** We have significantly improved the presentation of the paper, enhancing both the context and motivation, as well as the clarity of the figures. Specifically, we expanded the discussion of traditional clustering techniques within the context of feature partitioning in the Introduction. In Section 3.3, we strengthened the scientific rationale for applying our approach to the examined biological dataset used in the experiment. Furthermore, we revised Fig. 1 for improved clarity and updated the captions of both Fig. 1 and Fig. 4 for better understanding.


2. **Experiments.** We made the second experiment more comprehensive by including the results of traditional clustering techniques (spectral clustering and k-means) and providing a thorough quantitative comparison with our results (Sections 3.2 and D.2). Originally, such a comparison was only made in the experiment within Section 3.1. Additionally, we added an ablation study on a subset of the COIL-20 dataset (Section H).


3. **Proofs.** The Proofs section (Section F) was significantly enhanced. An overview of the proofs has been added (page 21), and the derivations within the proofs were substantially expanded, ensuring a much more detailed and clear presentation.


4. **Practical Implications and Considerations.** Section G has been introduced to discuss the practical aspects of our algorithm, providing readers with a clearer understanding of how to use it effectively. For example, it discusses how to determine the number of feature partitions, and how a suboptimal choice of this number affects the partitioning results produced by our algorithm.
(This section is placed at the end of the Appendix to preserve the paper's structure during the rebuttal stage but will appear near its beginning in the camera-ready version).


We hope these revisions address the reviewers’ concerns and improve the overall quality of the manuscript.

---

### Meta-Review · Area_Chair_LsrA · 2024-12-23

**Metareview:**

This paper provides new insights for visualizing (and embedding) high dimensional data using Laplacian feature partitioning. The reviews lean positive while being split along the border; authors agree on the merits of this paper but also had significant reservations. There was some engagement in the rebuttal period but no discussion ensuing. Overall, significant revisions such as overhauls and additions to the proofs of the main results suggest that this revised submission should undergo a full round of review. I encourage the authors to improve the paper accounting for the constructive comments of the reviewers

**Additional Comments On Reviewer Discussion:**

No reviewers discussed despite my comment

---

### Decision · Program_Chairs · 2025-01-22

Reject